# DNA-guided transcription factor interactions extend human gene regulatory code

Zhiyuan Xie[1,11], Ilya Sokolov[2,3,11], Maria Osmala[4], Xue Yue[1], Grace Bower[5], J. Patrick Pett[6], Yinan Chen[2,3], Kai Wang[1], Ayse Derya Cavga[2], Alexander Popov[7], Sarah A. Teichmann[8], Ekaterina Morgunova[9], Evgeny Z. Kvon[5], Yimeng Yin[1,10 ✉] & Jussi Taipale[2,3,4,9 ✉]

In the same way that the mRNA-binding specificities of transfer RNAs define the genetic code, the DNA-binding specificities of transcription factors (TFs) form the molecular basis of the gene regulatory code[1,2]. The human gene regulatory code is much more complex than the genetic code, in particular because there are more than 1,600 TFs that commonly interact with each other. TF–TF interactions are required for specifying cell fate and executing cell-type-specific transcriptional programs. Despite this, the landscape of interactions between DNA-bound TFs is poorly defined. Here we map the biochemical interactions between DNA-bound TFs using CAP-SELEX, a method that can simultaneously identify individual TF binding preferences, TF–TF interactions and the DNA sequences that are bound by the interacting complexes. A screen of more than 58,000 TF–TF pairs identified 2,198 interacting TF pairs, 1,329 of which preferentially bound to their motifs arranged in a distinct spacing and/or orientation. We also discovered 1,131 TF–TF composite motifs that were markedly different from the motifs of the individual TFs. In total, we estimate that the screen identified between 18% and 47% of all human TF–TF motifs. The novel composite motifs we found were enriched in cell-type-specific elements, active in vivo and more likely to be formed between developmentally co-expressed TFs. Furthermore, TFs that define embryonic axes commonly interacted with different TFs and bound to distinct motifs, explaining how TFs with a similar specificity can define distinct cell types along developmental axes.

In simple organisms, transcription is controlled by individual DNA-binding proteins that bind next to their specific target genes[3,4]. By contrast, higher organisms that have multiple distinct organs require the integration of positional information in three spatial dimensions to place organs in the right positions during development[4–7]. Integrating positional information requires highly specific and combinatorial information processing, which can be accomplished biochemically by cooperative binding of TFs to DNA.

The TF families that contribute to the formation of embryonic axes are well defined. For example, members of the homeodomain protein family are involved in specifying the embryonic anterior–posterior (A–P) axis, and are expressed differentially along this axis. However, the precise mechanism by which the differential expression is converted to different developmental outcomes is unclear. For example, the anterior homeodomain proteins (HOX1–HOX8) bind to identical TAATTA motifs, despite the fact that each protein has a distinct function during development[3]. This 'hox specificity paradox' is further exacerbated by the large number of other homeodomain proteins that are not part of the classical homeodomain gene clusters but that

nevertheless bind to the same TAATTA primary motif[8]. Furthermore, the disconnection between primary binding specificity and biological function is observed not only in the homeodomain family—all major TF families, with the possible exception of the KRAB-type zinc fingers, show higher diversity in biological function than in primary DNA recognition specificity.

Previous studies have established that the number of sequence elements that a TF can specifically recognize can be increased by the formation of cooperative TF–TF–DNA complexes[9–12]. Some TFs bind to DNA as pre-formed multimeric protein complexes; such complexes are typically dimers or trimers, consisting of members of the same family of TFs (for example, bHLH, bZIP and BTB domain zinc fingers)[11,13]. However, analyses of the formation of TF complexes on DNA have revealed that many TFs also bind together across the family boundaries, in a DNA-dependent or DNA-facilitated manner[14]. A classic example of such a TF–TF pair is the POU5F1 (OCT4)–SOX2 pair, which is involved in maintaining the pluripotency of embryonic stem cells; several other developmentally important TF–TF pairs have also been characterized[9,12,15]. Because the DNA-bound

[1]State Key Laboratory of Cardiovascular Diseases and Medical Innovation Center, Shanghai East Hospital, School of Medicine, Tongji University, Shanghai, China. [2]Department of Biochemistry, University of Cambridge, Cambridge, UK. [3]Generative and Synthetic Genomics Programme, Wellcome Sanger Institute, Hinxton, UK. [4]Applied Tumor Genomics Program, Biomedicum, University of Helsinki, Helsinki, Finland. [5]Department of Developmental and Cell Biology, University of California, Irvine, Irvine, CA, USA. [6]Cellular Genetics Programme, Wellcome Sanger Institute, Hinxton, UK. [7]European Synchrotron Radiation Facility (ESRF), Grenoble, France. [8]Department of Medicine and Cambridge Stem Cell Institute, University of Cambridge, Cambridge, UK. [9]Department of Medical Biochemistry and Biophysics, Karolinska Institutet, Stockholm, Sweden. [10]Clinical Center for Brain and Spinal Cord Research, Tongji University, Shanghai, China. [11]These authors contributed equally: Zhiyuan Xie, Ilya Sokolov. ✉e-mail: yy461@tongji.edu.cn; jt37@sanger.ac.uk

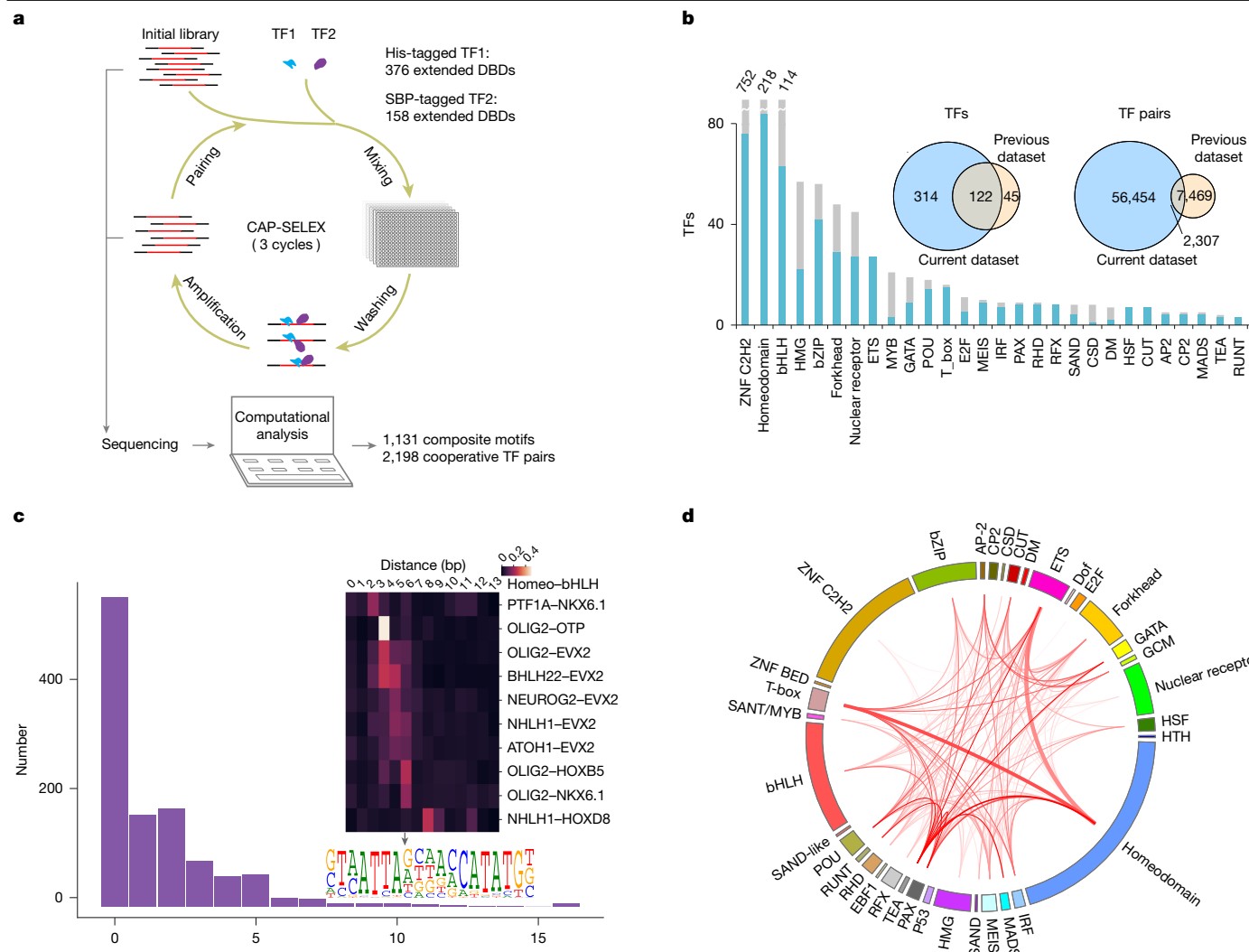

**Fig. 1 | Mapping of the DNA-guided interaction landscape between TFs.** **a**, Schematic description of the high-throughput CAP-SELEX process. A library of random sequences is incubated with two TFs, and the TFs are then purified consecutively, followed by amplification of the bound DNA and sequencing. The process is repeated three times, and the cooperative motifs are detected from the enriched sequences. DBD, DNA-binding domain; SBP, streptavidin-binding peptide. **b**, Coverage of TFs by structural family. Blue bars are this work; grey bars are all TFs[9]. Inset, percentage of overlap of individual TFs (left) and TF-TF pairs (right) in this study and in the previous study[9]. **c**, Most TF-TF interactions are short range. Histogram shows spacing preference between characteristic 8-mers across all interacting TF-TF pairs from CAP-SELEX. Inset, heat map of spacing preferences of characteristic 6-mers across selected bHLH-homeodomain pairs. Logo of OLIG2-NKX6.1 with a spacing of 5 is also shown. **d**, TF-TF interactions across structural TF families. Circos plot shows the prevalence of interactions across family boundaries. Note that TEA family TFs and C2H2 zinc finger proteins (ZNFs) participate in interactions more and less frequently than other TFs, respectively.

TFs are fixed at particular positions relative to each other, the contact surfaces required for cooperative DNA-facilitated binding are very small, and can evolve rapidly[16]. Therefore, the number of DNA-facilitated interactions is expected to greatly exceed the number of individual TFs, expanding the gene regulatory lexicon far beyond what could be accomplished by simple protein–protein interactions.

We previously described consecutive-affinity-purification systematic evolution of ligands by exponential enrichment (CAP-SELEX)[9], a method for identifying cooperative binding motifs for pairs of TFs in vitro, and applied it to 9,400 TF–TF–DNA interactions, discovering 618 TF–TF pair motifs. Here, to map the human TF–TF interactome, and to determine the molecular basis of the biological specificity of TF action, we have improved the throughput of CAP-SELEX, and used it to identify sequence-mediated, cooperative DNA binding across more than 58,000 TF–TF pairs.

## CAP-SELEX

To increase the throughput of the CAP-SELEX procedure, we adapted it to a 384-well microplate format (Fig. 1a). We then expressed a set of human TFs enriched in proteins that are conserved in mammals (Methods and Supplementary Table 1) in *Escherichia coli*, combined them into a total of 58,754 TF–TF pairs and analysed their interactions by CAP-SELEX (Methods). As a positive control, eight TF–TF pairs that are known to interact with each other were included on each 384-well plate (CEBPD–ETV5, CEBPD–ATF4, FOXO1–ETV5, FOXO1–GCM1, TEAD4–ONECUT2, TEAD4–CLOCK, HES7–TFAP2C and HES7–ETV5). The expressed proteins represented all major TF families (Fig. 1b and Supplementary Table 1; note, however that the focus on conserved TFs led to an underrepresentation of some subfamilies, such as KRAB family C2H2 zinc fingers). Three CAP-SELEX cycles were performed, and the selected DNA ligands were sequenced using a massively parallel

sequencer. To facilitate downstream analysis, we also determined the binding specificity of some of the individual TFs included in the assay using high-throughput SELEX (HT-SELEX) (Extended Data Fig. 1a). The resulting dataset (European Nucleotide Archive (ENA) PRJEB66722) is more than six times larger than that in our previous study[9] (for comparison, see Fig. 1b), and contains information about the TF pair interactions of more than 58,000 TF–TF pairs.

To evaluate the extremely large quantity of data generated, we developed two novel algorithms. The first algorithm is based on mutual information, which enables the identification of TF–TF pairs that show preferential binding to particular spacings and orientations relative to each other (Extended Data Fig. 1b; Methods). This method identified 1,329 interacting TF–TF pairs, including several known cases in which two TFs exhibit a preferred orientation and spacing, including HOXB13–MEIS1 and TEAD4–CLOCK (Extended Data Fig. 1b and Supplementary Table 2).

As the binding specificities of TFs can change when they bind DNA together, we also developed a second algorithm that is capable of detecting such novel composite motifs by comparing subsequence (*k*-mer) enrichment in CAP-SELEX with the enrichment observed in HT-SELEX experiments for the individual TFs (Extended Data Fig. 1c). This method could identify composite motifs that were partially or completely different from the individual TF specificities, such as FOXI1–ELF2 (Extended Data Fig. 1c). In total, 2,198 screened TF–TF pairs showed specific interaction, including 1,329 spacing and orientation preferences and 1,131 composite motifs (Fig. 1a and Supplementary Tables 2 and 3). Analysis of Encyclopedia of DNA Elements (ENCODE) chromatin immunoprecipitation followed by sequencing (ChIP–seq) data[17] also confirmed that in 42 out of 93 cases (45%), the composite motifs were more enriched in overlapping ChIP–seq peaks than in separate peaks for the individual TFs ($P < 0.012$; Extended Data Fig. 2a,b and Supplementary Table 2). Furthermore, more than half of the composite motifs could also be recovered by mixture-SELEX, in which the two TFs are simply mixed, and SELEX is performed using the mixed sample (Extended Data Fig. 2c–g), indicating that the composite motif formation is robust to experimental design, and not specifically caused by conditions of the CAP-SELEX procedure.

## Global analysis of TF–TF interactions

To determine whether there is a generally preferred spacing of individual TF-binding sites among the pairs that show spacing and orientation preferences, we analysed the interacting TF–TF pairs for preferred spacings and orientations between the characteristic *k*-mers contained in their individual motifs[9] (Extended Data Fig. 1b). For this, we performed a global analysis of the data from the *k*-mer mutual-information-based method described above to identify the optimal spacing across the interacting TF–TF pairs in all orientations with respect to each other. We then averaged the mutual information across all of the pairs for which we detected an interaction. This analysis revealed that short binding distances are generally preferred (Fig. 1c); distances of more than 5 bp between the TFs' characteristic 8-mer sequences were rare. The few observed longer-range interactions were also often weaker than the shorter-range interactions, with the exception of some cases, such as BACH2 and LMX1A, which cooperatively bind to DNA over 8-bp and 9-bp gaps (Extended Data Fig. 2h). Notably, the spacing preferences were often specific, with different members of the same family preferring different spacings with the same or related partners (Fig. 1c, inset).

Analysis of all types of interactions across TF families revealed that TF–TF interactions commonly crossed TF family boundaries (Fig. 1d). Consistent with our previous study, some TF families such as TEA (TEAD TFs) were very promiscuous in their interactions. C2H2 zinc finger TFs, by contrast, seemed to have fewer interactions than other TF families ($P < 1.51 \times 10^{-93}$). Despite this, many strong interactions could still be found between C2H2 zinc finger TFs and TFs of other structural families

(Extended Data Fig. 2i). For example, the GLI proteins GLI2 and GLI3, which respond to Hedgehog signalling[18], bound to DNA together with RFX3 (Extended Data Fig. 2i). RFX3 is a regulator of genes that is important for the formation of primary cilia[19] – cellular structures that are required for Hedgehog signalling[20,21].

Clustering of TFs on the basis of their interactions revealed that many TFs in structural families showed similar interaction profiles (Extended Data Fig. 2j and Supplementary Table 2), and that many motifs formed by TF–TF pairs were similar to each other. On the basis of the observed similarities, the spacing and/or orientation preferences could be grouped into three main types: family, subfamily and paralogue group-specific interactions (Supplementary Table 2).

## Analysis of the composite motifs

To identify previously published motifs, we compared our collection with that of the JASPAR database, and determined the most closely similar motif for each new dimer composite motif using TOMTOM. This initial assessment revealed several previously known composite motifs, including TBX4–HOXC10, ETV1/2–FIGLA, and FOXO1–FLI1 (refs. 9,22,23) (Extended Data Fig. 3a). To compare the motifs with the previously published CAP-SELEX motifs, we calculated similarities across the novel motifs and composite motifs from a previous study[9], and drew a graph of the similarities (Extended Data Fig. 3b). This analysis indicated that 169 novel composite motifs were identified in this study.

Similar to the spacing and/or orientation preferences, the composite motifs could also be classified to family, subfamily and paralogue group-specific motifs (Supplementary Table 2). For example, paralogues of FOXK and FOXI forkhead proteins interacted with the ELF paralogue group of class I ETS factors to form a specific type of composite motif that was different from those formed by other forkhead proteins such as FOXA1 (Extended Data Fig. 3c). Whereas the spacing and/or orientation preferences were often family or subfamily specific, the composite motifs were more commonly restricted to paralogue groups (Fig. 2a; $P < 3.3 \times 10^{-7}$). For example, most anterior HOX proteins interacted with TBX21 (Extended Data Fig. 2j and Supplementary Table 2). However, the PITX subclass of HOX proteins specifically interacted with FOXAs and FOXDs (Fig. 2b and Extended Data Fig. 3d), and paralogues of HOX2 (HOXA2 and HOXB2) interacted specifically with PROX1 and PROX2 (Fig. 2b, Extended Data Fig. 3e and Supplementary Table 2).

## Composite and single motifs are distinct

We next assessed the similarity of the composite motifs and individual TF motifs using MoSBAT. As expected, almost all of the composite motifs were clearly different from the individual motifs (Extended Data Fig. 3f). Because TFs bind to distinct sequences and not to motifs per se, we also calculated the score for each motif towards its consensus sequence, and compared that with the best match score on the corresponding composite motif consensus. This analysis revealed that the affinity of the individual TFs to the composite consensus was relatively low. This could be explained by changes in the sequence preference flanking the core motif of the TFs. Specifically, Jaccard index analysis of *k*-mers and Jensen–Shannon divergence (JSD) analysis of divergence of the composite motifs from the individual motifs revealed that the central region of the composite motif – which corresponded to the 'inner' flanking sequence of overlap between the two TF motifs – was commonly different to what one (89% by Jaccard and 56% by JSD) or both (54% and 24%) of the individual TFs would prefer to bind (Extended Data Fig. 3g–i). In some cases, such as ELF1–FOXK2, we observed even larger differences, and on visual inspection the composite motifs also appeared quite different from either one or both of the individual motifs (Fig. 2c). Despite the clear difference, the ELF1–FOXK2 motif was also highly enriched in overlapping ELF1 and FOXK2 ChIP–seq

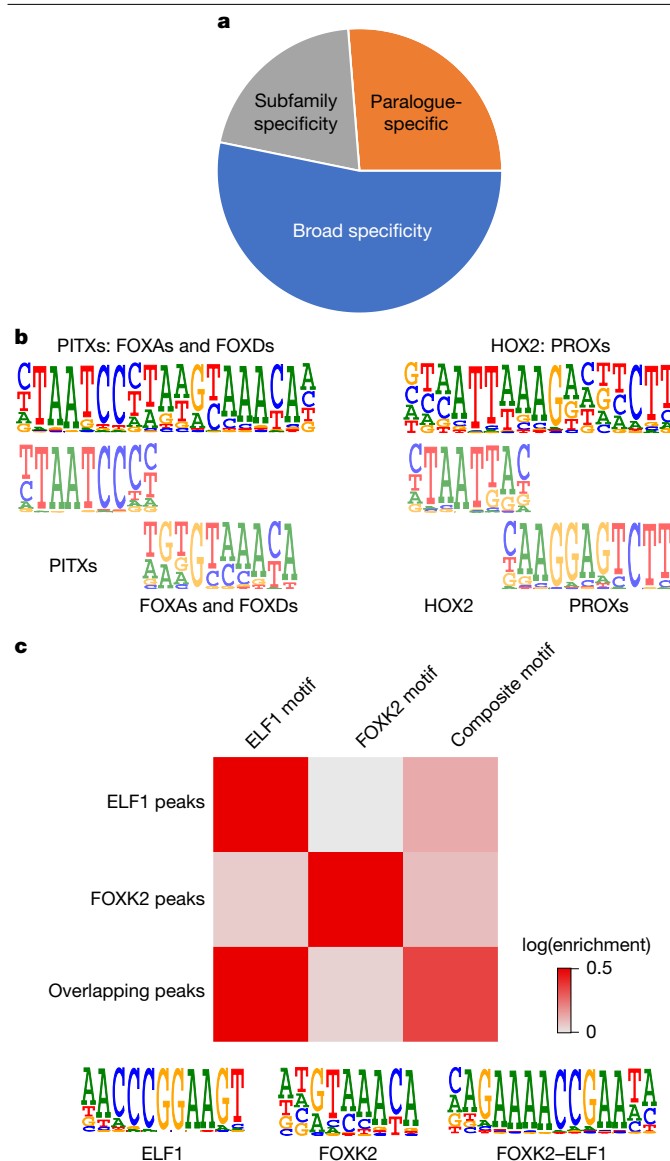

**Fig. 2 | Specificity of TF–TF pairs. a**, Close to half of all composite motifs are either specific within TF paralogue groups (orange) or subfamilies (grey). The remaining interactions are observed across broader structural families (blue). **b**, Specific composite motifs are formed between PITX and the FOXA/FOXD subfamily, and between HOXA2 and HOXB2 with the PROX subfamily. Alignment of individual motifs with composite motifs is shown. **c**, The indicated FOXK2–ELF1 composite motif detected by CAP-SELEX is enriched in overlapping ChIP–seq peaks for FOXK2 and ELF1. Red colour indicates enrichment of motif matches.

peaks, but not in the non-overlapping peaks (Fig. 2c), indicating that even such novel motifs that do not resemble the individual motifs are bound by TF–TF pairs in vivo.

## Pioneer factors and developmental TFs

To determine whether TFs with specific biological or biochemical functions showed different types of interaction from those of other TFs, we first analysed known pioneer factors that can induce transdifferentiation between cell lineages[24]. We identified several specific interactions between the known pioneer factors and other TFs (Fig. 3a, Extended Data Fig. 4 and Supplementary Data 1).

Of note, members of some classical pioneer factor families, including FOXA1 (ref. 25) and SOX11, have many partners, but other pioneer

factors, including GATA3 and GATA4, CEBPs, PAX6 and PAX7, and SOXs other than SOX11, are more selective in their choice of partners. In summary, pioneer factors differ in their promiscuity of partner choice, and as a class, the pioneer factors did not have more specific interactions than other TFs.

Second, we analysed interactions of TFs that are known to have roles in axial patterning across dorsoventral (D–V) and A–P axes. Analysis of TFs involved in patterning of the neural tube along the D–V axis[7,26] revealed that most specific cell types had specific TF–TF pairs that bound to composite motifs (Fig. 3b). Similarly, many interactions were identified for TFs that specify the A–P axis[27,28] (Extended Data Fig. 5). For example, a highly paralogue-specific interaction was identified between HOX5 and the bHLH protein OLIG2 (Extended Data Fig. 5), and HOX2 proteins (HOXA2 and HOXB2) and prospero-like homeodomain protein PROX1 (Fig. 2b).

## TFs with activation domains

Studies have shown that only a small minority of human TFs have strong transcriptional activation domains[29,30]. Notably, many TFs with strong activation domains, including TCF4, ATOH1, SRF, and HOXB2, specifically interacted with other TFs (NFKB2, POU2F2, ONECUT1, and PROX1 and PROX2, respectively; Fig. 3c). In total, only 16 of the tested TFs had strong activation domains[30]. However, 171 TFs interacted with them, indicating that TFs that do not have activation domains can commonly interact with other TFs to recruit strong transcriptional activation domains to DNA.

To test whether the composite motifs that are bound by TFs containing activation domains drive gene expression in vivo, we made reporter constructs containing six motifs[31], and performed a transgenic enhancer–reporter assay[32] in embryonic day (E) 11.5 mouse embryos. The construct containing HOXA2–PROX2 motifs drove strong and reproducible expression of a LacZ reporter in the apical ectodermal ridge, forebrain, midbrain, hindbrain, facial mesenchyme and otic vesicle (Fig. 3d and Extended Data Fig. 6). This motif is also found on the conserved enhancer of *Prox1* itself, which represses haematopoiesis in the lymphatic vasculature[33] (Fig. 3e).

GLI2 and GLI3, which can act as activators in the presence of Hedgehog ligands[20,21,34], formed a composite motif with RFX3, a regulator of cilia-specific genes[35]. The GLI3–RFX3 composite motif drove expression in the ventral midbrain, neural tube, forebrain and zone of polarizing activity (ZPA) of the limb buds, consistent with locations of Sonic hedgehog (*Shh*) expression at this stage[36] (10 out of 11 embryos; Fig. 3d and Extended Data Fig. 7). By contrast, other composite motif reporters that contained GLI- or RFX-binding sites showed no activity (Extended Data Fig. 7), indicating that RFX or GLI motifs are not sufficient for expression. Given that many Hedgehog pathway components are localized in the primary cilia in vertebrates[21], the GLI–RFX3 composite motif might have a role in regulating Hedgehog signalling.

## Conservation of motifs

Matches to TF–TF interaction motifs have been shown to be conserved in mammals (Fig. 3e), indicating that they are biologically important and affect organismal fitness[9,37]. To determine whether the novel interactions we discovered also have a biological role, we analysed the conservation of the corresponding motif matches in mammals (using conservation data from the Zoonomia Consortium[38]; Supplementary Table 4). To analyse orientation and spacing preferences, we mapped 300,000 motif matches of the primary motifs of the TFs to the human genome, and then compared the conservation of the preferred spacings and orientations with a null distribution derived using scrambled control motifs. This analysis revealed that of 1,291 spacings and orientations tested, 165 were significantly conserved (compared to 116 expected by random, $P < 5 \times 10^{-6}$; total matches 2,273,

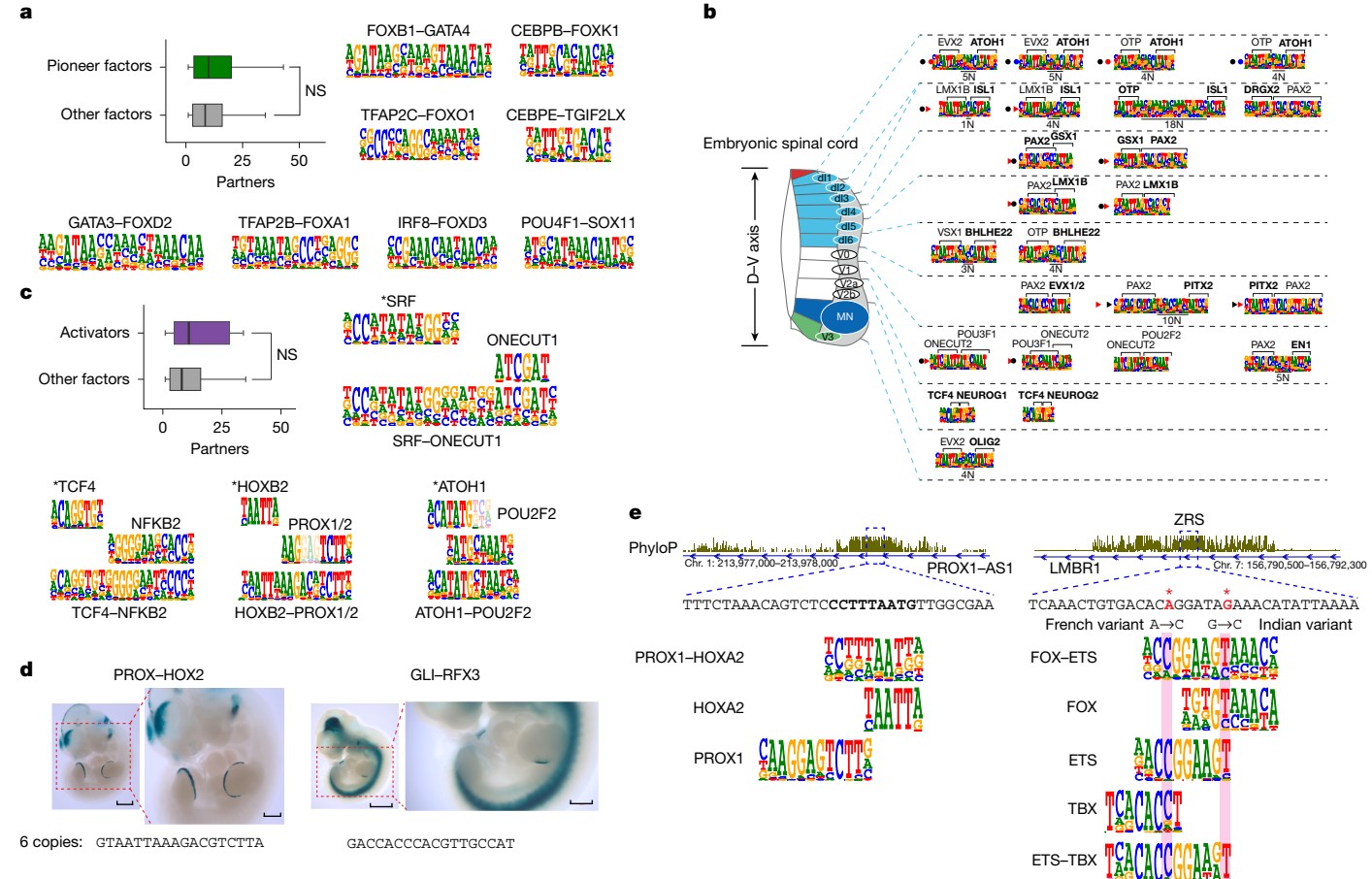

**Fig. 3 | Biological roles of the composite motifs. a**, Pioneer factors differ in their ability to form composite motifs. Left, box plot showing the number of interactions between known pioneer factors (green; $n = 36$) compared with other TFs (grey; $n = 257$). Note that on average, there is no statistically significant difference (two-sample $t$-test, two-sided) between pioneer factors and other TFs, but that pioneer factors vary a lot in their number of interacting partners. Right, examples of composite motifs between known pioneer factors and other TFs. NS, not significant. **b**, Almost all specific cell types formed across the developing spinal cord have their own composite motifs that form between the lineage-specific TFs. Oriented and palindromic motifs are indicated by arrowheads and filled circles, respectively. Bold typeface indicates TFs known to regulate establishment of cell identity of the distinct neuron subtypes. MN indicates motorneurons, and numbers after V and dl indicate specific types of ventral and dorsal interneurons, respectively. **c**, TFs with strong activation domains (indicated by asterisks) differ in their ability to form composite motifs. Left, box plot showing the number of interactions between TFs with strong activation domains (purple; $n = 16$) compared with other TFs

(grey; $n = 277$). Again, the average number of interactions between activator TFs and other TFs is not significantly different (two-sample $t$-test, two-sided), but activator TFs show high variance in the number of partners. Right, examples of pairs between TFs with strong activation domains. Note that HOXA2 and HOXB2, the homeodomains with a strong activation domain, but not other homeodomain proteins such as LMX1A, HOXB5 and HOXD10, form a highly specific complex with prospero homeodomains PROX1 and PROX2 (PROX1/2). **d**, Composite motifs of TFs containing a constitutive (PROX–HOX2) and regulated (RFX3–GLI3) activation domain drive highly specific expression in transgenic E11.5 mouse embryos. Scale bars, 1 mm (left images); 500 μm (right (magnified) images). **e**, Composite motifs potentially explain the conservation of developmental enhancers. Left, PROX–HOX2 motif match in a conserved PROX1 enhancer. The sequence matching the PROX–HOXA2 motif is indicated in bold. Right, FOX–ETS and ETS–TBX motif matches in the ZRS enhancer. Mutations that affect limb development (positions indicated by red asterisks) are expected to increase the affinity of the composite sites[50]. Illustration in **b** is adapted from ref. 7 under a CC BY 4.0 licence.

compared with 1,394 expected by random), indicating that the interactions discovered using CAP-SELEX have an effect on organismal fitness (Extended Data Fig. 8).

To analyse the conservation of composite motifs, we first identified the subset of novel motifs that represented distinct specificities using minimum dominating set analysis (Methods) of 3,933 SELEX-derived TF motifs. This yielded 347 representative motifs, which we mapped to the human genome, and derived conservation scores for each base pair. We then compared the conservation of the motif match sequences with control sequences matching artificial motifs. Out of the matches to 347 distinctly different composite motifs, 81 (family-wise error rate (FWER) < 0.05) were conserved more than matches to control motifs (Methods). We also computed conservation score correlations across each base position within the motif matches and compared them with

the corresponding correlations in the set of scrambled control motif matches (Methods). For 17 composite motifs, the conservation of one half-site of the matches significantly increased the probability that the matches of the other half-site would also be conserved (Extended Data Fig. 8), indicating that the conservation of the composite motif is not simply the result of conservation of matches to motifs of one of the partners.

## Structural analysis

Because there are more than 1.3 million possible TF–TF pairs, it will be difficult to experimentally measure the cooperativity constants for all of them. For that reason, it would be useful if computational predictive models could be built; for example, using modern machine-learning

tools. To assess the feasibility of predicting TF–TF interactions using currently available tools, we collected a test set of known structures that correspond to protein dimers (MYC–MAX, CEBPD–ATF4 and FOS–JUN) and two DNA-facilitated dimers (MEIS1–HOXB13 and MEIS1–DLX3). We also generated an unseen test set by solving six structures that represent several TF–TF–DNA complexes bound to optimal DNA sequences. The structures included two BARHL2 homodimers with two spacings, HOXB13 homodimer and TEAD4–HOXB13, MEIS1–HOXB13 and FOXK1–ELF1 heterodimers (Supplementary Table 5).

We first used the AlphaFold v.2.0 multimer function (Methods) to predict complexes for some indicative TF–TF pairs for which the structure was either previously known or solved by us here. AlphaFold can predict structures of protein complexes, but does not predict DNA structure; therefore, we predicted the structure of the TF–TF pair, and compared it with the protein component of the corresponding experimentally determined TF–TF–DNA structure. This analysis correctly predicted the structure of the protein component of several known heterodimeric TF–TF–DNA complexes such as bHLH dimer MYC–MAX (root mean squared deviation (r.m.s.d.) 0.428 Å versus Protein Data Bank (PDB) ID 1NKP; Fig. 4a). The ability to predict the structures in the absence of DNA was consistent with these TF–TF pairs binding to DNA as a pre-formed dimer. However, AlphaFold 2 was unable to predict the interaction of the protein components of DNA-facilitated and DNA-dependent TF–TF complexes such as MEIS1–DLX3 (Fig. 4a).

Recently released versions of RoseTTAFold (RoseTTAFold2NA v.0.2)[39] and AlphaFold (AlphaFold v.3.0)[40], which can predict the structures of protein–nucleic acid complexes (Methods), performed much better in predicting the overall geometry of TF–TF–DNA complexes (Fig. 4b). However, they still failed in most cases to correctly predict preferred spacings and orientations of TF–TF pairs bound to DNA (Fig. 4b, Extended Data Fig. 9a,b and Supplementary Table 6), owing potentially to their inability to predict key TF–TF and TF–DNA contacts that depend on the orientation and spacing of the TF pairs. This conclusion was supported by detailed analysis of the structures we had solved. For example, both RoseTTAFold2NA and AlphaFold 3 correctly predicted the two homodimers of BARHL2. However, only AlphaFold 3 was able to predict the structure of the HOXB13 homodimer. In predicting heterodimers, AlphaFold 3 succeeded in predicting the overall geometry of the HOXB13–TEAD4, FOXK1–ELF1, MEIS1–HOXB13 and DLX3–MEIS1 heterodimers; RoseTTAFold2NA could predict only HOXB13–TEAD4 and FOXK1–ELF1. Neither program was able to correctly predict the contacts between FOXK1–ELF1, MEIS1–HOXB13 and DLX3–MEIS1 complexes that contribute to the composite motif formation (Fig. 4c).

## Biological roles of the TF–TF pairs

The biochemical evidence that two TFs interact on DNA does not as such mean that the interaction has a key role in a biological process. Because there are more than 1.3 million possible interactions, it is expected that the number of interactions will greatly exceed the number of biological functions. To assess the role of the identified pairs in human development, we compared our TF pair matrix with a co-expression matrix from Human Cell Atlas (https://www.humancellatlas.org/). Co-expressed TFs formed composite motifs significantly more frequently than random pairs of motifs (Fig. 5a; $P < 6 \times 10^{-14}$). For example, many co-expressed TFs that are involved in limb development[41,42] formed specific composite motifs (Fig. 5b and Extended Data Fig. 10a). Similarly, in the intestine[43], the intestine-specific homeobox (ISX) was co-expressed with and formed composite motifs with ATF4, ONECUT2 and ETV4 (Extended Data Fig. 10b).

We next analysed the enrichment of the TF–TF pair motifs in cell-type-specific cis-regulatory elements (CREs) identified using single-cell assay for transposase-accessible chromatin with sequencing (ATAC-seq)[44]. Analysis of the enrichment of motifs in cell-type-specific open chromatin regions derived from the cis-regulatory atlas (CATlas)[44]

revealed that the spacing preferences were also observed in vivo. The TF pair motif matches were commonly enriched in cell-type group-specific candidate CRE sets (cCRE sets hereafter), with 211 of the 347 representative composite motifs and 74 of the 112 representative spacing motifs being enriched in at least one cCRE set (log₂-transformed fold change > 0.75; $P < 0.01$). Motifs with particular spacings were also enriched in specific cCRE sets. For example, bHLH–homeodomain TF pairs in which the motifs were spaced 4 bp apart were heavily enriched in cCRE sets from stromal and fibroblastic cells[16], whereas the motifs spaced 3, 5 and 6 bp apart were enriched in cCRE sets from pancreatic β cells, astrocytes and glutamatergic neurons, respectively (Fig. 5c). On the basis of logistic regression analysis, the composite motif matches could also be used to predict which elements were specific for particular cell types (Fig. 5d); however, it should be noted that individual motifs were also predictive, indicating that specific elements can also be built using TF–TF spacings and orientations that are not optimal biochemically. These results indicate that the discovered TF–TF pair motifs are biologically relevant, and that they are commonly enriched in cell-type-specific regulatory elements.

## Discussion

We report here that the interaction landscape between DNA-bound TFs is much richer than that of the TF proteins in solution. Our work greatly extends the gene regulatory lexicon, by introducing novel composite motifs and spacing and orientation preferences between TFs. By analysing more than 58,000 TF–TF pairs, we found that approximately 3.7% of the tested pairs interacted specifically in the presence of DNA. We also found that most TF–TF interactions evolve faster than individual motifs: many TFs that bind to essentially identical motifs alone showed specificity of partner choice, or orientation or spacing in such a way that they could recognize specific sequences that could not be bound by closely related TFs. Most interactions we observed were specific to paralogue groups or subfamilies of TFs; some also represented common preferences across TF structural families. The motifs we discover here were active in vivo and more likely to be formed between developmentally co-expressed TFs. Their matches were also conserved in mammals and enriched in cell-type-specific enhancers, suggesting that several of the newly discovered motifs could have a role in integrating positional information during development.

The large number of TF–TF interactions that require DNA is consistent with biophysical principles. The binding of proteins to each other in solution requires a relatively large interaction surface to overcome the loss of rotational and translational entropy that accompanies binding. Binding of the TFs to DNA, however, results in loss of the same entropy, enabling even interactions that involve few amino acids to contribute to substantial cooperativity between TF pairs. As humans have more than 1,600 TFs, assuming the 3.7% interaction rate, the more than 1.3 million pairs of human TFs are estimated to participate in around 50,000 interactions, in which the TF pairs bind to motifs that cannot simply be explained by analysis of monomeric motifs. In this work, we used around 10% of all human TFs as bait proteins, and studied the interactions of each of these proteins with around 25% of all human TFs. Although the total sample represents only around 4% of all potential interactions, the fact that most interactions were shared between members of paralogue groups or even entire TF families indicates that the coverage of all interactions in our screen is much higher than 4%. On the basis of subsampling analysis of the data (Methods), we estimate that our screen discovered between 18% and 47% of human TF–TF composite motifs.

Most (1,329 out of a total of 2,198) of the interactions we discovered represented spacing preferences between motifs that were highly similar to the respective individual TF motifs. The interactions generally occurred at a short distance, whereby the motifs were closer than 10 bp from each other. In addition to the spacing and orientation preferences,

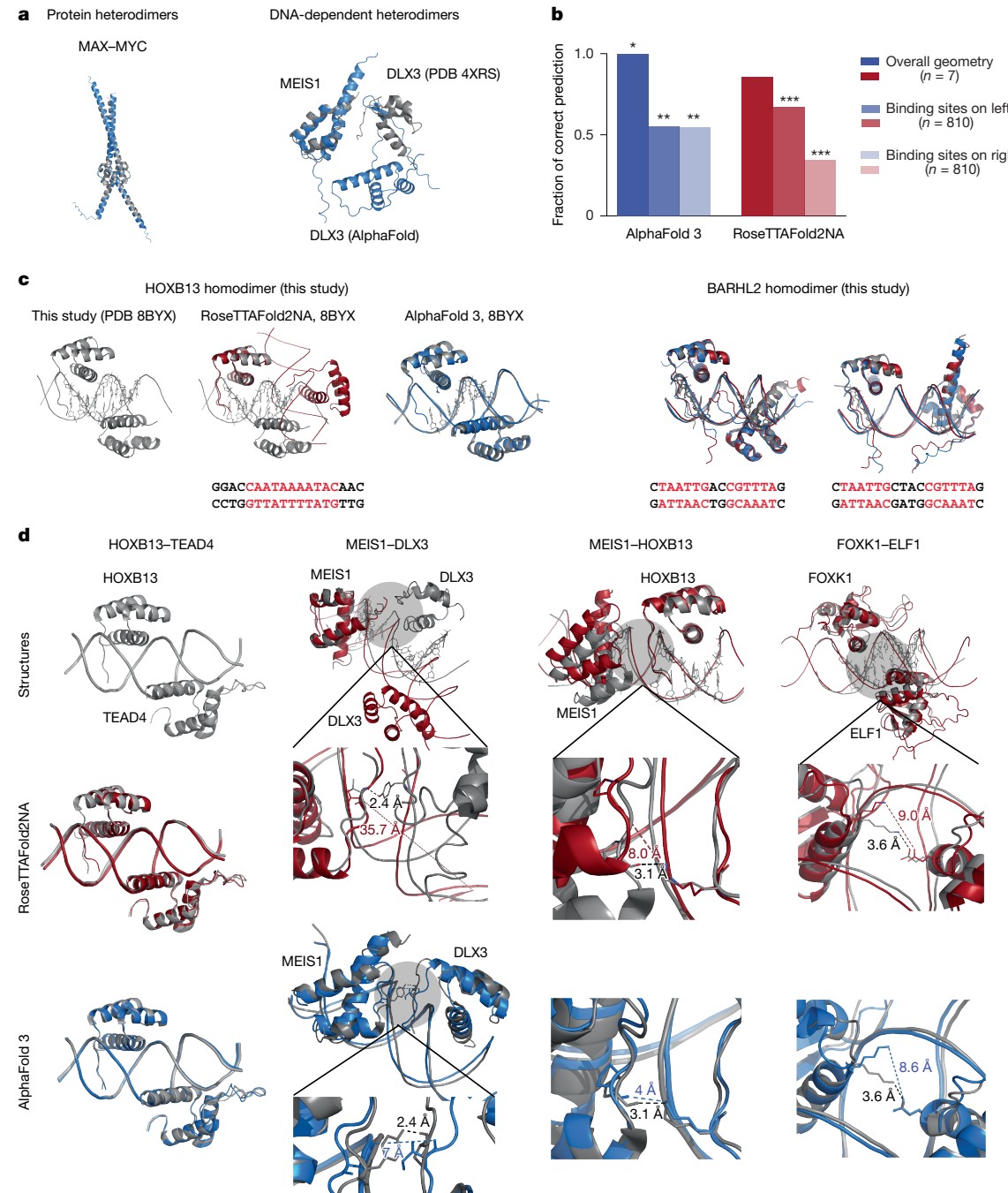

**Fig. 4 | Structural analysis shows that very small TF–TF contact surfaces result in high levels of cooperativity. a**, AlphaFold 2 can accurately predict protein-level heterodimers (left) but fails to predict DNA-dependent dimers (right). Alignments between predicted structures (blue; 4XRS refers to the PDB ID) and crystal structures (grey) are shown. **b**, Overall accuracy of computational prediction of TF–DNA structures. Both AlphaFold 3 (blue) and RoseTTAFold2NA (red) can predict the overall geometry of TF–DNA complexes (n = 7; dark-coloured bars; individual details shown in other panels), but are not much better than a random guess (probability = 0.5) at predicting which side of DNA a TF–TF pair binds to if both an optimal sequence and a suboptimal sequence are included on the same double-stranded DNA. To assess bias, the optimal sequence was placed on either the left (light-coloured bars; n = 810) or

the right (very light bars; n = 810) side of the DNA. One, two and three asterisks indicate P < 0.05, P < 0.01 and P < 0.001 (binomial two-sided test), respectively. **c**, Right, when given a locally optimal DNA sequence, RoseTTAFold2NA (red) and AlphaFold 3 (blue) can predict the overall geometry of BARHL2 bound to DNA (crystal structure in grey) in two different preferred spacings. For HOXB13 dimer (left), RoseTTAFold2NA (red) fails, whereas AlphaFold 3 (blue) predicts the overall structure correctly. DNA sequences used for the analyses are shown below, with TF motifs indicated in red. 8BYX refers to the PDB ID. **d**, Although the overall geometry of TF binding is predicted correctly for heterodimers, both RoseTTAFold2NA (red) and AlphaFold 3 (blue) fail to correctly position the side chains of the amino acids that form the contacts between the shown cooperative TF pairs.

we also found 1,131 composite motifs, 391 of which were distinctly different from each other. The composite motifs of most TFs were markedly different from individual motifs of the same factors, suggesting that

many individual TFs bind weakly, if at all, to the composite site. This affects the mechanism by which TFs find their sites, as in order for the pair to find its composite motif, at least one of the TFs should have

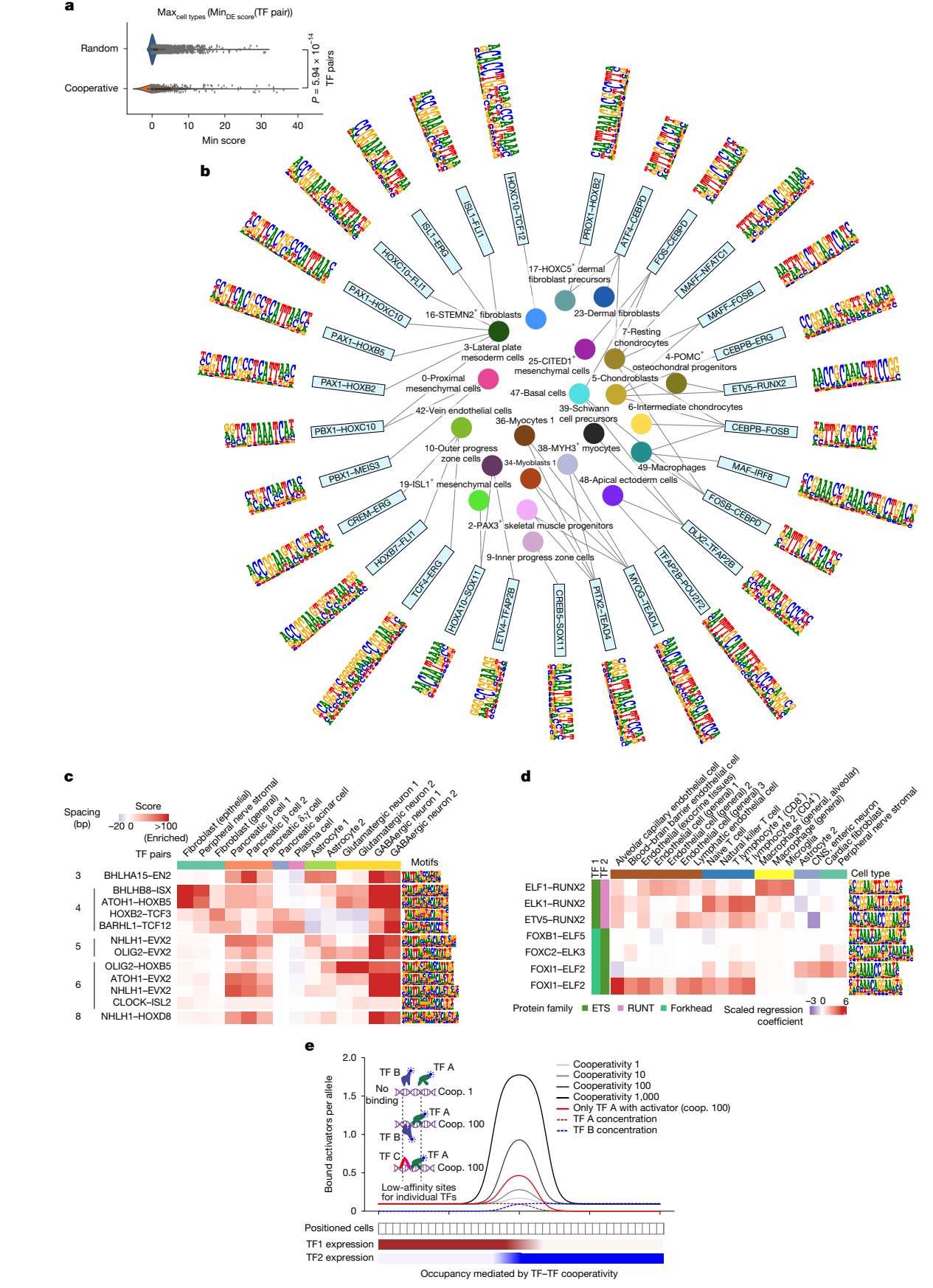

**Fig. 5** | See next page for caption.

**Fig. 5 | Co-expressed TFs form composite motifs. a**, TFs co-expressed in the limb form composite motifs more frequently than random sets of TFs. Minimum differential expression $Z$ score ($Min_{DE\,score}$) of both TFs in a pair is shown. For each TF pair the maximum value across cell types was selected ($Max_{cell\,types}$) from differential expression of each cell type versus all others, reflecting the existence of a cell state in which both TFs are upregulated. Interacting TF pairs (Cooperative) are compared to randomly sampled TF pairs from the same set of TFs (Random). Wilcoxon rank-sum test, two-sided, no multiple testing correction. **b**, Composite motif logos for the limb-specific TF pairs. **c**, Cell-type-specific open chromatin regions are heavily enriched in matches of specific motif spacing patterns. Heat map shows scores of $-\log_{10}(P)$ for enrichment and $\log_{10}(P)$ for depletion of the indicated bHLH–homeodomain motif matches in open chromatin regions specific for the indicated cell types (Fisher's exact test, one-sided, no multiple comparison adjustment). Enrichment is defined as the frequency of matches to cell-type-specific elements divided by the frequency of matches to the set of all tissue-specific open chromatin regions. **d**, Logistic regression analysis showing that TF–TF composite motifs can predict tissue-specific elements. Note that the specificities of the composite motifs of ETS factors differ according to the binding partner. The normalized regression coefficients are shown. CNS, central nervous system. **e**, Composite motifs enable efficient integration of signals and high cooperativity. The model shows the occupancy of TF activation domains ($y$ axis) as a function of the expression of two TFs (red and blue) under different conditions. The level of two TFs varies in cells located at distinct positions ($x$ axis). Inset, schematic representation of how the binding of two TFs (TF A and TF B) is affected by binding cooperativity. Note that despite highly specific pairing with TF B on one DNA, TF A can interact also with TF C on another DNA sequence.

sufficient affinity to the composite motif to facilitate binding of the other partner. Motifs in which both half-sites are weak, in turn, require a search process in which the two TFs bind to each other before binding to the composite motif.

Several groups have reported that clusters of low-affinity motifs are important in development and plasticity[45,46]. This illustrates the fact that biological optimality and biochemical optimality are not the same thing; during development, it is necessary to carefully calibrate the response of particular regulatory elements to changes in TF activities, and it is therefore not always beneficial to use the most biochemically active sequence. However, given that most composite motifs contain low-affinity sites for one or both members of the TF pair, and that developmental enhancers are commonly conserved outside of known individual TF motifs, care should be taken to establish that the flanking sequences of any biologically active low-affinity motifs do not contain elements that enable high-affinity cooperative binding of TF–TF pairs. The advantage of using composite motifs instead of individual low-affinity motifs in developmental control of gene expression is fourfold: (1) combining two TFs enables integration of positional information; (2) low affinity of individual TFs to the composite motifs allows dual TF functionality: a TF can act as a partner, and also independently on its own motifs to affect other housekeeping or developmental functions; (3) because high-affinity composite motifs can be occupied at a lower TF concentration than individual low-affinity motifs, using composite motifs allows increased specificity of gene expression; and (4) the change in specificity associated with composite motif formation greatly increases the cooperativity coefficient of binding of the two TFs to the composite site, which enables genes to respond very sharply to changes in TF concentration (Fig. 5e). Consistently with these biochemical advantages, we find here that composite motifs are present in highly conserved developmental enhancers. Notably, we found that specific composite motifs were formed between marker TFs of almost all cell types formed during perhaps the best-studied dorsoventral patterning event in development—the specification of neuronal cell types in the spinal cord. The utility of composite motifs in development is directly related to their biochemical cooperativity; this allows the gene regulatory activity of the composite motif to be more than the sum of the activities of the individual TFs. It also enables more specific gene expression, more exact cell-fate determination across a developmental field and the formation of sharper developmental boundaries. However, because the composite motifs and specific spacings and orientations of TF pairs have a higher information content than individual motifs, their use is likely to be more common in elements that require highly optimized and/or specific activities (for example, signal integration during development).

Our work also has implications for the broader molecular recognition field. Several algorithms have been developed that can partially solve the sequence-to-structure problem, by predicting protein structures from sequences and multiple sequence alignments with high accuracy[47,48] (see, however, ref. 49). In initial analysis, it seemed that these state-of-the-art structural prediction tools were able to accurately predict only a very small subset of TF–TF–DNA structures. There remains substantial room for algorithms to be developed; for example, by increasing the weight of the minor differences in the structure of DNA grooves that are sensed by TFs when they identify their motifs. Despite this, it is likely that the problem of solving TF–TF–DNA structures based on sequence remains limited mostly by data rather than by algorithm. Compared with 227,344 structures in the PDB (November 2024), much fewer protein–DNA structures are available, of which only 42 are sequence-specific TF heterodimer–DNA complexes. Furthermore, many protein–DNA structures are solved using DNA with suboptimal affinity, which makes it difficult for programs to learn the optimally bound sequences. Moreover, solving the gene regulatory code (that is, the sequence-to-expression problem) requires solving the sequence-to-affinity problem—determining the binding affinities of all macromolecules to each other on the basis of their sequence. Without knowing parameters of protein–DNA affinity, it is impossible to learn how DNA sequences facilitate TF–TF interactions across the human TFome. Given that measuring all the parameters would be an exceptionally large task, it would be beneficial to develop tools that can predict TF–TF–DNA complex affinities on the basis of sequence. However, this will require more training data than has been available in the PDB or in TF–DNA interaction databases. Because the work we present here contains more information about TF–TF–DNA interactions than the entire published literature, we believe that it also provides a unique dataset for building models aimed at solving the second genetic code—the code that determines when and where genes are expressed.

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

# Methods

## Clones and protein expression and purification

Cloning, protein expression and purification were performed essentially as described previously[9]. Most coding sequences for the expression constructs encoded[9] extended DNA-binding domains (eDBDs) that enriched expected motifs in a previous study[51] (Supplementary Table 1). New eDBD constructs were designed as described previously[9] and synthesized by GenScript. Gateway recipient vectors for protein expression were constructed using pETG20A modified to include an N-terminal 6×His tag[51] (prey construct), or pDEST15 (Invitrogen, 11802014) modified to include a C-terminal streptavidin-binding peptide (SBP; bait construct). The prey constructs were from a previous study[51], bait constructs were generated by Gateway cloning to the modified pDEST15 bait vector and the activities of the produced proteins were evaluated using HT-SELEX. In total, 158 SBP-tagged bait and 419 His-tagged prey TF proteins were expressed in *E. coli*.

The proteins were expressed using the auto-induction protocol as described previously[9]. In brief, the Rosetta 2(DE3) pLysS strain of *E. coli* (Millipore, 71403) and Invitrogen BL21-AI Competent *E. coli* (Thermo Fisher Scientific, C607003) were used for the expression of eDBDs that were inserted into pETG20A-6×His and pDEST15-SBP vectors, respectively. Transformed cells were first cultured at 37 °C in 500 µl TB medium in the wells of a 96-deep-well plate overnight, and then transferred into auto-induction medium (1:40 dilution) supplemented with either 0.05% glucose and 0.2% lactose (Rosetta 2(DE3) pLysS *E. coli*) or 0.05% glucose and 0.02% arabinose (BL21-AI *E. coli*). The cells were then cultured at 37 °C for 4–6 h, followed by 30 h at 16 °C. The cells were collected by centrifugation and resuspended in either buffer A (300 mM NaCl and 10 mM imidazole in Tris-Cl, pH 7.5; Rosetta 2(DE3) pLysS *E. coli*) or PBS (BL21-AI *E. coli*). Both buffer A and PBS were supplemented with 0.5 mg ml$^{-1}$ lysozyme and 1 mM PMSF. The cells were then subjected to two freeze–thaw cycles to ensure lysis, followed by addition of DNase I and MgSO$_4$ to 10 µg ml$^{-1}$ and 1 mM final concentrations, respectively, to digest the released DNA. The Rosetta 2(DE3) pLysS *E. coli* lysates were incubated with Ni-Sepharose 6 Fast Flow resin (Cytiva, 17-5268-01), and the BL21-AI *E. coli* lysates with glutathione sepharose 4B (Cytiva, 17-0756-01) for 1 h with 1,200 rpm shaking, after which the lysates were transferred into individual wells of a 96-well filter plate (Nunc, 278011). Ni-Sepharose beads were washed three times each with 600 µl buffer A with 10 mM and 50 mM imidazole, and glutathione sepharose beads with 600 µl PBS. The bound proteins were eluted from beads by using either buffer A containing 300 mM imidazole or fresh 10 mM glutathione buffer in Tris-Cl, pH 8.0. The activities of the eluted proteins were examined using HT-SELEX, and the *E. coli* clones, in which active proteins were obtained, were used for larger-scale protein expression in 100 ml of auto-induction medium to obtain enough proteins for CAP-SELEX screening (using a similar protocol but with proportionally larger lysis and wash volumes). The concentration of the individual proteins obtained in a scaled-up culture was measured using spectrophotometry at 280 nm, and the proteins were diluted to 50 ng ml$^{-1}$ when possible, supplemented with glycerol to 50% final concentration, and stored at −20 °C.

## HT-SELEX, CAP-SELEX and mixture-SELEX

HT-SELEX and CAP-SELEX were performed as described previously[9], with some improvements. The Flag tag was replaced with a His tag to make affinity purification more efficient and less costly, and the data analysis pipeline was automated (see following paragraphs). The SELEX ligands were similar to those used in a previous study[51]; they contain 40-bp random sequences and Illumina sequencing adaptor sequences on both sides, and were designed for characterizing the cooperativities of the TF pairs. The single-stranded oligonucleotides containing the ligand sequences were purchased from Eurofins Genomics (Supplementary Table 1) and the double-stranded selection ligands were

obtained by performing one-step PCR with primers binding to the adapters[52]. In CAP-SELEX, 158 prey proteins were screened against a set of 376 bait proteins arranged on 384-well plates. During the screen, some bait proteins were replaced on the 384-well plate, resulting in analysis of a total of 58,754 interactions. All tested TF–TF pairs are listed in Supplementary Table 2. For CAP-SELEX, around 1 µg dsDNA ligands was mixed with around 100 ng purified His- and SBP-tagged proteins in 50 µl Promega buffer (4% glycerol, 1 mM DTT, 500 µM EDTA and 50 mM NaCl in 10 mM Tris-Cl, pH 7.5) in individual wells of 96-well MultiScreen MSDV 0.65-µm hydrophobic filter plates (Millipore, MSDVN6510). For a given plate, each sample well contained the bait protein for the plate, and a well-specific prey protein; each plate also contained eight control wells of known interacting pairs (from ref. 9; the bait–prey pairs used were CEBPD–ATF4, CEBPD–ETV5, FOXO1–ETV5, TEAD4–CLOCK, FOXO1–GCM1, HES7–TFAP2C, TEAD4–ONECUT2 and HES7–ETV5). After incubating for 30 min at room temperature, 1.2 µl Ni-Sepharose beads (GE, now Cytiva, 17-5268-01) were added and the incubation continued for 2 h on a Timix microplate shaker with a speed of 1,300 rpm. The beads were then washed three times with a microplate washer (405 LRS, BioTek), and the protein–DNA complexes were eluted from the beads using 60 µl Promega buffer with 300 mM imidazole. The eluted complexes were transferred into Echo Qualified 384-Well Polypropylene Microplates, in which each well contained 1.25 µl His Mag Sepharose Ni Beads (Cytiva, 28967390) pre-blocked by 0.5% BSA and 0.1% Tween 20. After incubation for 30 min on a VWR DMS 2500 Microplate shaker with a speed of 1,900 rpm, the beads were washed seven times with a Tecan Hydrospeed 384-well microplate washer and the bead suspension was used for PCR as previously described[9]. The selection process was repeated three times and the PCR amplicons from the third cycle as well as the initial library were sequenced using Illumina HiSeq 2000 with 50-bp single-end reads. The screen revealed 2,198 TF–TF interactions and the preference of the proteins from one TF family to interact with proteins from other TF families was calculated and visualized using circos v.0.69-8 (Fig. 1d).

Mixture-SELEX was used to test whether the composite motifs could also be recovered by simply mixing the two TFs and performing SELEX (Extended Data Fig. 2c–g). It was performed as described previously[53]. The protein expression was performed as described above, except the bacterial cells were incubated for 8 h at 37 °C until a density suitable for induction was achieved, followed by incubation for 40 h at 17 °C. Only one overnight freeze–thaw cycle was used for lysis. The lysis buffer contained 400 mM NaCl, 100 mM KCl, 10% glycerol, 0.5% Triton X-100 and 10 mM imidazole in 50 mM potassium phosphate buffer, pH 7.8; the concentration of MgSO$_4$ during incubation with Ni-Sepharose beads was 15 mM. The concentration of imidazole for protein elution from beads was 500 mM, and Bradford reagent (B6916, Sigma) was used for estimating protein concentration. For mixture-SELEX, two TF proteins (200–600 ng) were added to individual wells of a 384-well plate containing the DNA ligands (in the first cycle approximately 1.5 µg of annealed and elongated oligonucleotides; in subsequent cycles 200–500 ng of PCR product from the previous cycle) in a final volume of 20 µl. The final composition of the binding buffer was 125 mM NaCl, 1.1 mM MgCl$_2$, 0.7 mM dithiothreitol, 0.07 mM EGTA, 17.8% glycerol, 3.7 µg ml$^{-1}$ poly-dI-dC and 1.4 µM ZnSO$_4$ in 22 mM Tris-Cl, pH 7.5. The proteins were incubated with the DNA ligands at room temperature for 20 min. Then, 1.75 µl of magnetic Ni-Sepharose beads (GE Healthcare) were washed in Promega (50 mM NaCl, 1 mM MgCl$_2$ and 4% glycerol in 10 mM Tris, pH 7.5) + 0.2% BSA, stored in the same solution with the addition of 0.01% NaN$_3$ and 0.1% Tween and diluted in 25 µl of SELEX buffer (100 µM EGTA, 1 mM dithiothreitol, 5.37 µg ml$^{-1}$ poly-dI-dC and 1.9 µM ZnSO$_4$ in Promega buffer, pH 7.5) right before adding to every well with SELEX reaction. The mixtures were then further incubated for 20 min at room temperature, aspirated and washed with cycles of volumes (1 × 300 µl, 6 × 50 µl, 3 × 100 µl, 6 × 50 µl, 3 × 100 µl) low-salt wash buffer (5 mM EDTA and 1 mM dithiothreitol in 5 mM Tris-Cl,

pH 7.5). The elution of ligands was performed by adding 35 µl of elution buffer (1 mM MgCl$_2$ and 0.1% Tween in 10 mM Tris, pH 7.8) followed by incubation at 80 °C for 10 min. Eluted ligands were amplified by PCR (Phusion DNA polymerase) and used as input ligands for the next SELEX cycle. Three SELEX cycles were performed. The selected ligands from the third cycle were sequenced (Illumina HiSeq 2000) and compared with existing input library sequences (from PRJEB20112).

## Discovery of spacing and orientation preferences

To detect the spacing preferences of DNA-bound TF–TF pairs, we compared data from the HT-SELEX and CAP-SELEX using a novel algorithm based on mutual information (MI) (https://github.com/YinLabTJ/MI_CAP-SELEX). The underlying rationale is that if two TFs cooperatively bind to DNA with a preferred spacing, the 4-mers at the positions with the preferred spacing will enrich together, which results in an increase of MI between the two positions.

First, for each individual TF, we identified the local maxima 8-mers (8-mers that were more enriched than any other sequence within one Huddinge distance[54]) from cycle-3 or cycle-4 HT-SELEX data that were enriched more than fivefold compared with input (cycle 0) sequences (Supplementary Table 4). We then partitioned each 8-mer as well as its reverse-complementary sequence into a set of ten 4-mers (Extended Data Fig. 1b), which were indexed as follows: 1 to 5 in the forward orientation, followed by the reverse complement 4-mers indexed 6 to 10 starting from 6 being the reverse complement of 4-mer number 5. This resulted in the identification of one or more sets of 4-mers for each TF, representing each locally maximal 8-mer.

For each CAP-SELEX TF pair, we then analysed the spacing between each pair of 4-mer sets between the two TFs. To detect all four possible orientations between the 8-mers, we paired the 4-mers of the two 4-mer sets in two ways: (1) forward orientation, in which a 4-mer with index number $j$ in a set is paired with the 4-mer in the other set that has the same index number; and (2) inverse orientation, in which $j$ is paired with $5 + j$ when $j = $ <5 or $j - 5$ when $j > 5$. The 4-mers that were present in both sets were excluded from the analysis to exclude signal from individual TF-binding sites. The spacing preferences were then evaluated by converting the counts of the 4-mers to probabilities (the number of reads with an expected 4-mer in a specific position was divided by the total number of reads), followed by MI analysis of the 20 4-mer pairs at all CAP-SELEX read positions at which the positions of the 4-mers do not overlap (equation (1)).

$$
\begin{aligned}
&\text{MI(pos1, pos2)} \\
&= \sum P(\text{4-mer} + \text{4-mer})\log_2 \frac{P(\text{4-mer} + \text{4-mer})}{P_{\text{pos1}}(\text{4-mer})P_{\text{pos2}}(\text{4-mer})}
\end{aligned}
\tag{1}
$$

where $P(\text{4-mer} + \text{4-mer})$ is the observed probability of a 4-mer pair in position 1 and position 2; $P_{\text{pos1}}(\text{4-mer})$ and $P_{\text{pos2}}(\text{4-mer})$ are the marginal probabilities of individual 4-mers in position 1 and position 2, respectively; and MI represents the sum of mutual information of all 20 pairs of 4-mers; for each TF pair, MI was calculated separately for each possible pair of enriched 8-mers (4-mer sets) of the individual TFs.

The 4-mers were used for the analysis because shorter $k$-mers are too frequently present in both sets, and longer $k$-mers are not present frequently enough to remove sampling noise, which will contribute to MI.

The MI heat map in Extended Data Fig. 1b shows the MI for all possible two positions of the 4-mers on the CAP-SELEX ligands. To automatically identify the TF pairs that cooperatively bind to DNA with a preferred spacing, a set of 380 TF–TF pairs was manually curated to detect spacing preferences, and this set was then used as a ground truth for setting a threshold for the automatic analysis based on analysis of receiver operating characteristic (ROC) curves (Extended Data Fig. 9c). We first ranked the MI values for each 4-mer pair, and selected the top 5% (representing 28 of 561 position pairs). If there is no cooperative binding,

these would be distributed randomly, and represent multiple possible spacings and orientations. However, we found that in the interacting pairs, the top 5% MI signals were concentrated to fewer than 6 spacings and orientations. This analysis generally identified strong spacing preferences, but was not able to detect weak spacing preferences, or most cases in which the TF pair bound to composite sites, where the motif differed from the binding motifs of the individual TFs.

## Detection of composite motifs

To detect composite motifs, we developed an algorithm (https://github.com/YinLabTJ/Relative_Affinity_CAP-SELEX) to compare the relative affinities of 10-mers in CAP-SELEX versus the individual HT-SELEX to identify 10-mers that are strongly enriched in CAP-SELEX but not in the individual TF HT-SELEX.

To derive relative affinities from counts, we used the following process. The model is based on the assumption that SELEX is a thermodynamic affinity-based selection process, with the following set of coupled equilibria for the DNA molecules in the pool:

$$
T : D \underset{K_i}{\overset{K_{-i}}{\rightleftarrows}} T_{\text{free}} + D_i
$$

Here $D_i$ is the $i$-th species of molecules in the DNA pool, $T$ the TF or TF pairs, and T:$D_i$ the TF–DNA complex. If $F$ denotes the fraction of $D_i$ in the pool and $K_d(D_i)$ the dissociation constant for the $i$-th equilibrium, the relationship between post-selection frequencies $F'$ and the preselection frequencies $F$ can be shown as in ref. 55:

$$
\frac{F_i'}{F_j'} = \left( \frac{K_d^{(D_j)} + [T_{\text{free}}]}{K_d^{(D_i)} + [T_{\text{free}}]} \right) \frac{F_i}{F_j}
$$

After multiple rounds of selection, most of the TF or TF pair is assumed to bind to the optimal DNA sequence and the relative affinity $K_a$ of DNA sequence $D_i$ in terms of the frequencies in cycle $r$ and cycle 0 can be expressed as:

$$
K_a(D_i) = \frac{K_{d,\text{ref}}}{K_d(D_i)} \approx \left( \frac{F_i^r / F_{\text{ref}}^r}{F_i^0 / F_{\text{ref}}^0} \right)^{\frac{1}{r}}
$$

Approximation is based on a previous study[55]. $F_{\text{ref}}$ indicates the frequency of the most abundant sequence in the pool. We next adapted the above equation to infer a table of relative affinities $K_a(k)$ for all $k$-mers $k$ with a length $l$, based on an assumption that a single $k$-mer on the DNA ligand dominates the rate at which the DNA ligand is selected. Then the relative affinity of a specific $k$-mer $k$ can be estimated as:

$$
K_a(k) \approx \left( \frac{F_k^r / F_{\text{ref}}^r}{P_0(k) / P_0(\text{ref})} \right)^{\frac{1}{r}}
$$

$P_0(k)$ denotes the expected frequency of $k$ in the initial DNA pool and is computed using a fifth-order Markov model (Extended Data Fig. 9d).

The relative affinities for individual 10-mers were calculated in the DNA libraries enriched from cycle 3 of CAP-SELEX and from that of the two corresponding HT-SELEX experiments. The data were then plotted in an $xy$ plot, in which the $y$ axis is the relative affinity in CAP-SELEX and the $x$ axis is the higher of the relative affinities in HT-SELEX (https://github.com/YinLabTJ/Relative_Affinity_CAP-SELEX). If the relative affinity of a 10-mer was ranked at the top 50% and 1.5 times higher in the CAP-SELEX library than in the corresponding HT-SELEX library, the 10-mer with the highest relative affinity within a Hamming distance of 1 was used as an initial seed to generate a composite motif. Note that for a given pair, multiple seeds could be generated this way.

### Generation of PWMs and motif logos

Position weight matrices (PWMs) were generated using Autoseed[54]. Autoseed[54] generates motifs using a multinomial model (figure 1b in ref. 56) that describe the enrichment of small set of sequences close to a highly enriched $k$-mer or set of $k$-mers described by an IUPAC degeneracy code (figure supplement 1 in ref. 54; open source code to identify local maxima and generate motifs are available in https://github.com/jutaipal/kmercount_with_localmax; https://github.com/jutaipal/multinomial_motif_generator; motifs can also be generated with the Moder package[57] https://github.com/jttoivon/moder2). Of note, the resulting models are not optimized to describe the entire sequence population enriched in CAP-SELEX, as this would generate a mixture of the dimeric and monomeric motifs. The motifs are based on the multinomial model, using background corrected counts[52]. If a seed is used to select sequences to be included in the motifs on the basis of a simple alignment using Hamming distance (for example, a Hamming distance of 1), there will be heavy bias towards the consensus sequence, and a motif resembling the seed will be recovered even from random sequences. This is because if a consensus base is present at position $i$ of a seed $k$-mer, $3(k-1)$ sequences where another position is not consensus are included in the motif (because they are one substitution away from the seed, at a Hamming distance of 1). However, if a non-consensus base is present at $i$, sequences with other substitutions will not be allowed, because they are a Hamming distance of 2 away from the seed, owing to two substitutions, one at site $i$ and another at the other position (left side of figure 1b in ref. 56). To eliminate this bias, we use a multinomial method[52]. In brief, to estimate mononucleotide distribution at position $i$, we identify the counts of the (degenerate) consensus sequence and three (classes of) other sequences that contain base substitutions at position $i$. After background correction[52], the counts of these four (classes of) $k$-mers are used to generate the PWM at position $i$ (see right side of figure 1b in ref. 56). Because the base whose mononucleotide distribution is interrogated does not contribute to the alignment, the seed alignment bias is eliminated. Note that because the multinomial method derives the distribution of mononucleotides at each position from different sequence populations, the nucleotide counts at each position do not add up to the same value (unlike in the case in which unseeded alignment is used to identify sequences to be included in a motif).

Autoseed was used with the following modifications. The third cycle of CAP-SELEX was used for PWM generation. The number of seeds for each TF pair was determined on the basis of $k$-mer local maxima[54], and the motifs were classified as primary, secondary, tertiary and so on, on the basis of the number of matches to the seed. Autoseed-generated seed was used at first, and was refined on the basis of the PWM generated as described previously[56] with minor modifications. The length of seed was defined by extending to flanking positions when the ratio between the most and the least frequent bases at the flanking position was greater than 2. Where indicated (Supplementary Table 3), to increase the number of reads included in the model, the seed was made more redundant to accommodate more sequences at positions at which the frequency of the most common base was less than 0.5. At these positions, N was used, except where the ratio between the second and the third most frequent bases was greater than 2, in which case the IUPAC symbol for the two or three most frequent bases was used. Furthermore, if the length of the seed was greater than 10 bp, where indicated, a multinomial model (multinomial = 2) allowing a single mismatch at any position (in addition to the position whose mononucleotide distribution was measured; see figure 1b of ref. 56) was used. As described previously[56], seed sequences were further manually curated to prevent the mixing of two distinct binding modes, and to distinguish between monomer and dimer models. The final seed and PWM were checked for internal consistency. Count motifs are provided in Supplementary Table 3. All motifs are also available through the CIS-BP database (https://cisbp.ccbr.utoronto.ca/). A subset of motifs that are distinct from each other is also available in HOCOMOCO (https://hocomoco11.autosome.org/).

### SELEX motif collection

Because the automated composite and spacing pipelines also discover some motifs of the other type, we classified each motif as either composite (motifs overlap or specificity changes) or spacing (TF–TF motifs that contain the two individual motifs but exhibit a specific spacing and/or orientation preference) as follows: all TF–TF pair motif and the individual motif logos were inspected separately by four experts (J.T., Y.Y., Y.C. and I.S.), and classified to the composite, spacing or unclear class. Discordant calls were then resolved at a meeting. The collection of new motifs contains 1,131 composite motifs, 205 spacing motifs and 12 HT-SELEX motifs. The motif collection was augmented with 2,585 previously published SELEX motifs for human and mouse TFs. A set of 133 mouse and 687 human HT-SELEX motifs was obtained from a previous study[56], excluding low count and replicate motifs. A set of 31 human HT-SELEX and 562 CAP-SELEX motifs were obtained from a previous study[9] by excluding technical replicates and ChIP-exo motifs. An E2F8 motif was obtained from another study[53], and ten human motifs were obtained from another study[54]. A set of 864 HT-SELEX and 297 Methyl-SELEX motifs were obtained from a previous study[51]. The set of Methyl-SELEX motifs included only the MethylPlus, Multiple effects and Inconclusive motif categories. In total, the collection contains 3,933 motifs. The protein families of TFs were extracted from ref. 3, from the original publications or from UniProt. The position count matrices and associated metadata of the 1,348 new motifs are provided in Supplementary Table 3.

### Dominating set analysis

To identify a smaller set of distinct motifs, commonly referred to as representative motifs, we performed a network analysis called the minimum dominating set in a similar manner to what was described previously[9,51,56]. This analysis identifies a minimum number of motifs in such a way that for every motif in the full set, there will be a representative motif that is similar to it. First, the similarities were computed between all motif pairs using SSTAT (Motif statistic software suite v.1.1)[58]. A network of motifs was constructed so that two motifs (nodes) were connected by an edge if the SSTAT Ssum similarity score was greater than $1.5 \times 10^{-5}$. Then, the minimum dominating set of the network was found so that every node not belonging to the dominating set was connected to at least one node in the dominating set; the minimum dominating set is the smallest of such sets. The problem is solved as an integer linear programming problem. The SSTAT Ssum similarity score was obtained with the following options: 50% GC content, pseudocount regularization and 0.01 type I threshold. The dominating set analysis of the collection of 3,933 motifs resulted in 1,232 representative motifs with distinct sequence specificities. Of the new motifs, 347 composite motifs and 112 spacing motifs were representatives (Supplementary Table 3). The dominating set analysis was also performed for the new 1,131 composite motifs alone, resulting in 391 distinct composite motifs. The motif similarity of zinc finger TFs was calculated and visualized using motifStack v.1.38.0 (https://github.com/jianhong/motifStack).

### Motif matching and motif enrichment analysis

The 3,933 SELEX motifs were matched to the repeat-masked human genome GRCh38 using MOODS[59,60] with default settings, except that the binding affinity score threshold was set so that approximately 300,000 matches were recovered for each motif. This was done by first setting a low threshold (2, natural logarithm), and then selecting a score threshold that resulted in the recovery of at least 300,000 matches. For some motifs, fewer than 300,000 matches had a score of 2 or more.

For analysing motif match enrichment at cell-type-specific candidate *cis*-regulatory elements (cCREs), the cCREs of 111 human adult cell types

were derived from the *cis*-element atlas (CATlas[44]). The cCREs were identified as non-coding open chromatin regions from single-cell ATAC-seq data on 30 adult tissues. The authors of the CATlas study[44] identified around 400,000 cell-type specific cCREs at which the ATAC-seq signal showed cell-type-restricted patterns, and divided them into distinct cCRE sets specific to 15 cell-type groups.

To investigate whether the motif matches are enriched or depleted at the cell-type-specific cCREs, we applied Fisher's exact test (one-sided) similarly to the previous study[44]. We computed the $P$ value to evaluate the deviation from the null hypothesis that the presence of a motif match at a cCRE is independent of whether the cCRE belongs to a set of cell-type-specific cCREs.

The number of cell-type-specific cCREs that have a motif match is a random variable $X$ and follows a hypergeometric distribution

$$P(X = k) = \frac{\binom{m}{k}\binom{N-m}{n-k}}{\binom{N}{n}}$$

where $k$ is the observed value of $X$, $m$ is the number of all cCREs with a motif match, $n$ is the number of cell-type-specific cCREs and $N$ is the number of all cCREs. The motif match enrichment at particular cell-type-specific cCREs was determined as relative enrichment compared with the frequency of matches of the same motif at all cell-type-specific cCREs, defined as relative frequency fold change $\log_2\left(\frac{k/n}{m/N}\right)$. The $P$ value for the enrichment or depletion was computed as follows: if the $\log_2$-transformed fold change was larger (enrichment) or smaller (depletion) than 0, the $P$ value was computed as the upper $P(X \geq k)$ or lower $P(X \leq k)$ tail cumulative density of the hypergeometric distribution, respectively. A representative composite or spacing motif was marked as enriched in a cell-type group if for any cell type in the group, the $P$ value was less than 0.01 and the $\log_2$-transformed fold change was greater than 0.75. Using the false discovery rate (FDR; Benjamini–Hochberg method) with a $q$-value cut-off of 0.01 (calculated either across motifs separately for each cell type or across all motifs in all cell types together) resulted in the same representative composite and spacing motifs enriched in at least one cCRE set, so for clarity we chose to report the uncorrected $P$ values. For a visualization of the enrichment and depletion of a set of representative bHLH–homeodomain spacing motif matches at cell-type-specific cCREs (Fig. 5c), only cell types for which the $-\log_{10} P$ value of the enrichment or depletion was equal to or more than 50 were selected. The $-\log_{10} P$ values were visualized as a heat map created with the ComplexHeatmap Bioconductor package (v.2.16.0, using a custom colour palette shown in the scale)[61,62]. In the heat map, red denotes the $-\log_{10} P$ value for the enrichment and blue denotes the $\log_{10} P$ value for the depletion. The statistical analysis and visualization were conducted in R.

For analysing motif enrichment in ChIP–seq peaks, we downloaded all TF ChIP–seq bed files for all cell lines from the ENCODE database (https://www.encodeproject.org/, ref. 17, downloaded on 6 October 2024) that were labelled 'Default analysis', used GRCh38 assembly and did not contain any match to the word 'treat' in the description. Experiments were excluded from the analysis if the data contained fewer than 1,000 peaks, or if the $P$ value of the 1,000th peak sorted by 'signalValue' was more than 0.01. We then identified regions of overlap for all interacting TF–TF pairs described in this study using bedtools v.2.30.0 (ref. 63), and applied the same peak quality filters to the sets of overlapping peaks. Bed files were then converted to fasta files using bedtools v.2.30.0 (ref. 63). The matching of PFM models to the fasta files was then performed by MOODS[60], using a default $P$-value cut-off of 0.0001. The final enrichment was calculated as the sum of MOODS scores divided by the length of the regions in the respective bed files. For statistical testing, the binomtest one-sided function from SciPy library v.1.10.0 was used. For a list of the bed files and motifs used in the

analysis, see Supplementary Table 2. For the analysis shown in Fig. 2c, ChIP–seq bed files for FOXK2 (experiment series: ENCSR786QMI, K562 cell) and ELF1 (experiment series: ENCSR918LYT, K562 cell) were downloaded from the ENCODE database and the common peaks in the two replicates of the same TF ChIP–seq experiment were used for downstream analysis as described above. The enrichment score was calculated as the occurrence frequency of the motif in ChIP–seq peaks.

## Logistic regression analysis

To study whether the adult human cell-type-specific cCRE elements could be predicted by the linear combinations of the effects of TF motif matches, we built a logistic regression classifier with lasso regularization. The logistic regression model predicts whether a cCRE $i$ is specific to a certain cell type (class 1) or not (0). The model is trained separately for all 111 cell types, the class 1 corresponding to cCREs of a specific cell type and class 0 corresponding to cCREs specific to all other cell types. The class of a cCRE $i$, $i \in 1, ..., N$ is a random variable $Y_i$, which is modelled as generalized Bernoulli regression with the logit link function, also known as the logistic regression

$$P(Y_i = y_i | x_i) = \text{Bernoulli}\left(y_i, \ \pi_i = \frac{\exp(x_i^T w)}{\exp(x_i^T w) + 1}\right)$$

where the probability of class 1 for sample $i$, $\pi_i$ depends on $d$ features $x_i = [x_{i1}, ..., x_{id}]^T$. The features $x_i$ correspond to the motif matches of $d - 1$ TFs or TF pairs and the intercept term. The coefficients $w$ describe the feature effects on the class 1 probability $\pi$. As features $x_i$, we adopted either binary presence (1) and absence (0) of the motif match, or the binding affinities of the motif matches (0 if no binding; if multiple matches per cCRE, consider the maximum affinity score). We only considered the matches of the representative motifs ($d = 1,232$).

We applied a lasso-regularized logistic regression by minimizing the penalized negative log-likelihood

$$-\sum_{i=1}^{N} \log P(Y_i = y_i | x_i) + \lambda ||w||_1,$$

where $||w||_1 = \sum_{j=1}^{d} |w_j|$ is the L1 norm of the vector $w$ and $\lambda$ is the regularization parameter. To evaluate the model performance and to optimize the value for the regulation parameter $\lambda$, a nested cross-validation (CV) was performed. The outer fivefold CV was used to obtain area under receiver operating characteristics curve (AUC) values, and the inner tenfold CV was used to optimize the regularization parameter $\lambda$. The lasso-regularized logistic regression was trained using the glmnet R package[61,62].

The means and standard deviations of the AUC values obtained from the fivefold CV for different cell-type-specific models within the cell-type groups are shown in Supplementary Table 4 for both the model with binary features and the model with affinity score features. The predictive performance varied between cell types, with the mean AUC values varying between 0.55 and 0.75. In general, the models with the affinity score features reached a better predictive performance and the further results were produced by this model. The final model was trained with all data, the regularization parameter again optimized with tenfold CV. The cell-type-specific regression coefficients were centred (subtracted by the mean) and scaled (divided by the standard deviation). For two pairs of TF protein families, ETS–RUNT and Forkhead–ETS, the representative motifs were selected. For these motifs, the regression coefficients are visualized as a heat map for those cell types for which the absolute scaled regression coefficient was at least 2 for at least one motif.

## Evolutionary conservation analysis

To identify the possible conservation of TF–TF–DNA interactions described by the motifs, we followed a procedure similar to one described previously[9,51]. In brief, to test whether the composite and

spacing motifs explain the patterns of evolutionary conservation observed in the non-coding genome, we compared the conservation at the motif matches of each TF–TF pair with the conservation at the motif matches of artificial control motifs.

To obtain a set of artificial heterodimeric control motifs that still contain the specificities of the individual TFs, each heterodimeric motif was divided in half using all possible split points so that the lengths of the divisions were at least one-third of the length of the whole motif (rounded down). Then, for each split, three control motifs were formed by concatenating the 'right' and 'left' halves, the 'left' and 'right' reverse complement halves and the 'left' reverse complement and 'right' halves. The artificial control motif generation was also performed for the monomeric motifs in the representative set. To prevent generating artificial control motifs that are similar to the corresponding true motif, the 10-mer gapped similarities (gapped $k$-mer similarity: https://github.com/jutaipal/motifsimilarity) were computed between the true motif and the artificial control motifs. Control motifs with a similarity larger than 0.1 with the original motif were excluded from the set of control motifs.

We compared the evolutionary conservation of genomic sites recognized by each heterodimeric motif with sites recognized by artificial control motifs. To identify the evolutionary conserved motif matches, we used the conservation scores and constrained nucleotides in the human genome defined by the Zoonomia Consortium[38,64,65]. By comparing the genome sequences of 241 placental mammal species, Zoonomia defined the slower than expected (conservation) and faster than expected (acceleration) evolution for every base in the human genome as positive and negative $P$ values, respectively, also known as phyloP scores[66]. In addition, Zoonomia identified the most significantly (FDR < 0.05) conserved nucleotides in the human genome (phyloP score ≥ 2.27, around 101 Mb or 3.26% of the genome) and called them the human constrained nucleotides.

For a given TF–TF–DNA interaction, we considered the matches of the true motif and the corresponding artificial control motifs that overlap the human constrained nucleotides. The matches overlapping protein-coding exons from GENCODE v.44 or simple tandem repeats from UCSC were removed from the analysis. For the matches of the true motif that overlap the human constrained nucleotides, the average phyloP scores were obtained and a two-component Gaussian mixture model (GMM) was fitted to the scores to separate constrained matches and non-constrained matches[67]. Ten GMMs were randomly initialized and the best model according to the Bayesian information criterion was chosen as the final model. The GMM fitting results in a threshold value for the motif match being constrained; if the average phyloP score of a motif is higher than the threshold value, the probability that the motif belongs to the constrained set is higher than the probability of not belonging to the constrained set. This analysis results in a different threshold for different motifs (Supplementary Table 4).

The matches of both the true motif and the artificial motifs were merged into one list and 10,000 non-overlapping matches with the highest affinity score were selected. The average phyloP scores were obtained at these motif matches, and the matches with an average phyloP score exceeding the motif-specific threshold were defined as conserved.

To study which TF–TF pairs the motif matches are enriched for in human constrained elements, Fisher's exact test was applied to the contingency table in Extended Data Fig. 8a to obtain the $P$ value for the statistical significance. We computed the number of conserved sites ($k$) and non-conserved sites ($n - k$) for the original motif, as well as the number of conserved sites ($m - k$) and non-conserved sites ($N + k - n - m$) for the corresponding artificial motifs. The $P$ values were computed only for the enrichment, using the upper tail cumulative density of hypergeometric distribution. The $P$ values were adjusted to control the FWER (the probability of making at least one false positive in the set), using Holm's method in R. The numbers of evolutionarily

conserved matches for true motifs and for artificial control motifs were compared as described previously[9], and the enrichment was quantified as the conditional frequency fold change $\frac{k/n}{(m-k)/(N-n)}$. The fold enrichment values were plotted as the function of the number of conserved motif matches among 10,000 highest-scoring matches (Extended Data Fig. 8). The number of the new conserved composite motifs was defined with an FWER threshold of 0.05. Statistics are provided in Supplementary Table 4.

To study whether the conservation of one half-site of the composite motif increased the probability of the conservation of the other half-site, we computed the correlation of phyloP scores across each base position within the matches of the composite motif that were included in the set of 10,000 true motif and the artificial motif matches (see above). To define an empirical Gaussian null distribution for the phyloP correlations between different base positions, the phyloP correlations were computed also within the matches of the corresponding artificial motifs. The $P$ value for correlations was obtained by calculating the probability, under the null distribution, of observing a correlation at least as high as that which was observed. The $P$ values for each motif were subjected to FDR correction in R (Benjamini–Hochberg). The correlations and the corresponding $q$-values can be represented as a triangular matrix. The triangular matrix can be divided into two triangles corresponding to the half-sites of the motif and a square corresponding to the correlations between the half-sites. The square was further divided into an upper triangle and a lower triangle. The lowest $q$-value of the lower triangle was chosen as the statistical significance of the correlation between the half-sites of the composite motif. The conservation of the halves of the composite motif was significantly correlated, with FDR < 0.05.

For the evolutionary conservation analysis of the orientation and spacing preferences of TF–TF pairs, we analysed the 1,291 TF–TF pairs that showed orientation and spacing preferences (Supplementary Table 2) for which the monomeric motifs of both individual TFs are found in the full collection of 3,933 SELEX motifs. Because the IRF8, MAFF, NFATC1, SOX11, TGIF2LX and ONECUT1 motifs are homodimers, their motifs were replaced by artificial half-site motifs. For each TF–TF pair, we chose the first 6-mer count table from Supplementary Table 7, and identified the most preferred orientation and spacing. Then, for each individual TF in the pairs, we chose a monomeric motif. If there were many motifs for the same TF, we first excluded any Methyl-SELEX motifs, and chose the shortest motif. If many motifs were identical in length, we chose the PWM that best matched the representative 6-mer (the 6-mer score was calculated for all positions and orientations within the corresponding PWM, and a maximum was taken, recording the position and orientation of the match so that the 6-mers could be aligned to the PWM matches later). For palindromic 6-mers that had identical scores in both forward and reverse orientations, we recorded the forward orientation. Next, for each monomeric TF motif, and for each set of corresponding artificial control motifs (see above), we identified approximately 300,000 genome-wide matches of the individual TFs. The matches overlapping protein-coding exons (from GENCODE v.44) or simple tandem repeats from UCSC were again removed from the analysis. The monomeric motifs matches that had a gap of fewer than 30 bp in the genome in different orientations and spacings were then identified, and compared with the spacings in the 6-mer tables. The analysis was performed for both the true motif pairs and for motif pairs in which one member is the true motif and the other is one member of the set of the artificial control motifs for that TF. We then extracted the average phyloP scores at the paired monomeric motif matches, and computed for each orientation and spacing the number of match pairs in which both matches are conserved, using the motif-specific thresholds for the average phyloP score (Supplementary Table 4). The number of conserved motif match pairs was then compared with the empirical null distribution derived using the control motif sets. The $P$ value for the conservation for each orientation and spacing is computed as the

probability of observing the number of conserved true motif match pairs or a more extreme value given the null distribution. The $P$ values were subjected to FDR correction using the Benjamini–Hochberg method, with an FDR threshold of 0.01. To determine whether the number of conserved orientations and spacings is significantly higher than what is expected by random, a Fisher's exact test (one-sided) was used.

### Relative affinity of individual motifs to composite motifs

To determine how well individual TFs could bind to composite motifs in the absence of their partner, we calculated a score differential. For this, a consensus sequence was derived from both composite motifs and the corresponding individual motifs. Motif matching for individual motifs against the corresponding collected consensus sequence was performed using MOODS with the $P$ value set to 0.0001. The maximum score of the individual motifs against the composite motif consensus sequence and the maximum score of the individual motif against its own consensus sequence were compared and are plotted in Extended Data Fig. 3g.

### Difference between the composite motif core and the individual TF motif flanks

To determine how often the part of the composite motif that overlaps with both individual motifs (composite overlap) differs from the individual TF motifs, we aligned the motifs of individual TFs against the composite motifs and quantified the difference between the composite overlap regions using two methods: JSD and the Jaccard index of the highest-affinity $k$-mers. First, for each individual TF in the composite pairs, we chose a monomeric motif from the collection of 3,933 SELEX motifs (excluding Methyl-SELEX motifs) in which the dimeric motifs for IRF6, MAFF, NFATC1, ONECUT1, SOX11 and TGIF2LX were replaced by the six artificially created monomer motifs. For TFs with multiple monomers in the collection, the shortest motif was chosen; for equal-width motifs, the one with the highest information content was selected. Of the 1,131 composite motifs, 15 do not have monomeric motifs for either one or both individual TFs; these were consequently not analysed. We then aligned the motifs of individual TFs against the composite motifs using TOMTOM[68], with a significance threshold of one. Only composites for which the flanks of the aligned monomers overlapped were selected for further analysis. In addition, cases in which the overlapping region did not correspond to the monomer flanks were removed from the analysis, resulting in a set of 977 TF1, TF2, composite motif triplets that were analysed further. For each triplet, the sections of the corresponding PWMs that constituted the overlapping region were extracted (composite overlap as $PWM_{co}$; TF1 and TF2 overlaps as $PWM_{TF1o}$ and $PWM_{TF2o}$, respectively).

To determine the distance between the overlapping PWMs, we used the JSD metric, which is symmetric and bounded, and enables measurement of the difference between PWMs as bits of information. For JSD analysis, for each base position $i$ in the overlap, we quantified how distinguishable the base distributions are between the overlapping regions of the composite motif and the individual motifs by computing the total JSD as the sum of JSDs of all PWM positions. A high JSD indicates different base distributions between the overlapping parts of the PWMs. Composite $PWM_{co}$s having a total JSD of higher than 0.5 bits to either of $PWM_{TF1o}$ or $PWM_{TF2o}$ were considered to be distinct.

To determine whether the DNA sequences matching the overlapping regions of the motifs are different, we used the Jaccard index of $k$-mers. For this analysis, we determined the difference between the composite motif core and individual TF motif flanks on the basis of whether they have high affinity to the same $k$-mers. The 975 triplets in which the overlap was shorter than 13 bp were analysed. All $k$-mers of the length of the overlap were scored against $PWM_{co}$, $PWM_{TF1o}$ and $PWM_{TF2o}$, and $k$-mers with at least 90% of the maximum score were selected. Subsequently, we computed the Jaccard index between the composite $k$-mers and TF1 and TF2 $k$-mers. Composite cores with a Jaccard index lower than 0.5 against both of the flanks were defined

as different from the flanks. The Jaccard indexes and the JSDs for the composite core–flank pairs are shown in Extended Data Fig. 3h,i with examples of individual TF motifs aligned against composites with the overlapping composite cores and flanks highlighted. Extended Data Fig. 3h,i also illustrates the correspondence between the JSD and the Jaccard index for the selected composite motifs.

### Protein expression, purification and crystallization

Expression and purification of the DBD fragment of human HOXB13 (residues 209–283), MEIS1 (residues 279–333), TEAD4 (residues 40–112), FOXK1 (residues 302–400), ELF1 (residues 186–305) and BARHL2 (residues 232–292) were performed as described previously[51,69,70]. The DNA fragments used in crystallization were obtained as single-strand oligos (Eurofins), and annealed in 20 mM HEPES (pH 7.5) containing 150 mM NaCl and 0.5 mM Tris(2-carboxyethyl)phosphine (TCEP) and 10% glycerol. For each complex, the purified and concentrated protein was first mixed with a solution of annealed DNA duplex at a molar ratio of 1:1.2–1.5 and after one hour on ice was subjected to the crystallization trials.

The crystallization conditions for HOXB13 homodimer and HOXB13–MEIS1 heterodimer complexes were optimized using an in-house-developed crystal screening kit combining different PEGs with different additives. For the first screening of other complexes, we used the screens Nuc-Pro from Jena (Jena Bioscience), ProPlex from Molecular Dimensions (Anatrace Products) and PEGRX and Natrix from Hampton Research (Hampton Research). All crystals were obtained in sitting drops by using the vapour diffusion technique. The HOXB13 homodimer–DNA complex was crystallized from the solution containing 19.6% PEG(3350)MME, 0.15 M KCl and 8% PEG(400) in 50 mM Tris-Cl, pH 8.0. The HOXB13–MEIS1–DNA complex was also crystallized from the solution containing 17.6% PEG(5000)MME, 0.06 M MgCl$_2$, 0.15 M KCl and 4% pentanol in 50 mM Tris-HCl, pH 8.0. The HOXB13–TEAD4–DNA complex was crystallized from the solution containing 15% PEG(4000), 0.5 M (NH$_4$)$_2$SO$_4$ and 4% PEG(550)MME in 50 mM MOPS, pH 7.24. The FOXK1–ELF1–DNA complex was crystallized from the solution containing 14.4% PEG(3350)MME, 0.1 M MgCl$_2$ and 8% PEG(200) in 50 mM Na-HEPES, pH 7.2. Crystals of BARHL2 complexes with two different DNAs were obtained in the same conditions from the reservoir solution containing 50 mM sodium cacodylate buffer (pH 6.5), 7–10.5% PEG(4000) and 5% butanol (longer DNA) or PEG(200) (shorter DNA). All datasets were collected at the ESRF from a single crystal on beamline ID23-1, at 100 K using the reservoir solution as cryo-protectant. Data were integrated with the program XDS[71] and scaled with SCALA[72,73]. Data collection statistics are presented in Supplementary Table 5.

### Structure determination and refinement

All structures were solved by molecular replacement using the program Phaser[74] as implemented in PHENIX[75] and CCP4 (ref. 73), with the structure of related proteins obtained from the PDB or from the structure collection of our laboratory: structure of HOXB13–DNA(CAA) (PDB ID 5EEA, chain A) as a search model for HOXB13; structure of MEIS1 (5BNG, chain A) as a model for MEIS1; structure of TEAD4 (5GZB, chain A) as a model for TEAD4; structure of FOXK1A (2C6Y, chain A) as a model for FOXK; and structure of ELF3 (3JTG, chain A) as a model for ELF1. For the two BARHL2 dimers, we used the BARHL2 monomer structure[76]. After the positioning of protein, the DNA sequence was manually built to the density using Coot[77]. The rigid body refinement with REFMAC5 was followed by restrain refinement with REFMAC5, as implemented in CCP4 (ref. 73) and Phenix.refine[78]. The manual rebuilding of the model was done using Coot. The refinement statistics are presented in Supplementary Table 5. Figures showing structural representations were prepared using PyMOL (https://www.pymol.org/pymol.html).

The atomic coordinates and diffraction data for six novel structures presented in this study have been deposited to the PDB with the accession codes 8R7Z and 8R7F for homodimers of BARHL2 bound to

different DNAs; 8BZM for the FOXK1–ELF1–DNA structure; 8BYX for HOXB13 homodimer bound to DNA; 5NO6 for the TEAD4–HOXB13–DNA structure; and 5EG0 for the MEIS1–HOXB13–DNA structure.

## Computational prediction of structures

The AlphaFold multimer model[79] was used to predict the structures of cooperative TF pairs that were mediated by protein–protein interaction (MYC–MAX, CEBPD–ATF4 and FOS–JUN), facilitated by DNA (MEIS1–HOXB13 and MEIS1–DLX3) or dependent on DNA (HOXB13–TEAD4) by running the following command:

```
python3 docker/run_docker.py \
  --fasta_paths=multimer.fasta \
  --max_template_date=2020-05-14 \
  --model_preset=multimer \
  --data_dir=$DOWNLOAD_DIR \
  --output_dir=$OUTPUT_DIR
```

The predicted structures were refined and are shown in Fig. 4a. The r.m.s.d. was calculated for each structure and is shown in the figure legend.

The predictions of protein–DNA complexes were performed with RoseTTAFold2NA v.0.2 and AlphaFold 3. For details, see Supplementary Table 6.

DNA sequences extracted from PWM files of TFs showing spacing or orientational preferences were used as input for the program. Using the max_index function for vertical axis on PWM values, the DNA sequence has been reduced to the form kmer1_XN_kmer2, where kmer1 is the most probable $k$-mer for TF1, kmer2 is for TF2 and XN is spacing preference with X nucleotides between them. The final DNA used for input had the form of either flanks_kmer1_XN_kmer2_flanks_competing_flanks (preferred DNA is on the 5′ side of the top strand), or the reverse case: flanks_competing_flanks_kmer1_XN_kmer2_flanks (preferred DNA is on the 3′ side). In that sequence, flanks were GCGAG and XN was X nucleotides from the cycled GCGA sequence. The competing sequences were for several cases covering all possible orientation preferences and three or four spacing preferences:

orient_1_2: kmer2_XN_kmer1
orient_1_3: revcompl(kmer1)_XN_kmer2
orient_1_4: kmer1_XN_revcomp(kmer2),

where revcomp is a reverse complement function for kmer. Next, competing sequences were constructed for the spacing preference analysis:

spac_m2: kmer1_(X-2)N_kmer2 (this was done if X was greater than 1)
spac_m1: kmer1_(X-1)N_kmer2
spac_p1: kmer1_(X + 1)N_kmer2
spac_p2: kmer1_(X + 2)N_kmer2.

The protein sequence was taken from a previous study[51]. Altogether, there were tested 119 PWMs, giving, in sum, up 1,620 structures to analyse. After program runs, the output structures were analysed with code (https://github.com/i-l-sokolov/RosettaFoldNA_output_analysis). In brief, the distance of the closest atoms was calculated for any nucleotide and amino acid. The contact was counted if the distance was less than 0.45 nm. The prediction was considered as true if there were more contacts on the site with sequence extracted from PWM than on the competing site. Extended Data Fig. 9a,b shows the histogram results for described cases. The statistical analysis was done with the SciPy package v.1.10.1 in Python using a two-sided binomial test with a probability of success equal to 0.5. The $P$ values for every group are shown in Supplementary Table 6.

## Estimation of the number of composite motif clusters

To estimate the fraction of all composite motifs discovered in this study, we performed two analyses using different similarity metrics and estimation methods. In the first method, we identified 391 clusters of similar motifs using SSTAT Ssum similarity (threshold $1.5 \times 10^{-5}$) and then estimated the total number of undiscovered motifs (similarity clusters) by extrapolation using a square-root function.

To build a curve for the number of motif clusters discovered as a function of motif pairs tested, we subsampled the tested motif pairs 300 times. We then fitted a curve to this data using curve_fit functions from SciPy with a p(N) equation:

$$p = A \times \mathrm{sqrt}(N) + B$$

where $A$ and $B$ are constants, $p$ is the number of clusters and $N$ is the number of pairs. The approximation gives values for $A$ and $B$ of 1.93 and −51.57, respectively, with the approximation shown in Extended Data Fig. 9e,f. The substitution of $N$ with the number of all possible interactions, including self-interactions, is $1{,}639 \times 1{,}640/2$, where 1,639 is the number of transcription factors according to ref. 3. This yields an estimate for the total number of clusters of 2,186. Given that we have found 391 clusters using this threshold, this estimate suggests that we have discovered approximately 18% (391/2,186) of all possible composite motifs.

For the second method, we calculated motif similarity using gapped 10-mer similarity (motifsimilarity program), and then clustered the motifs using the SciPy 1.10.0 Python library using 'ward' linkage and $(1 - s)$ as distance between motifs. We then used multiple thresholds for determining the clusters. The threshold range from 0.98 to 1.02 with step 0.001 was studied in detail, as the number of clusters changes markedly in that region. For every subsampled curve, the approximation was done with a linear function using the 50,000th point and last point (Extended Data Fig. 9e,f). Such an approach allows us to estimate the lower bound of the discovered fraction of composite motifs, because the linear function must always be above the real number of clusters. For the final estimation, two final values of thresholds were chosen: 0.999 and 1.000 (corresponding curves are labelled with asterisks). Similarly to above, substitution of $N$ gives us a range of covered motifs between 20% and 48%.

## Reporter assays in $F_0$ mouse embryos

To test whether the identified cooperative TF pairs are responsible for unique transcription programs in distinct cell types, a set of sequences that were preferred by TF pairs, but not by the individual TFs (Supplementary Table 8), were selected. Six copies of the motifs separated by spacer sequences that do not bind to known TFs (from a previous study[80]) were then cloned into a PCR4-Shh::lacZ-H11 vector (Addgene 139098), a LacZ knock-in reporter construct[32]. To introduce the reporter to the endogenous H11 locus, each of the reconstructed vectors was injected into the pronucleus of FVB embryos together with Cas9 protein (final concentration of 20 ng μl⁻¹; IDT 1074181), sgRNA (50 ng μl⁻¹) in the injection buffer (10 mM Tris, pH 7.5; 0.1 mM EDTA). Mice (FVB and CD-1 strains) were kept in standard housing conditions (temperature 19–23 °C and humidity 40–60%) with food and water provided ad libitum on a reversed 12-h dark–light cycle. Time of gestation was identified by the presence of vaginal sperm plugs, indicating E0.5. Pregnant dams were humanely euthanized, and E11.5 embryos were carefully removed under bright-field stereoscopes in ice-cold PBS (Cytiva, SH30256.01). Both sexes of embryos were presumed to be included. Yolk sacs were collected for genotyping, and successful integration events at the H11 locus were determined by PCR using primers described previously[32,81]. This research complies with all relevant ethical regulations. All animal procedures, including those related to generating transgenic mice, were conducted in accordance with the guidelines of the National Institutes of Health (NIH) and approved by the Institutional Animal Care and Use Committee at the University of California, Irvine under protocol no. AUP-23-005.

## Reporting summary

Further information on research design is available in the Nature Portfolio Reporting Summary linked to this article.

## Data availability

All sequence data are available in the ENA under accession number PRJEB66722. Structure coordinates are available in the PDB (entries: 8R7Z, 8R7F, 8BZM, 8BYX, 5NO6 and 5EG0). The ChIP–seq data used in this study are from ENCODE, with accession numbers ENCSR786QMI, ENCSR918LYT, ENCFF517TKD, ENCFF766VUQ, ENCFF641ZFM, ENCFF992LDJ, ENCFF882AEU, ENCFF093KLR, ENCFF053UZX, ENCFF146SYU, ENCFF164MPE, ENCFF294IGP, ENCFF601WHE, ENCFF451AII, ENCFF934JFA, ENCFF637UJN, ENCFF170IZO, ENCFF172KBM, ENCFF118UKC, ENCFF827VVQ, ENCFF700TAS, ENCFF114CWH, ENCFF440FTA, ENCFF308NZT, ENCFF100KKH, ENCFF885PQR, ENCFF519GKM, ENCFF929XBL, ENCFF839JEO, ENCFF890WJZ, ENCFF788ONH, ENCFF398TIE, ENCFF712HHN, ENCFF314KET, ENCFF399YOO, ENCFF703JHN, ENCFF023EDF, ENCFF488DML, ENCFF603PUS, ENCFF164JZB, ENCFF355HRT, ENCFF605GXY, ENCFF331BSI, ENCFF132AJP, ENCFF314CDT, ENCFF919ZSN, ENCFF011QFM, ENCFF696OTJ, ENCFF358INM, ENCFF759YCY, ENCFF798IVN, ENCFF005YUC, ENCFF647OBG, ENCFF904DOZ, ENCFF634YFK, ENCFF943XDP, ENCFF438PCR, ENCFF718VHT, ENCFF489EME, ENCFF250MUC, ENCFF951BFN, ENCFF493DXA, ENCFF882ARK, ENCFF065RZP, ENCFF970QKS, ENCFF763MXW, ENCFF715WGN, ENCFF118GMS, ENCFF324ELP, ENCFF518EGY, ENCFF566PRZ, ENCFF438IYI, ENCFF592NJN, ENCFF289ZIR, ENCFF594YBN, ENCFF112JVK, ENCFF030DLI, ENCFF091NQE, ENCFF427XNV, ENCFF751VAZ, ENCFF131TYZ, ENCFF987QLI, ENCFF381ULU, ENCFF290IBP, ENCFF463DWW, ENCFF567HPV, ENCFF833ZAC, ENCFF946KCK, and ENCFF219OPP (see also Supplementary Table 2). Genome assembly GRCh38 was downloaded from UCSC (ftp://hgdownload.soe.ucsc.edu/apache/htdocs/goldenPath/hg38/bigZips/hg38.fa.masked.gz).

## Code availability

Source code for analysis and figure generation is publicly available at GitHub: https://github.com/YinLabTJ/Relative_Affinity_CAP-SELEX, https://github.com/YinLabTJ/MI_CAP-SELEX, https://github.com/MariaOsmala/TFBS, https://github.com/MariaOsmala/TFBS-evolutionary-conservation and https://github.com/i-l-sokolov/RosettaFoldNA_output_analysis.

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

**Acknowledgements** We thank K. Jones for technical assistance, A. Taipale for help with the TF binding model and A. Jolma for suggestions. This work was supported by the National Key Research and Development Program of China (2021YFC2701400), the National Natural Science Foundation of China (32070606), Shanghai Blue Cross Brain Hospital, Shanghai Tongji University Education Development Foundation and the 2019 Thousand Youth Talents Plan of China to Y.Y., and by Cancer Research UK (RG99643), the UK Research and Innovation Medical Research Council (G105296), the Biotechnology Biological Sciences Research Council (G107673), the Finnish Center of Excellence in Tumor Genomics (352814), a Swedish Cancer Society grant (232693Pj01H) and the Swedish Research Council (2023-03773_3 and D0815201) to J.T. The protein production and purification for the structural work was facilitated by H. Ampah-Korsah, E. Strandback and T. Nyman of the Protein Science Facility at the Karolinska Institutet, Stockholm. We also thank CSC–IT Center for Science, Finland, for computational resources.

**Author contributions** Y.Y. and J.T. designed the experiments. Y.Y. performed CAP-SELEX and I.S and A.D.C. performed mixture-SELEX. E.M. and A.P. performed X-ray crystallography and E.M. solved the structures. Y.Y. and X.Y. designed the constructs for microinjection, G.B. and E.Z.K. performed reporter assays and J.P.P., S.A.T. and M.O. performed single-cell analyses. Y.Y., Z.X., I.S., M.O., K.W. and J.T. wrote computer programs and performed computational analyses of the data. Y.Y., J.T., I.S. and Y.C performed the expert analysis of motifs. Y.Y., Z.X., I.S., M.O. and E.M. prepared illustrations, and Y.Y. and J.T. wrote the article with the help of Z.X., I.S. and M.O. All authors contributed to data analyses and reviewed the manuscript.

**Competing interests** S.A.T. has consulted for or been a member of scientific advisory boards at Qiagen, Sanofi, GlaxoSmithKline and ForeSite Labs. She is a consultant and equity holder for TransitionBio and Ensocell. J.T. is a consultant for Google DeepMind. The remaining authors declare no competing interests.

**Additional information**
**Correspondence and requests for materials** should be addressed to Yimeng Yin or Jussi Taipale.

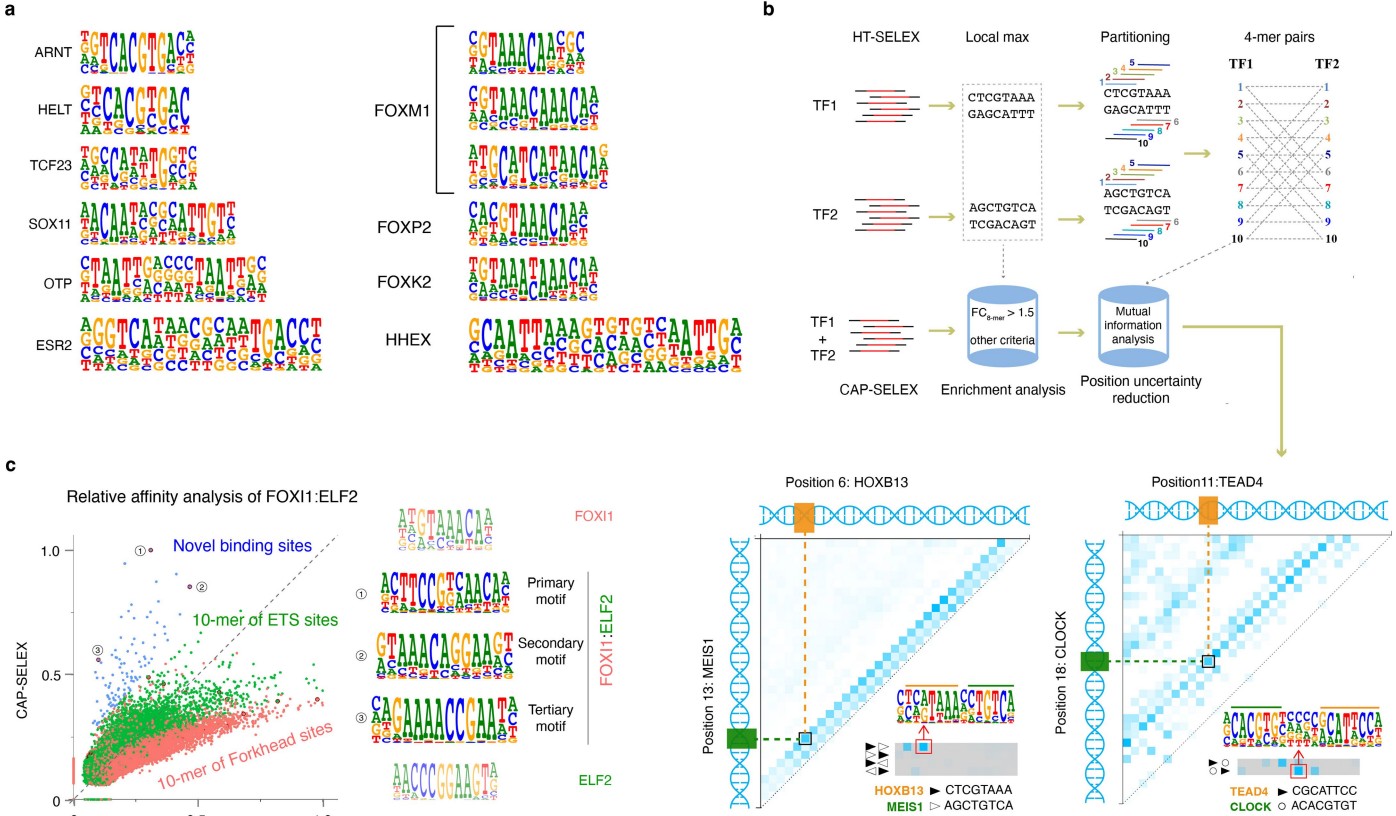

**Extended Data Fig. 1 | HT-SELEX and CAP-SELEX data analysis. a**, Logos of motifs for the 10 TFs for which no previous HT-SELEX motif existed prior to this study. For PWMs, see Supplementary Table S3. **b**, Mutual information analysis to detect TF–TF spacing preferences. Top: Mutual information analysis of characteristic $k$-mers enriched by the individual TFs in HT-SELEX to detect the spacing and orientation preferences of TF pairs tested. Bottom: Detection of spacing preferences between the TFs. Enriched spacings were detected based on analysis of spacing between characteristic k-mers enriched by the individual TFs in HT-SELEX. Inset: heat map showing enrichment of different spacings of

the indicated $k$-mers for HOXB13–MEIS1 and TEAD4–CLOCK, and sequence logo corresponding to the most enriched spacing and orientation. **c**, Detection of novel composite motifs that differ from the individual TF motifs. Left: Plot shows relative affinity of 10-mers in FOXI1–ELF2 CAP-SELEX (y axis) vs HT-SELEX of the individual TFs (x axis). Colour indicates whether the 10-mer is more similar to ETS (ELF2, green), Forkhead (FOXI1, red), or the novel composite motif (blue). Note that the CAP-SELEX specific 10-mers are located above the diagonal (dotted line). Right: FOXI1 and ELF2 motifs and the three detected FOXI1–ELF2 composite motifs.

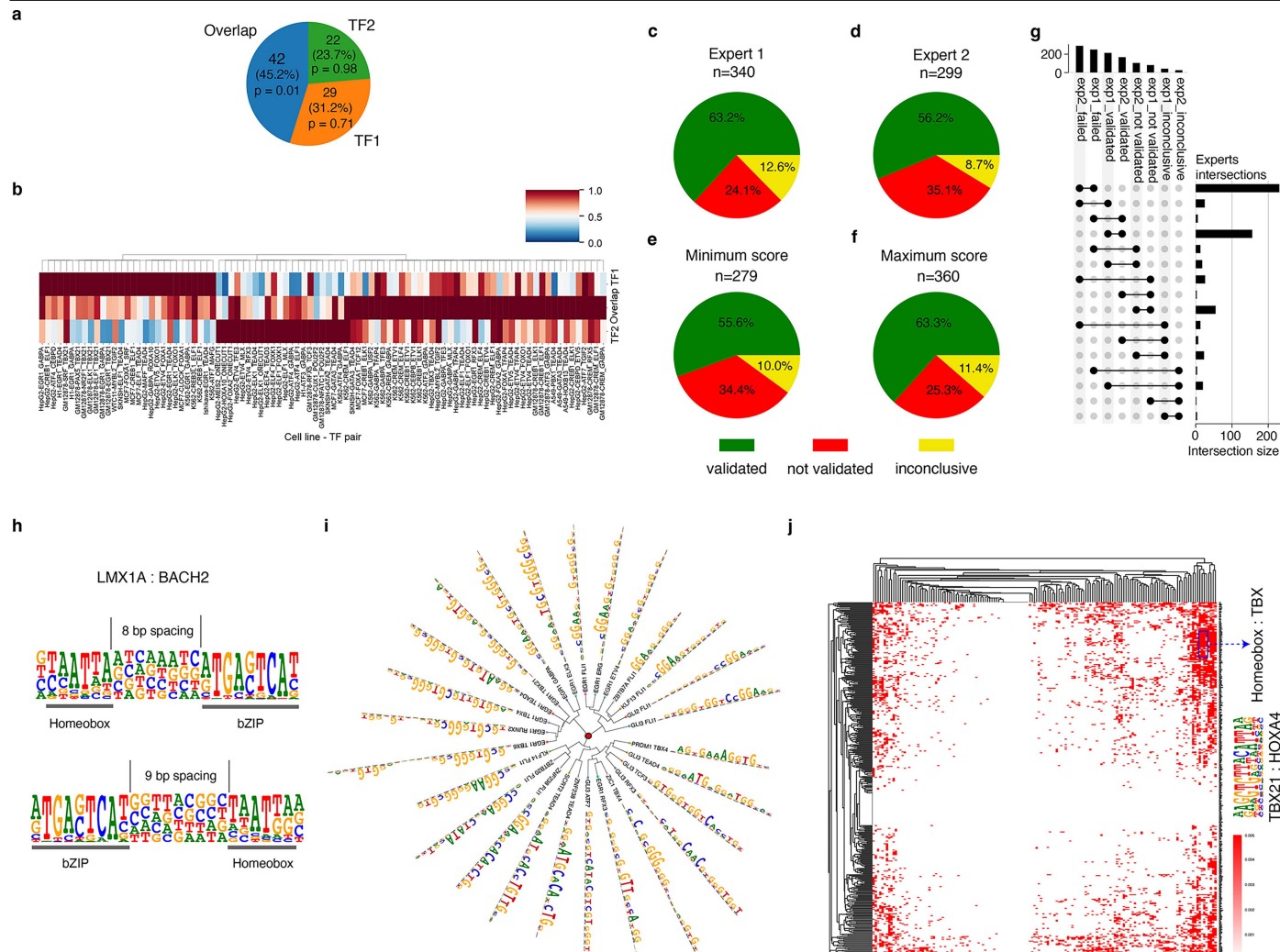

**Extended Data Fig. 2 | Validation and overview of DNA-guided TF–TF interactions. a**, Composite motif enrichment analysis in ENCODE ChIP–seq peaks. Pie chart shows number of cases with highest composite motif match enrichment in TF1, TF2 or overlapping ChIP–seq peaks. The p-values shown were derived from a binomial one-sided test. **b**, Cluster map normalized by maximum (TF1, TF2, overlap) enrichment of composite motif matches in TF1, TF2 or overlapping ChIP–seq peaks for different TF1-TF2 pairs across the tested cell lines. Clustering was performed using Hamming distance as a distance metric, and average linkage. **c**,**d**, Distributions of Expert 1 (**c**) and Expert 2 (**d**) scores presented in the form of pie charts, where "validated" (green) indicates a similar motif in mixture- and CAP-SELEX, "inconclusive" (yellow) indicates cases that partially fulfil validation criteria, and "not validated" (red) means composite motifs in CAP-SELEX and mixture-SELEX were not similar. The number of experiments scored (n) is also indicated for each pie chart.

The numbers differ as experiments where one or both of the individual motifs of the TFs were not detected were scored by each expert separately as failed, and not included in the pie-chart analyses shown. **e**, Pie chart showing distribution of minimum scores between the two experts for each TF pair. Fraction of cases where both experts agreed that mixture-SELEX validated the CAP-SELEX motifs is in green. **f**, Pie chart showing distribution of maximum scores between the two experts. **g**, An UpSet plot showing overlap between all scores between the two experts. **h**, Motif logos showing spacing preferences of LMX1A and BACH2. **i**, Composite motifs between C2H2 zinc finger TFs and other TF families. **j**, Interaction map between TFs. TFs that have at least one interaction are included. Note that many TFs do not interact well, but others have a very dense interaction landscape. The high-frequency interactions between the developmentally important T-box and homeodomain proteins are indicated by a blue box. For full data, see Supplementary Table 2.

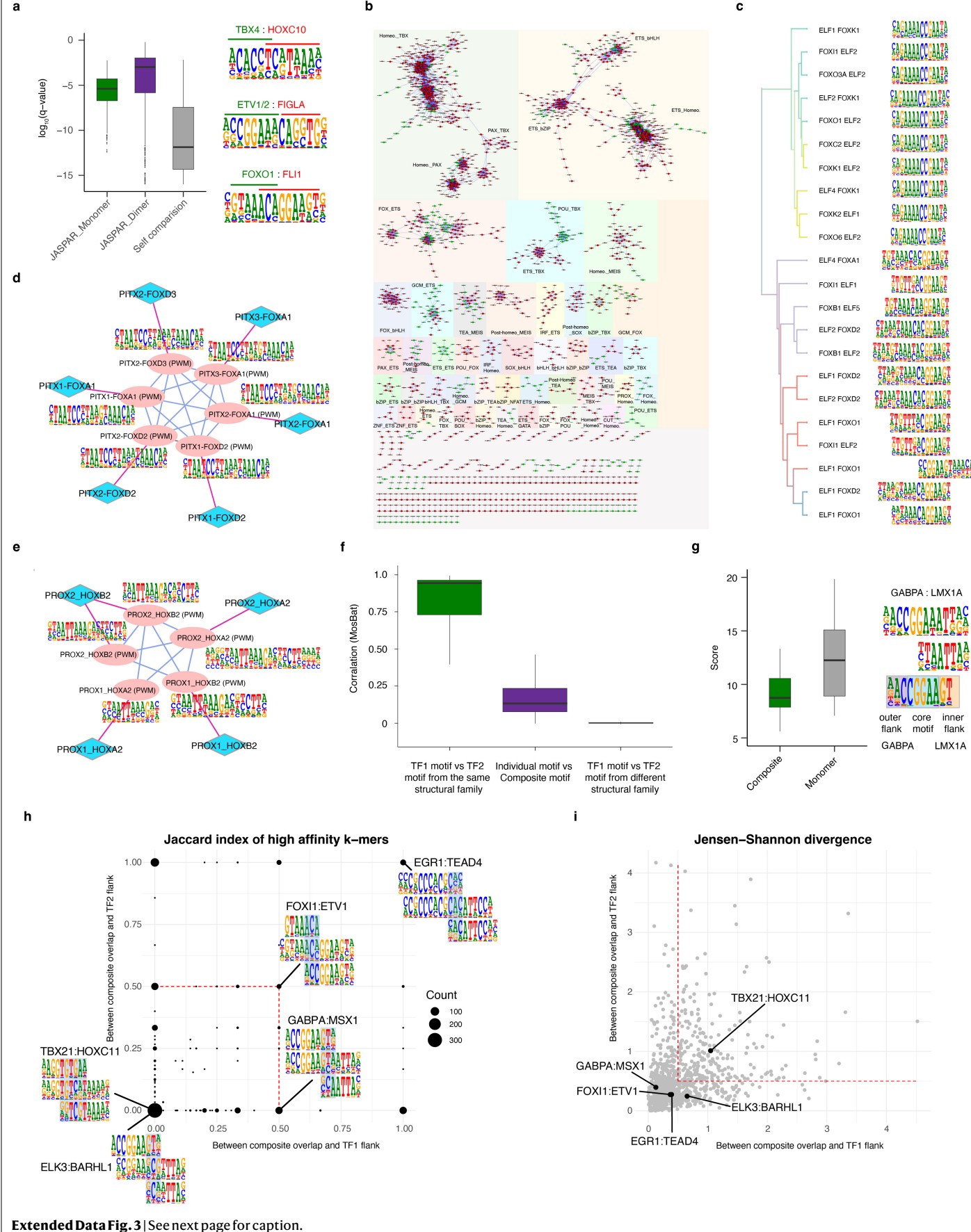

**Extended Data Fig. 3** | See next page for caption.

**Extended Data Fig. 3 | Analysis of similarity of motifs. a**, Comparison of similarity between the composite motifs and motifs in the JASPAR[82,83] database using TOMTOM[68]. Note that in the vast majority of cases, the score against JASPAR monomer (green, n = 1123) and JASPAR dimer (purple, n = 82) are much higher than against the motif itself (grey, n = 1130), indicating that most of the motifs are novel. Few examples of novel motifs are shown on the right. The centre line represents the median, the box limits are the 25th and 75th percentiles, the whiskers are the minimum and maximum values and outliers are not shown. **b**, Network analysis of the similarity of the composite motifs (red) to earlier CAP-SELEX data (green). Edges were drawn between the motifs if they had similarity p-value < $1 \times 10^{-5}$ (TOMTOM[68]). For larger groups, the TF families are also indicated. p-values were computed by TOMTOM[68], which uses linearly rescaled integral scores of each column in motifs to compute desired probability density function and offset p-values. The minimum p-value among the offset p-values is used as the motif p-value. The similarities were visualized using Cytoscape[84] (version 3.10.1). **c**, Motif logos of CAP-SELEX PWMs for FOX–ELF pairs are organized according to the similarity of the respective PWMs. **d**, Subclass of HOX and PITX proteins specifically interact with FOXAs and FOXDs. **e**, Paralogs of HOX2 (HOXA2 and HOXB2) specifically interact with PROX1 and PROX2. **f**, MosBat[85] analysis of the similarity between composite motif and the corresponding individual TF motif. Individual TF motifs within a structural family are commonly similar to each other (green box, n = 29), whereas composite motif enriched by a TF–TF pair are generally distinct from the motifs enriched by the corresponding individual TFs (purple box, n = 1069). Motifs enriched by TFs from different structural families (these are almost always distinct) are shown as a control (grey box, n = 106). The centre line represents the median, the box limits are the 25th and 75th percentiles, the whiskers are the minimum and maximum values and outliers are not shown. **g**, Most TFs bind their composite motifs weakly in the absence of the specific partner. Scores of individual motifs (n = 162) against consensus sequences from composite motifs that they bind to with a partner (green) are much lower than the corresponding scores against the individual motif consensus sequences (grey, n = 162). Low affinity is often caused by alteration of sequence preference flanking the core motif that is located towards the other TF (inner flank, yellow highlight). The centre line represents the median, the box limits are the 25th and 75th percentiles, the whiskers are the minimum and maximum values and outliers are not shown. **h**, The Jaccard index analysis of change of binding specificity. Plot shows difference in specificity of 975 TF1-TF2 composite motifs to corresponding TF1 (x axis) and TF2 (y axis) motifs in the region where the TF1 and TF2 motifs overlap, calculated using Jaccard index of high-affinity k-mers. Examples shown indicate changes in motifs at the overlapping regions (blue shading) for EGR1–TEAD4, FOXI1–ETV1, GABPA–MSX1, ELK3–BARHL1 and TBX21–HOXC11. Of the 975 tested composite motifs, 529 (54.2%) had a Jaccard index of less than 0.5 (cut-off indicated by red dashed lines) against both of the respective individual TFs. **i**, JSD analysis of change of binding specificity. JSD scores were computed for overlapping regions of 977 composite motifs. High JSD indicates different base distributions between the composite motif and TF1 (x axis) or TF2 (y axis). Of the 977 tested composite motifs, 236 (24.2%) had a total JSD higher than 0.5 (red dashed lines) against both of the respective individual TFs flanks and were defined different compared to the flanks.

**a**

Common interactions

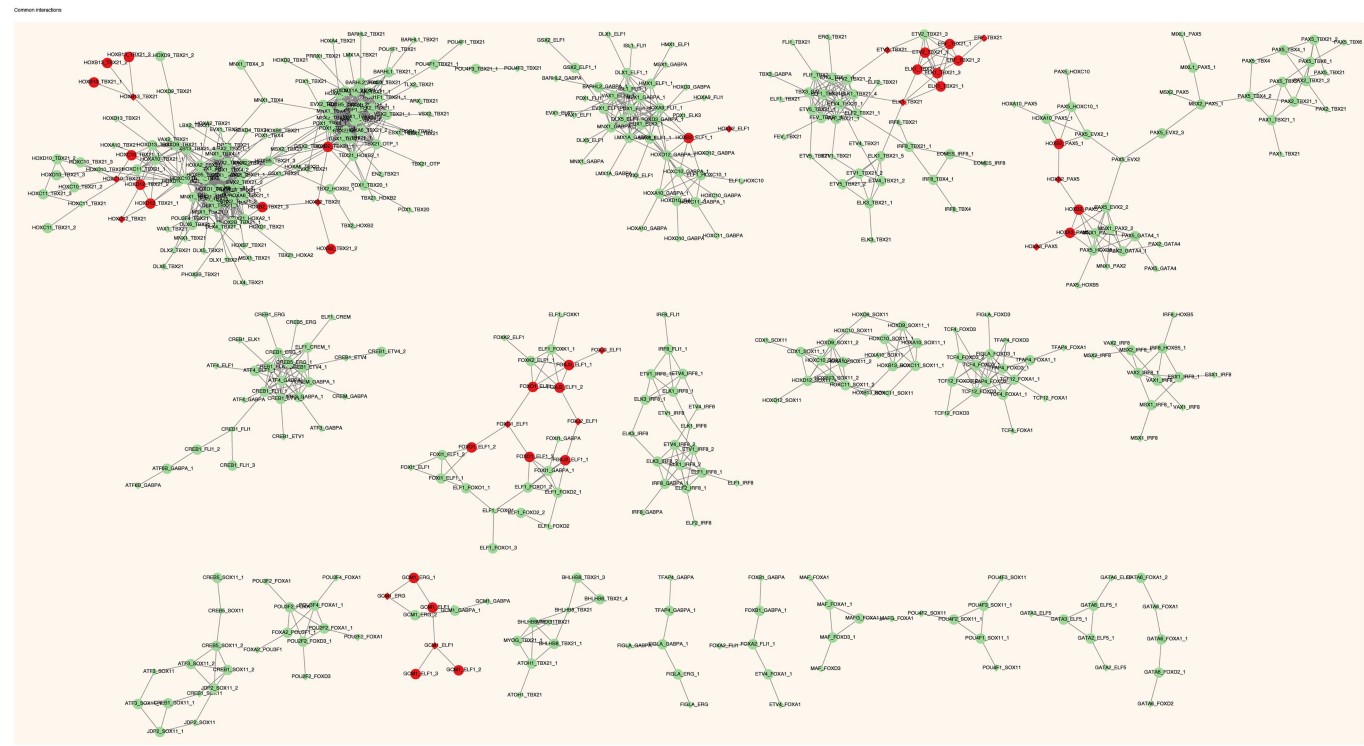

**b**

Specific interactions

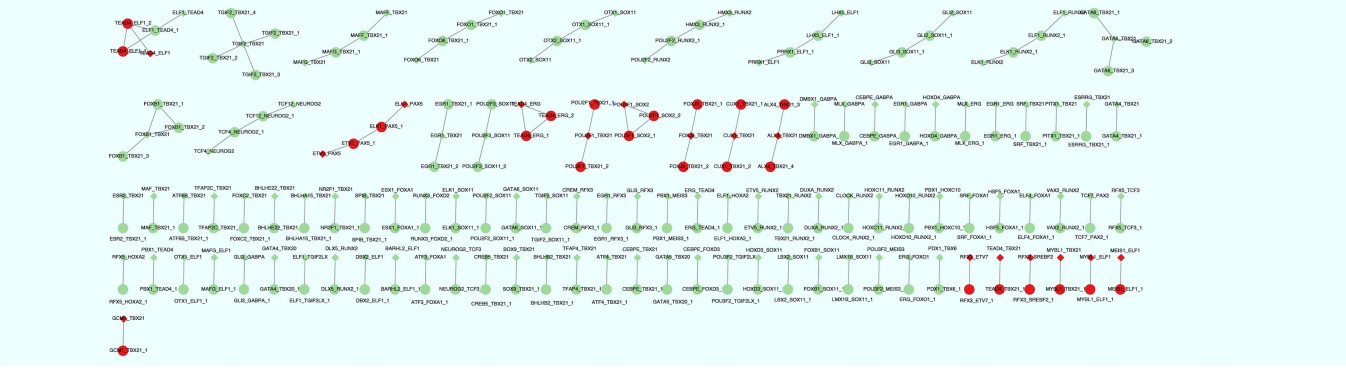

**Extended Data Fig. 4 | Interaction landscape of pioneer factors.**
**a**,**b**, Promiscuous (**a**) and specific (**b**) interactions exist between pioneer factors[24,86–90] and other TFs. The similarities were visualized using Cytoscape[84] (version 3.10.1). Composite motifs (Circular nodes) and its corresponding TF pairs (diamond-shaped nodes) were connected by edges, and edges between the motifs were drawn if they had similarity p value < 1 × 10$^{-5}$ (TOMTOM[68]). See Supplementary Data 1 for details.

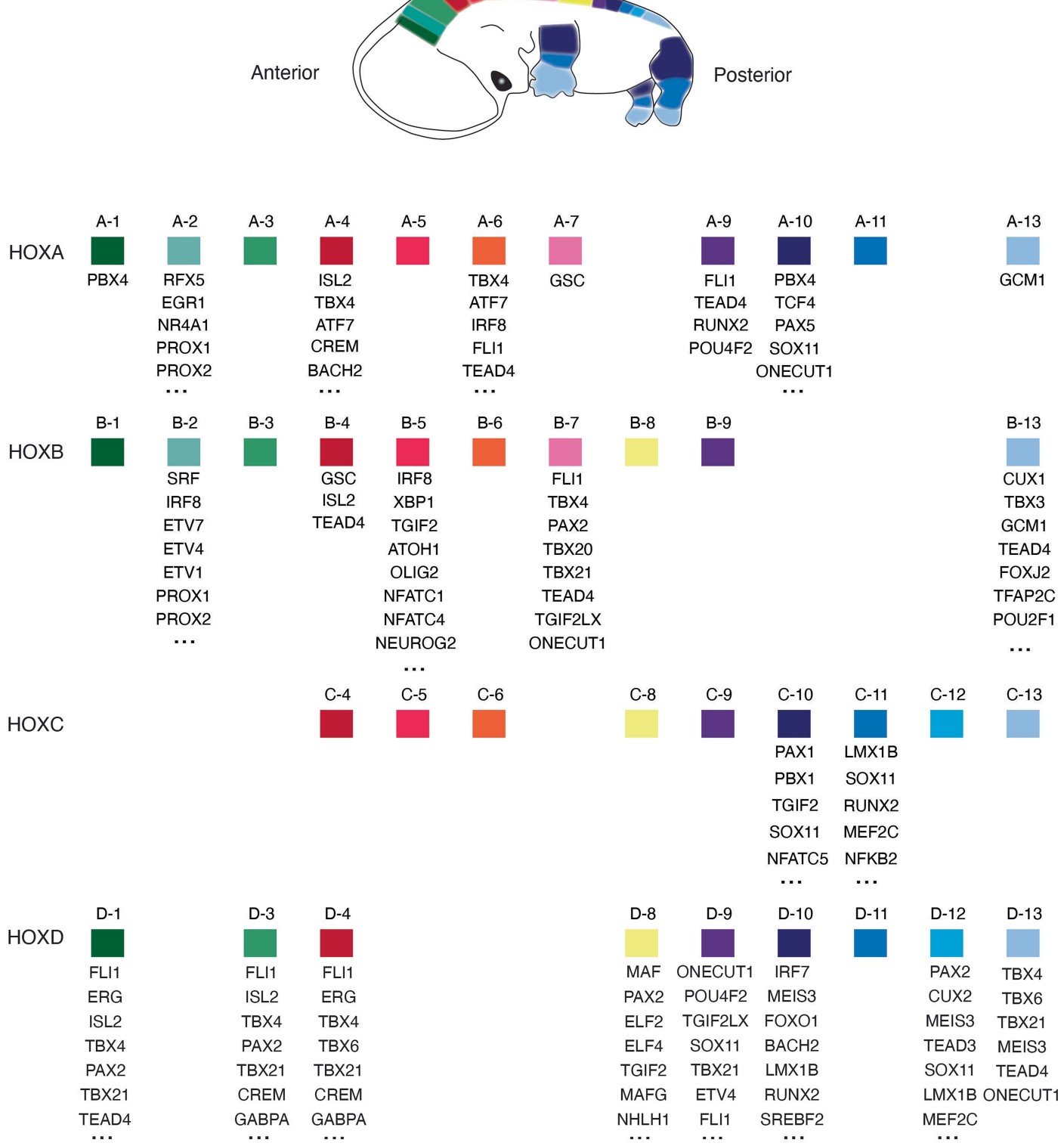

**Extended Data Fig. 5 | Biological roles of cooperative TF pairs in defining the A–P axis.** Top: expression pattern of HOX proteins in the developing mouse embryo. Bottom: TFs that interact with the indicated HOX proteins. For full data, see Supplementary Table 2. Illustration adapted from ref. 28 under a CC BY 4.0 licence.

**Extended Data Fig. 6 | Biological roles of PROX–HOX2 composite motifs.** Composite motifs of PROX–HOX2 that contain a constitutive activator domain drive highly specific expression of lacZ reporter gene in the apical ectodermal ridge, forebrain, midbrain, hindbrain, facial mesenchyme and otic vesicle of transgenic E11.5 mouse embryos.

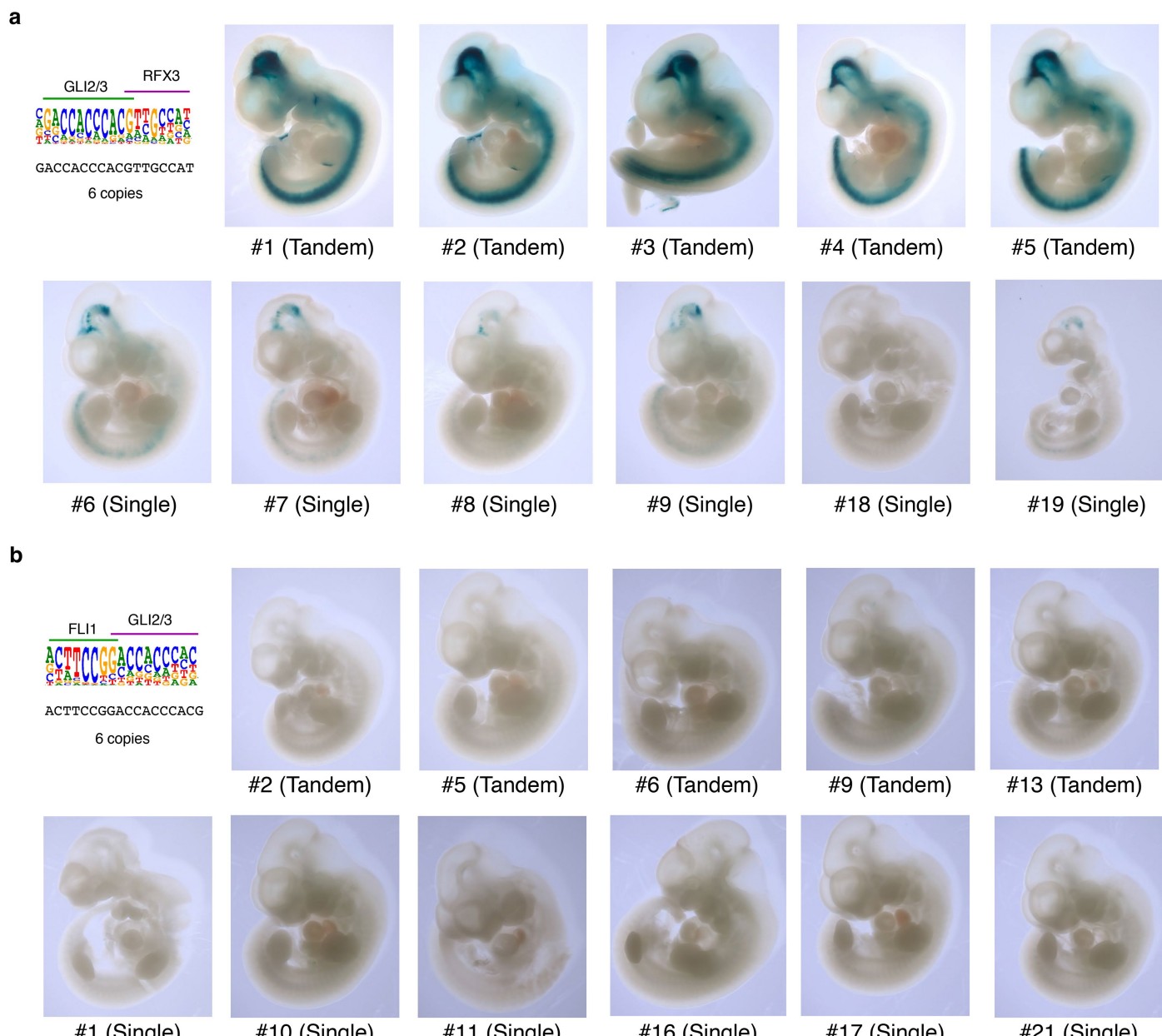

Mouse embryos E11.5

**a**

GLI2/3 RFX3
GACCACCCACGTTGCCAT
6 copies

#1 (Tandem)  #2 (Tandem)  #3 (Tandem)  #4 (Tandem)  #5 (Tandem)

#6 (Single)  #7 (Single)  #8 (Single)  #9 (Single)  #18 (Single)  #19 (Single)

**b**

FLI1 GLI2/3
ACTTCCGGACCACCCACG
6 copies

#2 (Tandem)  #5 (Tandem)  #6 (Tandem)  #9 (Tandem)  #13 (Tandem)

#1 (Single)  #10 (Single)  #11 (Single)  #16 (Single)  #17 (Single)  #21 (Single)

**Extended Data Fig. 7 | Biological roles of GLI3–RFX3 composite motifs.**
**a**, Composite motifs of GLI3–RFX3 that contain a regulated activator domains drive highly specific expression of lacZ reporter gene in the ventral midbrain, neural tube, forebrain and the zone of polarizing activity (ZPA) of the limb buds of transgenic e11.5 mouse embryos. **b**, FLI1–GLI3 motif reporter is silenced and does not express LacZ.

**a**

|  | Motif match is for the true original motif | Motif match is for a scrambled control motif | Total |
|---|---|---|---|
| Motif match is conserved | k | m-k | m |
| Motif match is not conserved | n-k | N+k-n-m | N-m |
| Total | n | N-n | N |

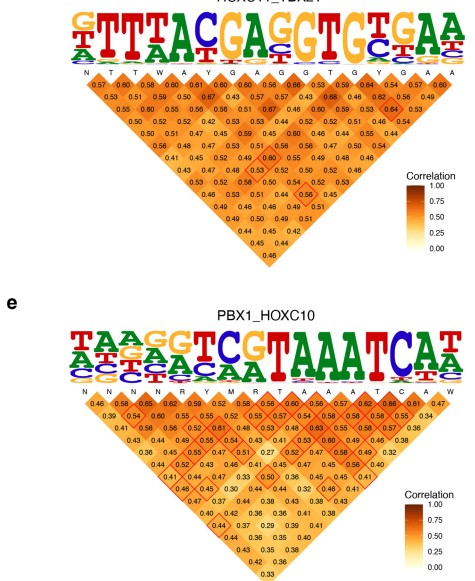

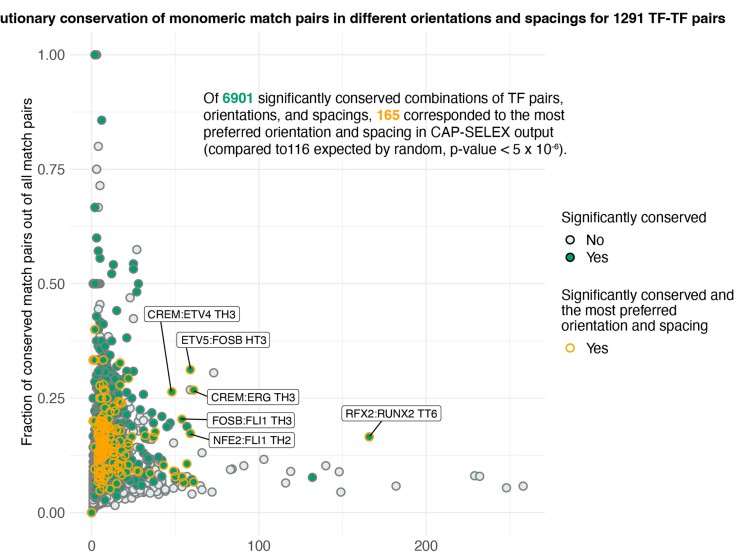

**b** Evolutionary conservation of heterodimeric motif matches

**f** Evolutionary conservation of monomeric match pairs in different orientations and spacings for 1291 TF-TF pairs

Of **6901** significantly conserved combinations of TF pairs, orientations, and spacings, **165** corresponded to the most preferred orientation and spacing in CAP-SELEX output (compared to 116 expected by random, p-value < 5 x 10⁻⁶).

**d** HOXC11_TBX21

**e** PBX1_HOXC10

**Extended Data Fig. 8 | Evolutionary conservation of the matches of the representative heterodimeric motifs. a**, Contingency table for Fisher's exact test to study the enrichment of composite motif matches in human constrained elements. **b**, Conservation of composite motif matches. Fold enrichment (y axis), representing the fraction of conserved matches for the indicated motif relative to the fraction of conserved matches for the control motif, is shown as a function of the number of conserved motif matches among the top ten thousand matches for each TF (x axis). The motifs with significantly conserved matches (FWER < 0.05) are marked by green dots. Significance was determined using Fisher's exact test (one-sided), with p-values adjusted to control FWER. Of these significant motifs, those with a high number of conserved matches among tested matches (>1,000) or high fold enrichment (>1.357) with at least 250 conserved motif matches among the tested matches are indicated by TF–TF pair symbols. Symbols corresponding to new heterodimeric motifs obtained in this study are highlighted in red (composite motif) or blue (spacing motif). Symbols corresponding to motifs from a previous study[9] are highlighted in black. The underlying data, including the corrected p-values, are provided in Supplementary Table S4. **c**, The effect of the motif matching threshold (number of top matches) on the conservation of representative composite and spacing motif matches. Note that matches overlapping exons and repeats were removed. The centre line represents the median, the box

limits are the 25th and 75th percentiles, the whiskers are the minimum and maximum values and outliers are shown as dot. **d,e**, Correlation triangle plots for 8 composite motifs. Heat maps show correlation of phyloP scores across each base position within the matches of the HOXC11–TBX21 (**d**) and PBX1–HOXC10 (**e**) motifs, respectively. For these composite motifs, there was a significant correlation (FDR < 0.05) at least in one position of the bottom part of the correlation triangle. Significant correlations (FDR < 0.05) are highlighted by red squares. **f**, Evolutionary conservation of the orientations and spacings of the paired matches for all combinations of 1291 TF–TF pairs. For each combination, the fraction of conserved match pairs out of all match pairs (y axis) is plotted as a function of the number of conserved matched pairs (x axis). TF–TF pair, orientation, and spacing combinations with a significant number of conserved match pairs are marked by green dots (empirical p-value, FDR < 0.01). Of the 6901 significantly conserved motif match pairs, 165 correspond to the most preferred orientation and spacing in CAP-SELEX (orange; compared to 116 expected by random, p-value < 5 ×10⁻⁶, Fisher's exact test, one-sided). The conserved and most preferred match pairs that have more than 40 conserved matches and a fraction of conserved match pairs higher than 0.15 are labelled by TF–TF pair, orientation, and spacing symbols. The orientations of 6-mers are head-to-tail (HT), tail-to-head (TH), tail-to-tail (TT), and head-to-head (HH).

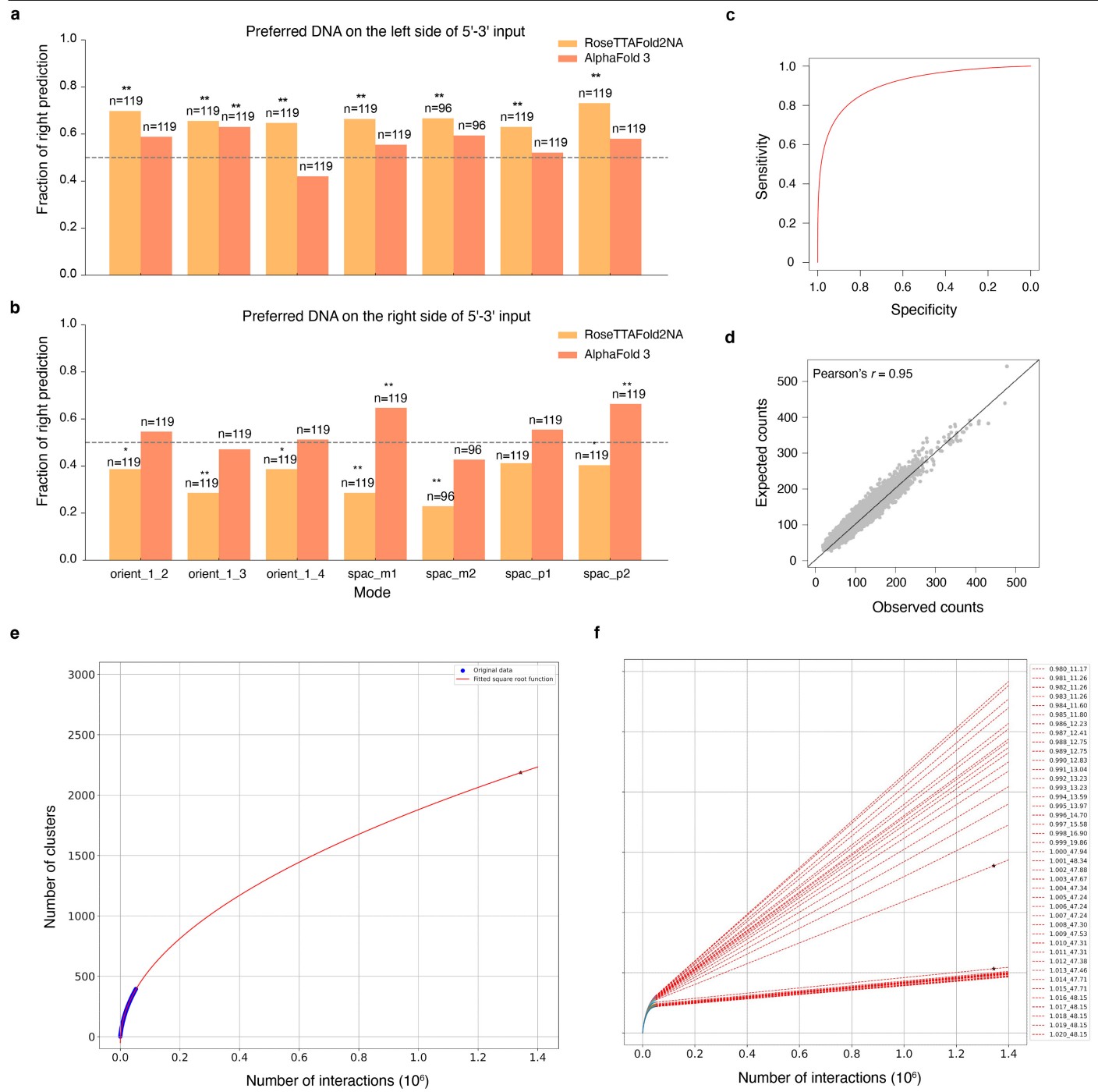

**Extended Data Fig. 9** | See next page for caption.

**Extended Data Fig. 9 | Prediction of TF–TF dimers and estimation of the coverage of the screen. a**,**b**, Bar charts show fraction of correct predictions by AlphaFold 3 (orange) and RoseTTAFold2NA (yellow) for a relatively easy task of computational identification of the sides of DNA fragments containing optimal spacing and orientation motifs for 119 TF–TF pairs from sequences containing those motifs and motifs with different orientations and spacings for the same TF pairs. The control motifs were placed in different orientations (orient) relative to each other, or closer (spac_m) or farther away (spac_p) from each other (see Methods and Supplementary Table 6 for full description). Note that only 96 TF–TF motifs could be placed 2 bp closer to each other (minus 2 bp; spac_m2). In **a**, the optimal motif is placed on the left (upstream) side of the fragment, whereas in **b**, the optimal motif is on the right side. Asterisks indicate a significant difference determined using binomial two-sided test with probability of success equal to 0.5; * indicates p-value less than 0.05, ** indicates p-value less than 0.01 (AlphaFold 3 on the left side of the preferred DNA: orient_1_2 $p = 0.0662913$, orient_1_3 $= 0.0057269$, orient_1_4 $p = 0.0985236$, spac_m1 $p = 0.2712483$, spac_m2 $p = 0.0821931$, spac_p1, $p = 0.7140263$), spac_p2 $p = 0.0985236$; AlphaFold 3 on the right side: orient_1_2 $p = 0.3593583$, orient_1_3 $p = 0.5824938$, orient_1_4 $p = 0.8546300$, spac_m1 $p = 0.0017105$, spac_m2 $p = 0.1842859$, spac_p1 $p = 0.2712483$, and spac_p2 $p = 0.0004453$; RoseTTAFold2NA on the left side: orient_1_2 $p = 0.0000196$, orient_1_3 $p = 0.0008881$, orient_1_4 $p = 0.0017105$, spac_m1 $p = 0.0004453$, spac_m2 $p = 0.0014245$, spac_p1 $p = 0.0057269$, spac_p2 $p = 0.0000005$; RoseTTAFold2NA on the right side: orient_1_2 $p = 0.0167863$, orient_1_3 $p = 0.0000033$, orient_1_4 $p = 0.0167863$, spac_m1 $p = 0.0000033$, spac_m2 $p = 0.0000001$, spac_p1 $p = 0.0662913$, and spac_p2 $p = 0.0432683$). Note that RoseTTAFold2NA shows statistical significance in majority of cases outperforming base model (binomial distribution with probability of success equal to 0.5) in case of placing optimal motif on left side and underperforming it if the optimal motif is placed on the right side. AlphaFold 3 does not have clear bias, and can statistically significantly outperform a random guess in three cases. **c**, A receiver operating characteristic (ROC) curve to assess the performance of mutual information based analysis at calling spacing and orientation preference of TF pairs. **d**, A trained fifth-order Markov Model to predict the frequencies of 8-mers in the initial library (cycle 0). **e**, Square-root function model based on SSTAT. Number of clusters versus number of interactions is shown. Number of clusters is based on the dominating set analysis (SSTAT Ssum similarity $> 1.5 \times 10^{-5}$). A curve fitted to a square-root function ($R^2$ 0.998) is also shown, extrapolated up to $1.4 \times 10^6$ interactions. **f**, Linear approximation by two points (50,000th and the last one) and extrapolation up to $1.4 \times 10^6$ interactions, legend indicates threshold used for cluster generation and percent of predicted clusters. The asterisks indicate number of clusters for all pairwise interaction among the 1,639 TFs.

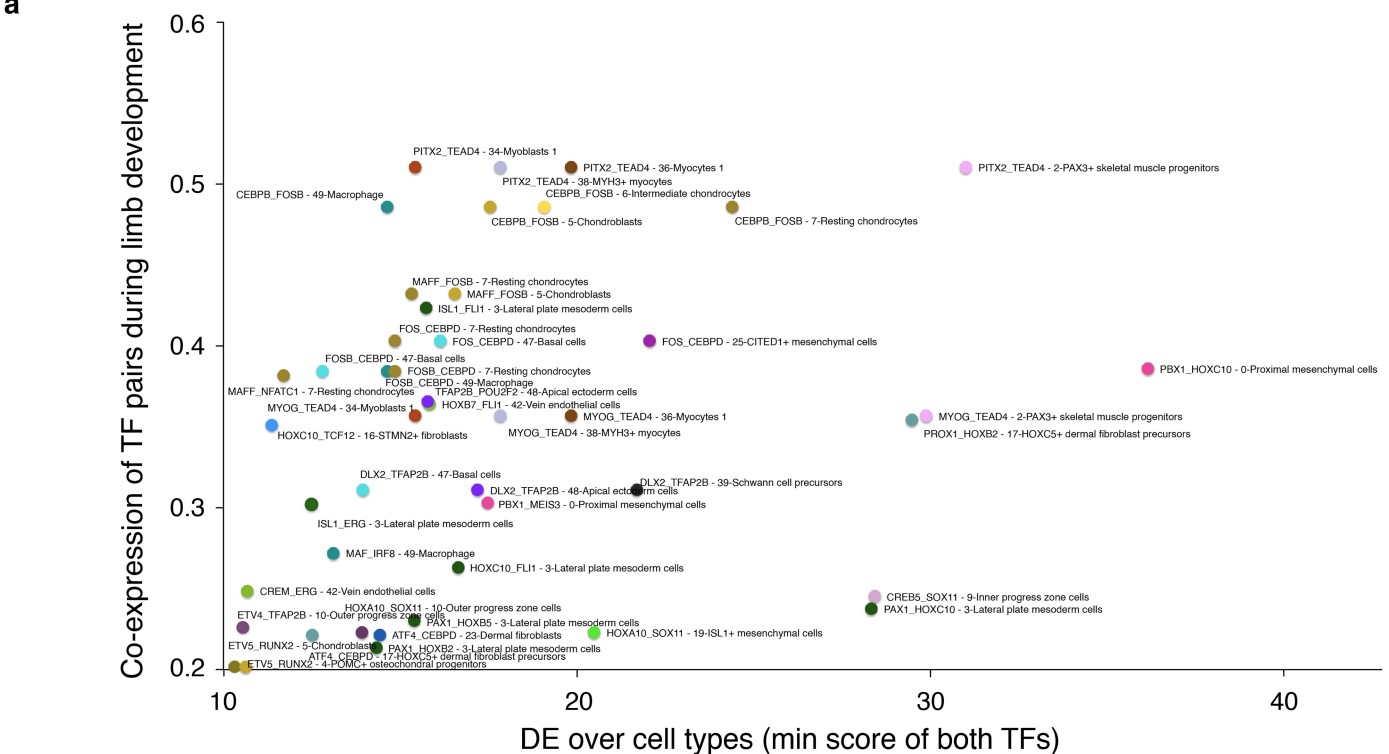

**a**

**b**

**Extended Data Fig. 10 | Single-cell analysis of co-expression of cooperative TF pairs during development. a,b**, Co-expression of cooperative TF pairs across distinct cell types during the processes of limb (**a**) and gut (**b**) formation.

# Reporting Summary

## Statistics

For all statistical analyses, confirm that the following items are present in the figure legend, table legend, main text, or Methods section.

| n/a | Confirmed | |
|---|---|---|
| ☐ | ☒ | The exact sample size (*n*) for each experimental group/condition, given as a discrete number and unit of measurement |
| ☒ | ☐ | A statement on whether measurements were taken from distinct samples or whether the same sample was measured repeatedly |
| ☐ | ☒ | The statistical test(s) used AND whether they are one- or two-sided<br>*Only common tests should be described solely by name; describe more complex techniques in the Methods section.* |
| ☒ | ☐ | A description of all covariates tested |
| ☐ | ☒ | A description of any assumptions or corrections, such as tests of normality and adjustment for multiple comparisons |
| ☐ | ☒ | A full description of the statistical parameters including central tendency (e.g. means) or other basic estimates (e.g. regression coefficient) AND variation (e.g. standard deviation) or associated estimates of uncertainty (e.g. confidence intervals) |
| ☐ | ☒ | For null hypothesis testing, the test statistic (e.g. *F*, *t*, *r*) with confidence intervals, effect sizes, degrees of freedom and *P* value noted<br>*Give P values as exact values whenever suitable.* |
| ☒ | ☐ | For Bayesian analysis, information on the choice of priors and Markov chain Monte Carlo settings |
| ☒ | ☐ | For hierarchical and complex designs, identification of the appropriate level for tests and full reporting of outcomes |
| ☒ | ☐ | Estimates of effect sizes (e.g. Cohen's *d*, Pearson's *r*), indicating how they were calculated |

*Our web collection on statistics for biologists contains articles on many of the points above.*

## Software and code

Policy information about availability of computer code

| Data collection | Crystallographic data were collected using the software developed in synchrotron beam-line ID23-1 in ESRF and listed in Material and Method section |
|---|---|
| Data analysis | MOtifSTAtistic Software Suite v1.1 (MOSTA-SSTAT); tomtom from meme 5.4.1; motifStack v1.38.0; circos v0.69-8; cytoscape v3.10.1; gapped k-mer similarity (https://github.com/jutaipal/motifsimilarity); moods-1.9.4.1; perl v5.34.1; GLPSOL--GLPK LP/MIP Solver 5.0; R 4.3.0 and the compatible packages; MoSBAT-AffiMx v9.0 (https://github.com/csglab/MoSBAT); Crystallographic data analysis: XDS and CCP4 suits 7.1 and 8.0; MR and refinement: Phaser and Refmac5 as implemented in CCP4 and Phenix.refine; Model building: Coot (versions 0.9.6 and o.9.8.92 (EL)) as implemented in CCP4 and Phenix; Structural visualisation: PyMol 2.5.4. custom code links: https://github.com/YinLabTJ/Relative_Affinity_CAP-SELEX; https://github.com/YinLabTJ/MI_CAP-SELEX; https://github.com/MariaOsmala/TFBS; https://github.com/MariaOsmala/TFBS-evolutionary-conservation; https://github.com/i-l-sokolov/RosettaFoldNA_output_analysis. |

For manuscripts utilizing custom algorithms or software that are central to the research but not yet described in published literature, software must be made available to editors and reviewers. We strongly encourage code deposition in a community repository (e.g. GitHub). See the Nature Portfolio guidelines for submitting code & software for further information.

## Data

Policy information about availability of data

All manuscripts must include a data availability statement. This statement should provide the following information, where applicable:
- Accession codes, unique identifiers, or web links for publicly available datasets
- A description of any restrictions on data availability
- For clinical datasets or third party data, please ensure that the statement adheres to our policy

All sequence data are available in ENA, under accession number PRJEB66722. Structure coordinates are available in PDB (entries: 8R7Z, 8R7F, 8BZM, 8BYX, 5NO6, and 5EG0). The ChIP-seq data used in this study are from Encyclopedia of DNA Elements (ENCODE) with accession number ENCSR786QMI, ENCSR918LYT, ENCFF517TKD, ENCFF766VUQ, ENCFF641ZFM, ENCFF992LDJ, ENCFF882AEU, ENCFF093KLR, ENCFF053UZX, ENCFF146SYU, ENCFF164MPE, ENCFF294IGP, ENCFF601WHE, ENCFF451AII, ENCFF934JFA, ENCFF637UJN, ENCFF170IZO, ENCFF172KBM, ENCFF118UKC, ENCFF827VVQ, ENCFF700TAS, ENCFF114CWH, ENCFF440FTA, ENCFF308NZT, ENCFF100KKH, ENCFF885PQR, ENCFF519GKM, ENCFF929XBL, ENCFF839JEO, ENCFF890WJZ, ENCFF788ONH, ENCFF398TIE, ENCFF712HHN, ENCFF314KET, ENCFF399YOO, ENCFF703JHN, ENCFF023EDF, ENCFF488DML, ENCFF603PUS, ENCFF164JZB, ENCFF355HRT, ENCFF605GXY, ENCFF331BSI, ENCFF132AJP, ENCFF314CDT, ENCFF919ZSN, ENCFF011QFM, ENCFF696OTJ, ENCFF358INM, ENCFF759YCY, ENCFF798IVN, ENCFF005YUC, ENCFF647OBG, ENCFF904DOZ, ENCFF634YFK, ENCFF943XDP, ENCFF438PCR, ENCFF718VHT, ENCFF489EME, ENCFF250MUC, ENCFF951BFN, ENCFF493DXA, ENCFF882ARK, ENCFF065RZP, ENCFF970QKS, ENCFF763MXW, ENCFF715WGN, ENCFF118GMS, ENCFF324ELP, ENCFF518EGY, ENCFF566PRZ, ENCFF438IYI, ENCFF592NJN, ENCFF289ZIR, ENCFF594YBN, ENCFF112JVK, ENCFF030DLI, ENCFF091NQE, ENCFF427XNV, ENCFF751VAZ, ENCFF131TYZ, ENCFF987QLI, ENCFF381ULU, ENCFF290IBP, ENCFF463DWW, ENCFF567HPV, ENCFF833ZAC, ENCFF946KCK, ENCFF219OPP (See also Supplementary Table S2). Genome assembly GRCh38 is downloaded from UCSC (ftp://hgdownload.soe.ucsc.edu/apache/htdocs/goldenPath/hg38/bigZips/hg38.fa.masked.gz).

## Research involving human participants, their data, or biological material

Policy information about studies with human participants or human data. See also policy information about sex, gender (identity/presentation), and sexual orientation and race, ethnicity and racism.

| | |
|---|---|
| Reporting on sex and gender | n/a |
| Reporting on race, ethnicity, or other socially relevant groupings | n/a |
| Population characteristics | n/a |
| Recruitment | n/a |
| Ethics oversight | n/a |

Note that full information on the approval of the study protocol must also be provided in the manuscript.

# Field-specific reporting

Please select the one below that is the best fit for your research. If you are not sure, read the appropriate sections before making your selection.

☒ Life sciences ☐ Behavioural & social sciences ☐ Ecological, evolutionary & environmental sciences

For a reference copy of the document with all sections, see nature.com/documents/nr-reporting-summary-flat.pdf

# Life sciences study design

All studies must disclose on these points even when the disclosure is negative.

| | |
|---|---|
| Sample size | We haven't formally calculated the sample sizes required. However, the sequencing depth of CAP-SELEX libraries was set to ensure that at least hundreds of thousands unique reads are available for each TF. Under this sample size, if a TF is binding DNA without restrictions, any non-random pattern of TF binding that has a biologically meaningful effect size (as observed in our study) can only occur with an extremely small p-value. |
| Data exclusions | The failed CAP-SELEX experiments were excluded according to the QC criteria. The criteria define successful CAP-SELEX experiments as having enriched motifs for both TF1 and TF2. The exclusion criteria is established before we perform conclusion-related analyses. |
| Replication | We performed multiple cycles (3) of CAP-SELEX for each TF pair. Each cycle is essentially a replicate of the same experiment. In addition, the whole CAP-SELEX procedure was also repeated for all TFs pairs. For all the reported signals, their enrichment is observed across multiple SELEX cycles, and are reproducible between two or more independent batches of SELEX. |
| Randomization | No grouping was involved in the experiments - no randomization was conducted as a result. |
| Blinding | Most analyses were performed using computational algorithms. Investigators were not blinded |

# Reporting for specific materials, systems and methods

We require information from authors about some types of materials, experimental systems and methods used in many studies. Here, indicate whether each material, system or method listed is relevant to your study. If you are not sure if a list item applies to your research, read the appropriate section before selecting a response.

## Materials & experimental systems

| n/a | Involved in the study |
|-----|------------------------|
| ☒ | Antibodies |
| ☒ | Eukaryotic cell lines |
| ☒ | Palaeontology and archaeology |
| ☐ ☒ | Animals and other organisms |
| ☒ | Clinical data |
| ☒ | Dual use research of concern |
| ☒ | Plants |

## Methods

| n/a | Involved in the study |
|-----|------------------------|
| ☒ | ChIP-seq |
| ☒ | Flow cytometry |
| ☒ | MRI-based neuroimaging |

## Animals and other research organisms

Policy information about studies involving animals; ARRIVE guidelines recommended for reporting animal research, and Sex and Gender in Research

| | |
|---|---|
| Laboratory animals | All animals used in this study were of Mus musculus species and FVB/NCrl strain at ages E11.5 |
| Wild animals | Study did not use wild animals. |
| Reporting on sex | Gender was not identified during collections; however, it is assumed that the groups contained approximately equal numbers of male and female mice |
| Field-collected samples | Study did not use field-collected samples. |
| Ethics oversight | This research complies with all relevant ethical regulations. All animal procedures, including those related to the generation of transgenic animals, were conducted in accordance with the guidelines of the National Institutes of Health (NIH) and approved by the Institutional Animal Care and Use Committee at the University of California, Irvine under protocol no. AUP-23-005. |

Note that full information on the approval of the study protocol must also be provided in the manuscript.

## Plants

| | |
|---|---|
| Seed stocks | n/a |
| Novel plant genotypes | n/a |
| Authentication | n/a |

