## [Peer Review file · Nature]

DNA-guided transcription factor interactions extend human gene regulatory code

Corresponding Author: Professor Jussi Taipale

Version 0:

Reviewer comments:

Referee #1

(Remarks to the Author)

This paper, entitled “Defining the molecular basis of the gene regulatory code”, describes technical updates to a previously-described CAP-SELEX method (Jolma 2015) for determining preferred dimeric TF binding sequence preferences, and also a dramatic expansion of the initial dataset of dimeric transcription factor motifs. The previous paper described 9,400 pairs of TFs, 315 dimeric TF-TF combinations, and 618 distinct dimeric motifs (accounting for multiple spacing and orientation preferences per pair). The dataset in the current paper is nearly an order of magnitude larger, with 58,000 pairs, and uncovers 2,198 dimeric TF combinations encompassing 1,131 distinct dimeric motifs (again, accounting for multiple spacing and orientation preferences per pair). The increase in dimeric motifs does not increase as dramatically presumably due to the fact that paralogs often have related behaviour in these assays.

In addition to describing the methodology and dataset, and confirming trends observed in the previous work are present in greater numbers here, the paper makes several new contributions, including that the composite motifs tend to be active in vivo. The analysis with mouse transgenes is a highlight of the paper. Analyses also explore dimeric TF sites within conserved elements, and their enrichment in ATAC peaks and among co-expressed TFs. The paper also highlights their application in protein-DNA structural analysis and interpretation thereof.

Significance, strengths, and weaknesses:

(1) This paper is a prime example of “stamp collecting” done well, and I think understates its own significance. There are currently fewer than ~1,400 TFs with monomeric binding motifs. This paper single-handedly contributes 2,198 dimeric motifs, and thus will make a major impact on regulatory genomics. It clearly describes an immense amount of work that was performed meticulously and analyzed carefully. The paper will be highly cited for the primary data alone, and in my view, its impact will not be greatly augmented by additional analyses or examples beyond what is already included (pending some modifications, mainly for ease of understanding).

(2) A skeptic might say that the paper only doubles the number of dimeric motifs and acknowledges that they didn’t get all of them, but the individual data points are useful as is the aggregate collection, even if it is incomplete, so this is not a strong criticism. The periodic table was useful even when it was incomplete (and it is still incomplete).

(3) Skeptics might also say that more demonstration of utility is needed, but the same analogy holds here: the periodic table isn’t made more convincing or more useful by careful study of selected elements.

(4) The demonstration that the dimeric motifs are utilized in vivo could be stronger, but I suspect there are theoretical limits to what can be detected within a single genome, and from conservation, given that both the regulatory sites and the TF-TF interactions are expected to evolve continuously. Taipale himself has written on this topic, and Zain and Patel (2021) considered the issue from a different perspective. Greater emphasis could be placed on these arguments, e.g. in the Discussion.

Specific concerns and comments that should be addressed:

(1) The title does not accurately describe what is in the paper. The title is also not supported by information in the paper.

Thus, the title should be changed, so that both requirements are satisfied – it should describe the contents, and if the title itself is a claim, the paper should support that claim. I might suggest that it should contain at least a subset of the keywords “transcription factor”, “motif”, “dimer”, and “specificity”.

(2) The data are highly depleted for C2H2-zf proteins, which are the largest class of TFs in human, and also account for the largest diversity of DNA binding motifs, as the paper notes. Figure 1B says there are 752 C2H2-zfs, so only about 10% are surveyed here – unlike most other classes, where it’s a clear majority. A decade ago it would have been reasonable to question whether they are all bona fide DNA binding proteins, but that is no longer the case, as it is now clear from ChIP-seq and other assays that most of them do bind DNA. The C2H2-zfs are relatively difficult to analyze biochemically, and I suspect that the protein production methods here, as well as the limited complexity of the ligand pool (since C2H2-zfs often bind very long sites), make them hard to analyze using these methods. This long binding sites of the C2H2-zfs may also eliminate the need for cooperative binding in order to achieve high specificity. The claims in the paper (including the title) should be modified or moderated to address these issues.

(3) I do not believe that the data analysis as described could be replicated by anyone outside of the Taipale sphere, because it relies on a computer program written by Taipale earlier in his career that is not commented in an understandable way, and that no other groups have ever been able to dissect or reproduce, to my knowledge. This does not mean they are wrong, but I feel this issue should be discussed in the review process if not the paper itself, and perhaps addressed while the relevant people are still alive.

(4) Treatment of PWMs should be more precise, i.e. are they log-odds, probability, what is the format of the matrices, how was the scanning done (maximum/sum of energy/sum of affinity, what tiling window, etc). Again, considering reproducibility.

(5) The data availability is acceptable, but not optimal. It would be best to deposit the motifs in all of the major motif databases, and it will take some time for the databases to format etc. It’s a small community that is much more cooperative than antagonistic, so it should be straightforward to have database updates coincide with publication of this work.

(6) There is also mention of a new method, mixture-SELEX (Extended Data 3). The sequencing data should be made available as part of the current paper; otherwise, it is not possible to verify the claims.

(7) It is surprising that the structure predictions did not include DNA-protein interactions from AlphaFold3. I think this section of the paper is dispensable, and could be removed, if needed. The structures could be introduced in another way. If it is to be included, however, AlphaFold3 is an obvious omission. The same point applies to the last paragraph of the Discussion.

(8) If dimeric motifs do indeed constitute the “basis of the gene regulatory code”, then one would expect that when interacting partners are co-expressed, then the dimeric motifs should appear in ATAC-seq peaks. Indeed, Figure 6 and the associated text is aimed at asking whether such a phenomenon is observed. Despite rereading it three times and examining the figure carefully, I cannot discern whether this is the case or not. There is clearly something of statistical significance, but what it represents conceptually, and how much of the open chromatin can now be explained, or would be expected to be explained, is opaque to me. I might suggest modifying the text and figure legend to make it more clear what is shown and what it means, literally. It’s an important issue because, if only a small fraction of regulatory sites can now be explained by TF dimer motifs, when the paper claims that we now have motifs for 19-47% of them, then clearly the “gene regulatory code” is not explained by the dimers. That doesn’t mean they aren’t useful, it just means that there must be other mechanisms at work. Which would be another reason to revise the title of the paper.

Clerical details:

(1) The paper would benefit from proofreading, for example the second sentence of the abstract starts with “Human gene regulatory code....” Which should be “The human gene regulatory code....”. There are subject-verb mismatches throughout. Any word processor would flag them. There are also some colloquial phrases that are not wrong, but are uncommon in the scientific literature, e.g. “plenty of partners”.

(2) Search for “zinc fingers” and replace with more specific terms. There are over a dozen types of zinc fingers, including several classes of TFs that are structurally different and have different evolutionary origins (e.g. C2H2, GATA) as well as RNA-binding proteins.

(3) The second sentence of the Results is rather confusing: “We then expressed 160 and 419 differentially tagged prey and bait TF proteins in E.coli, respectively, and screened the binding of each of the preys against a set of 376 bait proteins arranged on 384-well plates (see Methods for details).” I think this means that 419 baits were expressed, but only 376 were used in screens? $160 \times 376 = 60,160$, which is very close to the 58,000 claimed – perhaps the reciprocal and homomeric pairs are counted only once? The uninitiated reader will be grateful for some clarification, so we can see how to arrive at 58,000 pairs. (It may not be necessary to mention that 419 were expressed, if they weren’t all employed in assays).

(4) The section “Global analysis of TF-TF interactions” begins with “To determine if there is a generally preferred spacing between all transcription factors, we first analyzed spacings and orientations between characteristic k-mers contained in the TF motifs”. This description leads the reader to believe that all 58,000 TF-TF pairs were analyzed in this way, since it specifies “all transcription factors”. The resulting figure, however, explains that this analysis was done only for interacting TF pairs. For someone reading this paper for the first time, it should be explained that the characteristic spacing is only for either the 2,189 interacting pairs, or the 1,329 with spacing and orientation preferences (whichever it is).

(Remarks on code availability)

The first of the two URLs listed are dead links. I did not review the third.

A more pressing problem is the previously-published (sort of) "AUTOSEED". I've had two of my staff and one postdoc try to dissect it. One of the staff - probably the best data analyst I've got, with a computer science undergraduate and a PhD in computational biology - told me she would quit her job if she had to keep working on AUTOSEED.

It works, but no one seems to know what it really does over the hood. It's something like 15,000 lines long and I believe Jussi wrote it himself because he enjoyed it. And, it's the starting point for everything here, as far as I can tell.

Referee #2

(Remarks to the Author)

This is an ambitious study that should be of great interest to anyone working on eukaryotic transcriptional regulation. Cooperative interactions between transcription factors (TFs) are an essential aspect of how the cis-regulatory sequence of promoters and enhancers in higher eukaryotes including human is interpreted by the cell. As the authors compellingly show, structure-based prediction methods such as AlphaFold are still far from being able to predict TF-TF interactions in the presence of DNA. As a consequence, there is no substitute for empirical functional datasets such as the CAP-SELEX assay, which is applied in this study on a truly unprecedented scale. Since the protein-protein interfaces between specific pairs of TFs in the context of a TF-TF-DNA complex are often only defined by one or a few side-chain contacts, there is no substitute for comprehensive empirical mapping of many individual TF-TF pairs across the global TF-by-TF matrix. While the current dataset still necessarily leaves a large fraction of this matrix unexplored, the unprecedented increase in coverage of TF-TF pairs that is nevertheless achieved compared to Ref. 10 allows the authors to make many interesting observations about general trends. This comes in addition to the fact that our field's knowledge about specific interacting TF pairs is greatly increased by just this single study. I also appreciated the structural (several new TF-TF-DNA complexes) and functional (reporter assays on specific cooperative TF-TF motifs) validation that was performed as part of this ambitious study.

Specific comments:

- In main text, the authors should be more explicit about the relationship between the present study and their original CAP-SELEX paper (Ref. 10). The difference between the two studies is mostly a matter of scale. The methodology is essentially the same as before, and readers who are not experts in the field should be made aware of this. At the same time, the rightmost Venn diagram inset in Fig1b shows how dramatic the increase in coverage of TF-TF pairs is. This fact should be highlighted in the main text, as it is a major indicator of the importance of the present study, and provides some of the specific information that should be cited to back up the claim at the end of the Discussion that "the work we present here contains more information about TF-TF-DNA interactions than the entire published literature".
- Line 250-251, "in nine cases, the preferred spacing was significantly conserved" : What is the statistical significance of there being nine cases in which the preferred spacing was also the most conserved? What is the expected number given the conservation of the motifs of the individual TFs that make up the pair?
- Line 254, "domination set analysis": The authors should explicitly refer to the Methods section here, since this is not a standard method.
- Line 385-386, "as high affinity motifs are bound before low affinity motifs": This is a kinetic argument that may be overly specific. In a thermodynamic view of TF binding, the order of events by definition does not matter. Do the authors mean to imply that an explicit consideration of binding kinetics is essential to understanding TF function. If not, the wording should be modified, and language consistent with a thermodynamic point of view should be used.
- Line 706, definition of hypergeometric p-value: This equation is incorrect in two distinct ways. First, instead of ratios (e.g., m/k) it should be based on combinatorial factors (" m choose k "). If this is not just a typo in the equation, but the actual expression was used by the authors to compute p-values, the results are incorrect. Second, the definition of the p-value should be based on the "cumulative" hypergeometric distribution $P(X \geq k)$, i.e., there should have been a sum over all values greater than or equal to k ; using only the value of $P(X = k)$. Again, if this is what was actually done, the actual p-values will be greatly underestimated, again leading to seriously incorrect results. The author should clarify both these points, and redo the analyses if needed.
- Related to the point above, there are two more sections in the Methods section where the use of FET is described (lines 804-807 and lines 852-857). Here the details are different, and the various applications of FET may have been performed by different authors, and in different ways (with or without multiple testing correction via FDR, with or without using cumulative HG distribution, different definitions of enrichment, etc). The corresponding authors should have a careful look at this, make the presentation more uniform and consistent, and, if necessary, justify any differences in the use of FET between these three different contexts.
- Fig 2a: The color scale of the heat map should be defined, and especially the meaning of the white color at distance = 3bp.
- Fig 6b: Figure panel was unreadable due to very poor resolution. Several other panels also have poor resolution.

- Fig 6e: This panel is only cited in the Discussion (line 389) to support a biological narrative, but the authors fail to explain what is being shown in this panel. It is not clear what “position” denotes, and how the quantity shown on the y-axis was calculated. This should be addressed.

Referee #3

(Remarks to the Author)

The authors present a giant new dataset examining cooperative transcription factor (TF) binding. They use CAP-SELEX, an in vitro method that can measure TF DNA binding preferences individually, or in tandem. Using this approach, they screen more than 58,000 TF-TF pairs, identifying over 2,000 interacting pairs. The majority of these pairs show preferential DNA binding to specific spacings and/or orientations. They validate the majority of the new “composite motifs” using a related method (mixture-SELEX, where the two TFs are mixed), robustly demonstrating that the newly learned motifs are not specific to the CAP-SELEX method. The biological relevance of these new composite motifs is demonstrated through a variety of methods, including enrichment in ChIP-seq data, conservation across evolutionary time, co-expression analysis, and structural analysis (including several newly solved crystal structures).

In this reviewer’s eyes, the impact of this study is almost entirely due to its unprecedented scale. It is becoming increasingly appreciated that TF binding cooperativity is likely a major key component for understanding gene regulation, but other than a single previous study from this same group (Jolma et al, Nature, 2015, which used the same CAP-SELEX method to examine 9,400 TF-TF pairs), there has been little progress along these fronts. The data presented here are estimated to account for 19-47% of all functioning TF-TF pairs, providing (1) a highly valuable dataset for countless future studies and (2) a first chance to deep-dive into biological roles and functions of TF-TF interactions.

Major comments

1. The title is quite bold and needs to be changed: “Defining the molecular basis of the gene regulatory code.” Much too bold in my opinion, given the contents of the article. I concede that the data generated herein will be vital for future efforts at defining the gene regulatory code, but there is no direct progress along these lines in this study. A title along these lines would imply that this study has “solved the code” by effectively predicting gene expression (or even just TF binding) from DNA sequence, but nothing along those lines is presented here.
2. Fig 3F is compelling – the unique composite motif is highly enriched in overlapping ChIP-seq peaks, but not in the non-overlapping peaks. But this is only one example, and there are >10,000 human TF ChIP-seq datasets available in the public domain. This analysis should be performed for a large set beyond this single pair – it is vital for demonstrating the in vivo relevance of these unique composite motifs and showing they are not in vitro artifacts. I do not expect every pair to have this clean pattern, but there should be a significantly higher number of pairs than random expectation, for example.
3. In the “Conservation of Motifs” section, nine cases are presented where the preferred spacing between TF binding sites was significantly conserved across evolutionary time. This is a lot of tests performed to produce only nine positive results. Some sort of null model is therefore needed to see if this number (9) would have been expected by chance. Please repeat the analysis using a set of “nonsense composite motifs”. You can probably use the “control motifs” you use in the subsequent section. Or you could permute the order of the positions of the motifs (i.e., in order to maintain information content).

Minor comments

1. It is interesting that there are not many interactions between C2H2s and other families. Was there a tendency for C2H2s with longer finger arrays to have fewer interactions than C2H2s with shorter arrays?
2. When calculating the similarity of composite motifs and individual TF motifs, a consensus-sequence-based approach is used here. It would be useful to demonstrate the robustness of these results using a complementary method, which compares motifs based on how they score a large set of DNA sequences. One easy-to-use and straightforward method for this is “MosBat” (PMID:27466627)
3. “the “inner” flanking sequence that overlapped with the flanking sequence of the other TF was often different to what the TF would individually prefer to bind (Fig. 3d).” – This needs to be quantified a bit more – “often” is much too anecdotal. I would suggest that you set a cutoff, and count how frequently the middle is statistically different compared to the “left side” or the “right side” of the core.
4. “In total, only 16 of the tested TFs had strong activator domains. However, 171 TFs interacted with them.” – Need some sort of statistical test (e.g., proportions test) – is this more than expected by chance?
5. Extended Data Fig. 6. is indecipherable in its current format (font too small)

(Remarks on code availability)

I could not examine the code at the first two links (URL not found):

https://github.com/YinLabTJ/MI_CAP-SELEX

https://github.com/YinLabTJ/Relative_Affinity_CAP-SELEX

I did review the code here:
<https://github.com/jutaipal/motifsimilarity>
It is available, clean, and well-documented

Version 1:

Reviewer comments:

Referee #1

(Remarks to the Author)

The resubmission is accompanied by a thoughtful response to each point in the initial reviews. This is true for my own comments, and I believe it is true for the others. The paper itself has not changed fundamentally in its structure or findings, but it has been updated in many smaller ways to address the referee comments.

There are a few aspects which I believe should still be revisited, which I have enumerated below. These all relate to previous reviewer points. Most are writing tasks. Only the last point would require new analyses, but they are computational - parameter changes to operations already described in the manuscript.

(1) The paper still lacks a basic description of what CAP-SELEX is at the outset, and the fact that it was applied previously on a smaller (but still fairly large) scale by this same laboratory. A sentence or two in the last paragraph of the Introduction would be sufficient, something along the lines of "We previously described CAP-SELEX, a method for identifying cooperative binding motifs for pairs of TFs in vitro, and applied it to xxx TF-TF pairs, discovering xxxx. Here, we describe improved methods and analysis of 58,000...."

(2) The last sentence of the Introduction: "we have in this work used high throughput consecutive-affinity purification SELEX (CAP-SELEX)¹⁰ to analyze the cooperativity of binding across more than 58,000 TF-TF-DNA complexes". The vast majority of the 58,000 pairs showed no evidence for cooperative binding, or at least not the expected constraints in spacing and orientation of binding. The last phrase should therefore end something more like "....to identify sequence-mediated, cooperative DNA binding across more than 58,000 TF-TF pairs".

(3) The section "Global analysis of TF-TF interactions" still starts with the sentence "To determine if there is a generally preferred spacing between all transcription factors" and refers to Figure 1c. This is precisely the issue I mentioned in Point 4 on my previous list of clerical issues. Perhaps I did not make it sufficiently clear that the word "all" would describe all 58,000 pairs. That analysis is already done by this point in the manuscript – here, it is repurposed – so having it here is both confusing and incorrect.

I believe that the word "all" should be removed, at the very least. I might also propose that the first two or three sentences could be rewritten for clarity, into something like this: "To determine if there is a generally preferred spacing of individual binding sites between the x,xxx transcription factors that bind cooperatively in pairs, we performed a global analysis of the k-mer mutual information that was used above to identify preferred spacings and orientations between the two TF motifs (Fig. 1c and Extended Data Fig. 2; see also Ref.¹⁰). For each pair of interacting TFs, we first identified the optimal spacing between k-mer sets across all TF-TF pairs in all orientations with respect to each other. We then averaged the mutual information across all of the pairs for which we detected an interaction...." Or something like this.

If I have misunderstood exactly what was done, it is not for lack of trying.

(4) The updated structural analysis section is important, and the addition of Alphafold 3 will be of interest. The descriptions are oriented around anecdotes, however. These are helpful for understanding the process, but not for making conclusions about what works and what doesn't, in general, and how well.

It seems that two types of quantitative assessments have been made on a large enough sample size that it could be turned into a table or small heatmap that compares different structure predictors at different types of tasks, and which would be very helpful to the community. The first task, if I understand correctly, is predicting control and novel (unseen) structures in the presence and absence of DNA. The second is predicting any credible structure given a selected (i.e. real) in vitro dimeric binding site. It seems that the outcomes (one of which is now a large supplementary Excel document) could be summarized in a simple bar graph. The global outcomes would be highly citation-worthy, in my view.

(5) I am still concerned that the conservation/Zoonomia analyses, as well as those in Figure 6, do not give clear numbers that demonstrate the "functional" importance of the study to a broad audience, and thus paint a less flattering picture than they could.

First, these sections might be improved if an estimate of the total number of meaningful motif match instances in the genome were given, rather than focusing on the number of pairs with significant enrichment. That should give an idea of how many of the estimated >1 million nonexonic conserved elements, for example, are explained by cooperative TF dimers. Presumably the number is large, and even with a high background rate, the difference between real and permuted will then still be large.

Second, it seems worth exploring of parameter space with the motif matching – perhaps optimizing conservation or ATAC

enrichment or whatever for individual motifs. The MOODS default threshold seems to underpin these sections, but that isn't some kind of physical constant – it has no relationship to biochemistry - it's just a statistical threshold. Perhaps something simple could be altered in the computational steps (threshold choices on motif score seems obvious) that would result in greater effect sizes. Such an exploration seems particularly important given that the Discussion contains an entire paragraph arguing for the importance of suboptimal motif matches.

(Remarks on code availability)

The rebuttal has a lot to say about the code. Hopefully that's all on GitHub. Anyone can check whether the links work. I think I said enough about this topic earlier.

Referee #2

(Remarks to the Author)

The authors have responded to all previous criticisms in a highly constructive and satisfactory manner. I have no remaining concerns.

Referee #3

(Remarks to the Author)

The authors have done a fine job addressing my comments, no further issues on my end.

(Remarks on code availability)

The code I reviewed is in an appropriate form, with nice documentation, readmes, examples, etc.

Response to reviewers of Xie et al.

In general, all referees had a positive response to our manuscript, stating that our dataset is of “unprecedented scale” (#3) representing “a dramatic expansion of the initial dataset of dimeric transcription factor motifs” (#1), and that our work would be “highly cited” (#1), “of great interest to anyone working on eukaryotic transcriptional regulation” (#2), and provide “a highly valuable dataset for countless future studies” (#3). That said, the reviewers had several comments, mainly concerning data analyses, and description of the methods and algorithms. We have now all responded to all of the reviewers’ comments by performing the requested additional data analyses, releasing all code and sequence data, and rewriting the text to make the manuscript clearer. We have also corrected few minor errors in the text, and revised it throughout to improve grammar and readability.

In the following pages is a point-by-point response to all of the reviewers’ comments; the reviewers comments are in *italic*, our response in roman, and changes to the manuscript are indicated in **bold**.

Referees' comments:

Referee #1 (Remarks to the Author):

This paper, entitled “Defining the molecular basis of the gene regulatory code”, describes technical updates to a previously-described CAP-SELEX method (Jolma 2015) for determining preferred dimeric TF binding sequence preferences, and also a dramatic expansion of the initial dataset of dimeric transcription factor motifs. The previous paper described 9,400 pairs of TFs, 315 dimeric TF-TF combinations, and 618 distinct dimeric motifs (accounting for multiple spacing and orientation preferences per pair). The dataset in the current paper is nearly an order of magnitude larger, with 58,000 pairs, and uncovers 2,198 dimeric TF combinations encompassing 1,131 distinct dimeric motifs (again, accounting for multiple spacing and orientation preferences per pair). The increase in dimeric motifs does not increase as dramatically presumably due to the fact that paralogs often have related behaviour in these assays.

In addition to describing the methodology and dataset, and confirming trends observed in the previous work are present in greater numbers here, the paper makes several new contributions, including that the composite motifs tend to be active in vivo. The analysis with mouse transgenes is a highlight of the paper. Analyses also explore dimeric TF sites within conserved elements, and their enrichment in ATAC peaks and among co-expressed TFs. The paper also highlights their application in protein-DNA structural analysis and interpretation thereof.

Significance, strengths, and weaknesses:

(1) This paper is a prime example of “stamp collecting” done well, and I think understates its own significance. There are currently fewer than ~1,400 TFs with monomeric binding motifs. This paper single-handedly contributes 2,198 dimeric motifs, and thus will make a major impact on regulatory genomics. It clearly describes an immense amount of work that was performed meticulously and analyzed carefully. The paper will be highly cited for the primary data alone, and in my view, its impact will not be greatly augmented by additional analyses or examples beyond what is already included (pending some modifications, mainly for ease of understanding).

(2) A skeptic might say that the paper only doubles the number of dimeric motifs and acknowledges that they didn't get all of them, but the individual data points are useful as is the aggregate collection, even if it is incomplete, so this is not a strong criticism. The periodic table was useful even when it was incomplete (and it is still incomplete).

(3) Skeptics might also say that more demonstration of utility is needed, but the same analogy holds here: the periodic table isn't made more convincing or more useful by careful study of selected elements.

(4) The demonstration that the dimeric motifs are utilized in vivo could be stronger, but I suspect there are theoretical limits to what can be detected within a single genome, and from conservation, given that both the regulatory sites and the TF-TF interactions are expected to evolve continuously. Taipale himself has written on this topic, and Zain and Patel (2021) considered the issue from a different perspective. Greater emphasis could be placed on these arguments, e.g. in the Discussion.

We thank the reviewer for these positive comments. In particular, we appreciate that the reviewer realizes that finding a biological role for all of the TF pairs is beyond the scope of the current work, and that most of the systematic methods to do so are underpowered and/or require single cell data to find the cell type where a particular pair is important. We **have now rewritten the manuscript as described below to take into account these general points as well. We have also noted in the text (e.g. p. 8, l.405-406; p.9, l.407-408 and 431-435) that as the TF-TF interactions are expected be involved in integration of signals, and the motifs are “expensive” in the sense that they are long and have high information content, we do not expect multiple TF-TF pair motifs to be highly frequent in any given cell line.**

Specific concerns and comments that should be addressed:

(1) The title does not accurately describe what is in the paper. The title is also not supported by information in the paper. Thus, the title should be changed, so that both requirements are satisfied – it should describe the contents, and if the title itself is a claim, the paper should support that claim. I might suggest that it should contain at least a subset of the keywords “transcription factor”, “motif”, “dimer”, and “specificity”.

We agree, and **have changed the title to “DNA-guided formation of human transcription factor pairs extends the gene regulatory code”**.

(2) The data are highly depleted for C2H2-zf proteins, which are the largest class of TFs in human, and also account for the largest diversity of DNA binding motifs, as the paper notes. Figure 1B says there are 752 C2H2-zfs, so only about 10% are surveyed here – unlike most other classes, where it’s a clear majority. A decade ago it would have been reasonable to question whether they are all bona fide DNA binding proteins, but that is no longer the case, as it is now clear from ChIP-seq and other assays that most of them do bind DNA. The C2H2-zfs are relatively difficult to analyze biochemically, and I suspect that the protein production methods here, as well as the limited complexity of the ligand pool (since C2H2-zfs often bind very long sites), make them hard to analyze using these methods. This long binding sites of the C2H2-zfs may also eliminate the need for cooperative binding in order to achieve high specificity. The claims in the paper (including the title) should be modified or moderated to address these issues.

We agree, and **have now clarified in the beginning of the Results section that Zinc fingers are underrepresented in the set used, which focuses on TFs that are conserved between mammals (p.2, l.103-105)**. Probably because of their long recognition motifs and/or compact structure, the Zinc fingers we studied also have fewer interacting partners than other TFs on average ($p < 1.51 \times 10^{-93}$). **We now discuss these matters briefly in the Results (p.3, l.152-153) section.**

(3) I do not believe that the data analysis as described could be replicated by anyone outside of the Taipale sphere, because it relies on a computer program written by Taipale earlier in his career that is not commented in an understandable way, and that no other groups have ever been able to dissect or reproduce, to my knowledge. This does not mean they are wrong, but I feel this issue should be discussed in the review process if not the paper itself, and perhaps addressed while the relevant people are still alive.

We have now clarified that the data analysis of dimers is new, and not based on earlier published and available code (p.2, l.111-114 and 118-121). The analysis of dimers and spacing are based on new algorithms written by Yin lab, which are **now available on github** (https://github.com/YinLabTJ/Relative_Affinity_CAP-SELEX;

https://github.com/YinLabTJ/MI_CAP-SELEX). Generation of motifs and logos are the only aspects that uses the motif mining software developed in Taipale and Ukkonen groups that is optimized for SELEX. These tools are also publicly available, and the algorithm used is described, even if split between several prior publications. Briefly, the programs are based on local maxima in Huddinge distance, described in Nitta et al., *eLife* 2015 (see **Figure S1**; <https://elifesciences.org/articles/04837/figures#fig1s1>), and the multinomial algorithm described in Jolma et al., *Cell* 2013 (**Fig. 1B**; <https://ars.els-cdn.com/content/image/1-s2.0-S0092867412014961-gr1.jpg>). The Multinomial algorithm has been implemented twice by different authors using different programming languages, and can be used either via AUTOSEED or MODER (<https://pubmed.ncbi.nlm.nih.gov/29385521/>). The logo generation is open source and available in the MODER2 package (<https://github.com/jttoivon/moder2>). **We have now extended the Methods section to clearly indicate that new programs were used to identify dimers (p.13, l.572-574; p.14, l.626-629), and also more clearly describe how motif mining and logo generation were performed (p.15-16, “Generation of PWMs and motif logos” section). To facilitate understanding of the approach and potential further development by others, we have also now made separately available short programs derived from the “AUTOSEED” code base that count gapped kmers and identify local maxima (https://github.com/jutaipal/kmercount_with_localmax) and generate the PWM (https://github.com/jutaipal/multinomial_motif_generator).** We also tested that standard LLM tools to analyze code (e.g. Claude 3.5 Sonnet) can annotate the code and explain and edit it if necessary. We feel that these changes have materially improved the accessibility and interpretability of the code by the community.

(4) Treatment of PWMs should be more precise, i.e. are they log-odds, probability, what is the format of the matrices, how was the scanning done (maximum/sum of energy/sum of affinity, what tiling window, etc). Again, considering reproducibility.

We have now clarified the format of the PWMs, and that they are based on counts in the Methods section (p.15, l.680-683; p.16, l.693-702) and in the legend to Table S3. The motifs are directly suitable for input to MOODS, a very efficient algorithm for motif matching. As this is sometimes confusing, we have also clarified in Methods (p.16, l.698-702) section and Legend to Table S3 that in the multinomial algorithm, each column is based on a separate alignment of the seed (see Jolma et al., 2013;

Figure 1B), with that particular position masked (to avoid bias caused by matching the base position whose nucleotide distribution is counted). This results in motifs where different columns add to different total values.

(5) The data availability is acceptable, but not optimal. It would be best to deposit the motifs in all of the major motif databases, and it will take some time for the databases to format etc. It's a small community that is much more cooperative than antagonistic, so it should be straightforward to have database updates coincide with publication of this work.

We agree, and **have contacted curators of two major TF motif databases, CisBP and HOCOMOCO.** CisBP has agreed to release all the motifs via their site, and HOCOMOCO will include motifs that are distinctly different from each other (their database only contains such motifs). This should make the motifs available immediately to the public in several widely-usable formats.

(6) There is also mention of a new method, mixture-SELEX (Extended Data 3). The sequencing data should be made available as part of the current paper; otherwise, it is not possible to verify the claims.

We apologize for the oversight, **and have now loaded the reads for mixture-SELEX under accession PRJEB66722 as well. We also indicate that the input (cycle 0) libraries have been uploaded previously under accession PRJEB20112.**

(7) It is surprising that the structure predictions did not include DNA-protein interactions from AlphaFold3. I think this section of the paper is dispensable, and could be removed, if needed. The structures could be introduced in another way. if it is to be included, however, AlphaFold3 is an obvious omission. The same point applies to the last paragraph of the Discussion.

We feel that given the publicity and excitement around structural prediction, it would be important and interesting to the scientific community to see its current limitations (see also Ref#2 general comments). Therefore, we would prefer to keep these analyses as part of the current manuscript. Regarding AlphaFold3, it was released just before our submission, and only allowed 20 predictions per day. **We have now completed the**

AlphaFold 3 analysis, which shows that AlphaFold 3 is also unable to predict TF-TF orientation and spacing preferences (now included in Fig. 5 and discussed in p.6, I.313-326). In identification of preferred spacing and orientation preferences, RoseTTAFold2NA does not perform better than a random guess, as it has a bias of placing the TFs towards 5' end of DNA. Alphafold3 is a bit better in the sense that it does not display the positional bias, and has some weak signal above a random guess in some cases. **This analysis is now included as Figure S11 and discussed in the Results (p.6, I.313-318) and Discussion (p.9, I.439-441) sections.**

(8) If dimeric motifs do indeed constitute the “basis of the gene regulatory code”, then one would expect that when interacting partners are co-expressed, then the dimeric motifs should appear in ATAC-seq peaks. Indeed, Figure 6 and the associated text is aimed at asking whether such a phenomenon is observed. Despite rereading it three times and examining the figure carefully, I cannot discern whether this is the case or not. There is clearly something of statistical significance, but what it represents conceptually, and how much of the open chromatin can now be explained, or would be expected to be explained, is opaque to me. I might suggest modifying the text and figure legend to make it more clear what is shown and what it means, literally. It’s an important issue because, if only a small fraction of regulatory sites can now be explained by TF dimer motifs, when the paper claims that we now have motifs for 19-47% of them, then clearly the “gene regulatory code” is not explained by the dimers. That doesn’t mean they aren’t useful, it just means that there must be other mechanisms at work. Which would be another reason to revise the title of the paper.

We agree, and **have now replaced Figure 6 with a revised high resolution version, changed the title, and moderated the associated text to clarify that also motifs placed in biochemically suboptimal spacings and orientations can be used in regulatory elements (Results, p.7, I.353-355), and that biochemically optimal motifs are not always best for generating biologically optimal input-output functions (Discussion, p.8-9, I.405-408).**

Clerical details:

(1) The paper would benefit from proofreading, for example the second sentence of the abstract starts with “Human gene regulatory code” Which should be “The human

gene regulatory code....”. There are subject-verb mismatches throughout. Any word processor would flag them. There are also some colloquial phrases that are not wrong, but are uncommon in the scientific literature, e.g. “plenty of partners”.

We thank the reviewer for this comment, we initially used standard Word grammar check, which is very limited. To improve, we **have now asked several native English speakers to read the manuscript, and gone through the text carefully using Grammarly, and manually to correct grammar and to remove uncommon expressions.**

(2) Search for “zinc fingers” and replace with more specific terms. There are over a dozen types of zinc fingers, including several classes of TFs that are structurally different and have different evolutionary origins (e.g. C2H2, GATA) as well as RNA-binding proteins.

We agree, and **have now clarified the issues with Zf classes in the text (p.1, l.69; p.1, l.75; p.2, l.104; p.3, l.152 and 154; p.29, l.1245 and 1247).**

(3) The second sentence of the Results is rather confusing: “We then expressed 160 and 419 differentially tagged prey and bait TF proteins in E.coli, respectively, and screened the binding of each of the preys against a set of 376 bait proteins arranged on 384-well plates (see Methods for details).” I think this means that 419 baits were expressed, but only 376 were used in screens? $160 \times 376 = 60,160$, which is very close to the 58,000 claimed – perhaps the reciprocal and homomeric pairs are counted only once? The uninitiated reader will be grateful for some clarification, so we can see how to arrive at 58,000 pairs. (It may not be necessary to mention that 419 were expressed, if they weren’t all employed in assays).

We agree, and **have now clarified this part by only stating the number of pairs screened in the Results section (p.2, l.98), and adding a more detailed explanation to Methods (p.11, l.476), and the list of all pairs tested to Supplementary Table S2.**

(4) The section “Global analysis of TF-TF interactions” begins with “To determine if there is a generally preferred spacing between all transcription factors, we first analyzed spacings and orientations between characteristic k-mers contained in the TF motifs”.

This description leads the reader to believe that all 58,000 TF-TF pairs were analyzed in this way, since it specifies “all transcription factors”. The resulting figure, however, explains that this analysis was done only for interacting TF pairs. For someone reading this paper for the first time, it should be explained that the characteristic spacing is only for either the 2,189 interacting pairs, or the 1,329 with spacing and orientation preferences (whichever it is).

We agree, and have now clarified that the analysis was only performed for the interacting TF-TF pairs (p.2, l.135-137).

Referee #1 (Remarks on code availability):

The first of the two URLs listed are dead links. I did not review the third.

A more pressing problem is the previously-published (sort of) "AUTOSEED". I've had two of my staff and one postdoc try to dissect it. One of the staff - probably the best data analyst I've got, with a computer science undergraduate and a PhD in computational biology - told me she would quit her job if she had to keep working on AUTOSEED.

It works, but no one seems to know what it really does over the hood. It's something like 15,000 lines long and I believe Jussi wrote it himself because he enjoyed it. And, it's the starting point for everything here, as far as I can tell.

We have now clarified that the starting point is the dimer identification code developed by Yin lab (p.2, l.111-114 and 118-121; p.13, l.572-574; p.14, l.626-629

https://github.com/YinLabTJ/Relative_Affinity_CAP-SELEX;

https://github.com/YinLabTJ/MI_CAP-SELEX). We agree that low level C making use of the processor barrel shifter and bit encoding of DNA can be difficult to read and adapt, but unfortunately motif mining code analysing hundreds of millions of sequence reads needs to be highly optimized using low level code. **To help others to develop similar algorithms, we have now also released key subsections of extensively commented code that also refers to the papers describing the algorithms. The key parts perform identification of local maxima kmers (described in Nitta et al., eLife 2015; 707 lines of C; https://github.com/jutaipal/kmercount_with_localmax), and generation of the PWMs from seeds (described in Jolma et al., Cell 2013; 600**

lines; https://github.com/jutaipal/multinomial_motif_generator; also MODER
<https://github.com/jttoivon/MODER> will perform this task and draw motif logos).

Referee #2 (Remarks to the Author):

This is an ambitious study that should be of great interest to anyone working on eukaryotic transcriptional regulation. Cooperative interactions between transcription factors (TFs) are an essential aspect of how the cis-regulatory sequence of promoters and enhancers in higher eukaryotes including human is interpreted by the cell. As the authors compellingly show, structure-based prediction methods such as AlphaFold are still far from being able to predict TF-TF interactions in the presence of DNA. As a consequence, there is no substitute for empirical functional datasets such as the CAP-SELEX assay, which is applied in this study on a truly unprecedented scale. Since the protein-protein interfaces between specific pairs of TFs in the context of a TF-TF-DNA complex are often only defined by one or a few side-chain contacts, there is no substitute for comprehensive empirical mapping of many individual TF-TF pairs across the global TF-by-TF matrix. While the current dataset still necessarily leaves a large fraction of this matrix unexplored, the unprecedented increase in coverage of TF-TF pairs that is nevertheless achieved compared to Ref. 10 allows the authors to make many interesting observations about general trends. This comes in addition to the fact that our field's knowledge about specific interacting TF pairs is greatly increased by just this single study. I also appreciated the structural (several new TF-TF-DNA complexes) and functional (reporter assays on specific cooperative TF-TF motifs) validation that was performed as part of this ambitious study.

Specific comments:

- In main text, the authors should be more explicit about the relationship between the present study and their original CAP-SELEX paper (Ref. 10). The difference between the two studies is mostly a matter of scale. The methodology is essentially the same as before, and readers who are not experts in the field should be made aware of this. At the same time, the rightmost Venn diagram inset in Fig1b shows how dramatic the increase in coverage of TF-TF pairs is. This fact should be highlighted in the main text, as it is a major indicator of the importance of the present study, and provides some of the specific information that should be cited to back up the claim at the end of the Discussion that “the work we present here contains more information about TF-TF-DNA interactions than the entire published literature”.

We agree, and **have now clarified the similarities and differences between the present work and that of Jolma et al., 2015. We have now clarified that the wet lab methodology is very similar, except for the improved affinity tags (p.11, l.509). However, due to the high volume of data, we have developed several new algorithms for analysis of the data. This is also now clarified in the Results (p.2, l.111-114 and 118-121) and Methods (p.13, l.572-574; p.14, l.626-629) sections.**

- Line 250-251, “in nine cases, the preferred spacing was significantly conserved” : What is the statistical significance of there being nine cases in which the preferred spacing was also the most conserved? What is the expected number given the conservation of the motifs of the individual TFs that make up the pair?

We have now corrected the way conservation scores are extracted from PhyloP, and also revised the calculation of the p-value for the conservation of the pair so that it is based on matches to artificial scrambled motifs. This also allowed us to estimate the number of conserved matches that arise by random (p.5, l.263-264). These analyses are now described in the Results (p.5, l.262-266) and Methods sections (p.21-22, l.943-975).

- Line 254, “domination set analysis”: The authors should explicitly refer to the Methods section here, since this is not a standard method.

We thank the reviewer for this remark. **We now refer to the Methods section when discussing the minimum dominating set analysis (p.5, l.267-269).** We also cite our earlier work that has used the minimum dominating set method, and clarify that dominating set is a well-defined graph-theoretical concept (https://en.wikipedia.org/wiki/Dominating_set) that is well suited for identifying representative motifs as the relationships between CAP-SELEX motif similarities forms a graph that cannot be represented as a tree. Therefore, identifying representative motifs by other methods that use tree-based approaches such as clustering results in a loss of information.

- Line 385-386, “as high affinity motifs are bound before low affinity motifs”: This is a kinetic argument that may be overly specific. In a thermodynamic view of TF binding, the order of events by definition does not matter. Do the authors mean to imply that an

explicit consideration of binding kinetics is essential to understanding TF function. If not, the wording should be modified, and language consistent with a thermodynamic point of view should be used.

We thank the reviewer for this comment and apologize for the unclear statement. We agree that we cannot determine kinetics using the methods used, and **have rephrased the argument to refer to equilibrium conditions, with high affinity sites having higher occupancy than low affinity sites (p.9, l.418-420).**

- Line 706, definition of hypergeometric p-value: This equation is incorrect in two distinct ways. First, instead of ratios (e.g., m/k) it should be based on combinatorial factors (“ m choose k ”). If this is not just a typo in the equation, but the actual expression was used by the authors to compute p-values, the results are incorrect. Second, the definition of the p-value should be based on the “cumulative” hypergeometric distribution $P(X \geq k)$, i.e., there should have been a sum over all values greater than or equal to k ; using only the value of $P(X = k)$. Again, if this is what was actually done, the actual p-values will be greatly underestimated, again leading to seriously incorrect results. The author should clarify both these points, and redo the analyses if needed.

We thank the reviewer for noting this and apologise for the errors in the description. The equation indeed had a typo due to conversions between text formats. **This is now fixed (p.18, l.786).** Concerning the second point, we had computed the p-values correctly as the cumulative hypergeometric distribution $P(X \geq k)$ or $P(X \leq k)$ depending on the sign of the log2 fold change of the enrichment. **The calculation of p-values is now explained more clearly in the text (p.17-18, l.775-793).**

- Related to the point above, there are two more sections in the Methods section where the use of FET is described (lines 804-807 and lines 852-857). Here the details are different, and the various applications of FET may have been performed by different authors, and in different ways (with or without multiple testing correction via FDR, with or without using cumulative HG distribution, different definitions of enrichment, etc). The corresponding authors should have a careful look at this, make the presentation more uniform and consistent, and, if necessary, justify any differences in the use of FET between these three different contexts.

We agree that the choices and differences regarding the use of the Fisher's exact test described in different sections could have been better explained and justified. We have **now corrected these sections by clarifying the calculations of the p-values from the cumulative distributions (p.17-18, l.775-793; p.21, l.918-923), by justifying the different definitions of enrichment (p.18, l.788-790; p.21, l.921-923), and the decisions to use multiple-test adjusted p-values and the methods of p-value adjustment (p.18 l.796-799; p.21, l.919-921; FDR or FWER control) throughout.**

- Fig 2a: The color scale of the heat map should be defined, and especially the meaning of the white color at distance = 3bp.

We thank the reviewer for spotting this; **we have now included a color scale to Fig 2a.**

- Fig 6b: Figure panel was unreadable due to very poor resolution. Several other panels also have poor resolution.

We agree, **and have now include a redrawn Fig 6 with higher resolution throughout.**

- Fig 6e: This panel is only cited in the Discussion (line 389) to support a biological narrative, but the authors fail to explain what is being shown in this panel. It is not clear what "position" denotes, and how the quantity shown on the y-axis was calculated. This should be addressed.

We have now redrawn Fig 6e to clarify this point.

Referee #3 (Remarks to the Author):

The authors present a giant new dataset examining cooperative transcription factor (TF) binding. They use CAP-SELEX, an in vitro method that can measure TF DNA binding preferences individually, or in tandem. Using this approach, they screen more than 58,000 TF-TF pairs, identifying over 2,000 interacting pairs. The majority of these pairs show preferential DNA binding to specific spacings and/or orientations. They validate the majority of the new “composite motifs” using a related method (mixture-SELEX, where the two TFs are mixed), robustly demonstrating that the newly learned motifs are not specific to the CAP-SELEX method. The biological relevance of these new composite motifs is demonstrated through a variety of methods, including enrichment in ChIP-seq data, conservation across evolutionary time, co-expression analysis, and structural analysis (including several newly solved crystal structures).

In this reviewer’s eyes, the impact of this study is almost entirely due to its unprecedented scale. It is becoming increasingly appreciated that TF binding cooperativity is likely a major key component for understanding gene regulation, but other than a single previous study from this same group (Jolma et al, Nature, 2015, which used the same CAP-SELEX method to examine 9,400 TF-TF pairs), there has been little progress along these fronts. The data presented here are estimated to account for 19-47% of all functioning TF-TF pairs, providing (1) a highly valuable dataset for countless future studies and (2) a first chance to deep-dive into biological roles and functions of TF-TF interactions.

Major comments

1. The title is quite bold and needs to be changed: “Defining the molecular basis of the gene regulatory code.” Much too bold in my opinion, given the contents of the article. I concede that the data generated herein will be vital for future efforts at defining the gene regulatory code, but there is no direct progress along these lines in this study. A title along these lines would imply that this study has “solved the code” by effectively predicting gene expression (or even just TF binding) from DNA sequence, but nothing along those lines is presented here.

We agree, and have changed the title to “DNA-guided formation of human transcription factor pairs extends the gene regulatory code”.

2. Fig 3F is compelling – the unique composite motif is highly enriched in overlapping ChIP-seq peaks, but not in the non-overlapping peaks. But this is only one example, and there are >10,000 human TF ChIP-seq datasets available in the public domain. This analysis should be performed for a large set beyond this single pair – it is vital for demonstrating the in vivo relevance of these unique composite motifs and showing they are not in vitro artifacts. I do not expect every pair to have this clean pattern, but there should be a significantly higher number of pairs than random expectation, for example.

We agree that ChIP-seq is an additional way to validate the motifs in cell lines where a particular pair is important. We have now performed a systematic analysis of ChIP-seq data for enrichment of composite motifs in overlapping peaks. Although the total volume of ChIP-seq data is indeed large, there are actually not thousands of high-quality ChIP-seq datasets for two TFs corresponding to the identified TF-TF pairs that were performed in the same cell line without any perturbation. Using a filter that required that at least 1,000 peaks had p-value of less than 0.01, we found 93 ENCODE ChIP-seq datasets where both TFs in an interacting pair were chipped in the same cell line. We found that the composite motif matches were enriched in overlapping peaks significantly more than in the individual peaks. **This analysis is now described in Extended Data Fig. 3, and in the Results (p.2, I.125-128) and Methods (p.18-19, I.807-825) sections.**

3. In the “Conservation of Motifs” section, nine cases are presented where the preferred spacing between TF binding sites was significantly conserved across evolutionary time. This is a lot of tests performed to produce only nine positive results. Some sort of null model is therefore needed to see if this number (9) would have been expected by chance. Please repeat the analysis using a set of “nonsense composite motifs”. You can probably use the “control motifs” you use in the subsequent section. Or you could permute the order of the positions of the motifs (i.e., in order to maintain information content).

We agree (see the answers to the comments of Referee #1). **We now utilise the evolutionary conservation of the matches to scrambled control monomeric motifs to define a null distribution for the conservation (described in p.21-22, I.943-975).**

This results in 165 cases where the preferred spacing was also conserved. **This is now stated in the Results section (p.5, l.262-266).**

Minor comments

1. *It is interesting that there are not many interactions between C2H2s and other families. Was there a tendency for C2H2s with longer finger arrays to have fewer interactions than C2H2s with shorter arrays?*

We agree that this question is interesting. However, there are not enough TFs to draw statistically significant inferences. We did find, however, that some Zinc finger TFs including GLIs do relatively commonly interact with other TFs. **This is now noted on p.3, l.152-157.**

2. *When calculating the similarity of composite motifs and individual TF motifs, a consensus-sequence-based approach is used here. It would be useful to demonstrate the robustness of these results using a complementary method, which compares motifs based on how they score a large set of DNA sequences. One easy-to-use and straightforward method for this is “MosBat” (PMID:27466627)*

We have now used MoSBAT to calculate the similarity of composite and individual TF motifs and obtained similar results to the analysis using motif matching to consensus. Both results show that most of the composite motifs were clearly different from individual motifs. **This analysis is now shown as Extended Data Fig. 5, and briefly discussed in the Results section (p.4, l.187-189).**

3. *“the “inner” flanking sequence that overlapped with the flanking sequence of the other TF was often different to what the TF would individually prefer to bind (Fig. 3d).” – This needs to be quantified a bit more – “often” is much too anecdotal. I would suggest that you set a cutoff, and count how frequently the middle is statistically different compared to the “left side” or the “right side” of the core.*

We thank the reviewer for this comment, and have now performed this analysis using two metrics, Jensen Shannon divergence of the overlap region compared to the individual motifs, and Jaccard index of preferred k-mers. Both methods revealed that most motifs do change when they form a composite motif where the individual motifs

overlap. **The results are now included as Extended Data Fig. 5 and described in the Results (p.4, l.194-199) and Methods (p.22-23, l.987-1027) sections.**

4. *“In total, only 16 of the tested TFs had strong activator domains. However, 171 TFs interacted with them.” – Need some sort of statistical test (e.g., proportions test) – is this more than expected by chance?*

We now include statistical analysis, which reveals that it is not more common for TFs to interact with other TFs that have activator domains. However, the fact that there are so many interactions is biologically important. **We have now clarified this in the Results section (p.4, l.231-234 and legend to Fig. 4c).**

5. *Extended Data Fig. 6. is indecipherable in its current format (font too small)*

We have now redrawn ED Fig6 increasing the font size and resolution, and also provide the whole network as a Cytoscape file (Extended Data File S1).

Referee #3 (Remarks on code availability):

I could not examine the code at the first two links (URL not found):

https://github.com/YinLabTJ/MI_CAP-SELEX

https://github.com/YinLabTJ/Relative_Affinity_CAP-SELEX

We apologize for the oversight, and have now released these links to the public.

I did review the code here:

<https://github.com/jutaipal/motifsimilarity>

It is available, clean, and well-documented

Referee #1 (Remarks to the Author):

The resubmission is accompanied by a thoughtful response to each point in the initial reviews. This is true for my own comments, and I believe it is true for the others. The paper itself has not changed fundamentally in its structure or findings, but it has been updated in many smaller ways to address the referee comments.

There are a few aspects which I believe should still be revisited, which I have enumerated below. These all relate to previous reviewer points. Most are writing tasks. Only the last point would require new analyses, but they are computational - parameter changes to operations already described in the manuscript.

(1) The paper still lacks a basic description of what CAP-SELEX is at the outset, and the fact that it was applied previously on a smaller (but still fairly large) scale by this same laboratory. A sentence or two in the last paragraph of the Introduction would be sufficient, something along the lines of “We previously described CAP-SELEX, a method for identifying cooperative binding motifs for pairs of TFs in vitro, and applied it to xxx TF-TF pairs, discovering xxxx. Here, we describe improved methods and analysis of 58,000....”

We agree, and have modified this part according to reviewers' suggestion (p. 2, last paragraph of Introduction).

(2) The last sentence of the Introduction: “we have in this work used high throughput consecutive-affinity purification SELEX (CAP-SELEX)¹⁰ to analyze the cooperativity of binding across more than 58,000 TF-TF-DNA complexes”. The vast majority of the 58,000 pairs showed no evidence for cooperative binding, or at least not the expected constraints in spacing and orientation of binding. The last phrase should therefore end something more like “....to identify sequence-mediated, cooperative DNA binding across more than 58,000 TF-TF pairs”.

We agree, and have modified this part also according to reviewers' suggestion (p. 2, last sentence of Introduction).

(3) The section “Global analysis of TF-TF interactions” still starts with the sentence “To determine if there is a generally preferred spacing between all transcription factors” and refers to Figure 1c. This is precisely the issue I mentioned in Point 4 on my previous list of clerical issues. Perhaps I did not make it sufficiently clear that the word “all” would describe all 58,000 pairs. That analysis is already done by this point in the manuscript – here, it is repurposed – so having it here is both confusing and incorrect.

I believe that the word “all” should be removed, at the very least. I might also propose that the first two or three sentences could be rewritten for clarity, into something like this: “To determine if there is a generally preferred spacing of individual binding sites between the x,xxx transcription factors that bind cooperatively in pairs, we performed a global analysis of the k-mer mutual information that was used above to identify preferred spacings and orientations between the two TF motifs (Fig. 1c and Extended Data Fig. 2; see also Ref.10). For each pair of interacting TFs, we first identified the optimal spacing between k-mer sets across all TF-TF pairs in all orientations with respect to each other. We then averaged the mutual information across all of the pairs for which we detected an interaction....” Or something like this.

If I have misunderstood exactly what was done, it is not for lack of trying.

We agree that this segment was unclear, and **have now rewritten it to clarify that the analysis was based on the set of TFs that displayed orientation and spacing preferences (p. 3, first two sentences of “Global analysis of TF-TF interactions paragraph)**

(4) The updated structural analysis section is important, and the addition of AlphaFold 3 will be of interest. The descriptions are oriented around anecdotes, however. These are helpful for understanding the process, but not for making conclusions about what works and what doesn't, in general, and how well.

It seems that two types of quantitative assessments have been made on a large enough sample size that it could be turned into a table or small

heatmap that compares different structure predictors at different types of tasks, and which would be very helpful to the community. The first task, if I understand correctly, is predicting control and novel (unseen) structures in the presence and absence of DNA. The second is predicting any credible structure given a selected (i.e. real) in vitro dimeric binding site. It seems that the outcomes (one of which is now a large supplementary Excel document) could be summarized in a simple bar graph. The global outcomes would be highly citation-worthy, in my view.

We agree, and **have now included a bar graph to summarize the results of the analyses as a Figure 4b.**

(5) I am still concerned that the conservation/Zoonomia analyses, as well as those in Figure 6, do not give clear numbers that demonstrate the "functional" importance of the study to a broad audience, and thus paint a less flattering picture than they could.

First, these sections might be improved if an estimate of the total number of meaningful motif match instances in the genome were given, rather than focusing on the number of pairs with significant enrichment. That should give an idea of how many of the estimated >1 million nonexonic conserved elements, for example, are explained by cooperative TF dimers. Presumably the number is large, and even with a high background rate, the difference between real and permuted will then still be large.

Second, it seems worth exploring of parameter space with the motif matching – perhaps optimizing conservation or ATAC enrichment or whatever for individual motifs. The MOODS default threshold seems to underpin these sections, but that isn't some kind of physical constant – it has no relationship to biochemistry - it's just a statistical threshold. Perhaps something simple could be altered in the computational steps (threshold choices on motif score seems obvious) that would result in greater effect sizes. Such an exploration seems particularly important given that the Discussion contains an entire paragraph arguing for the importance of suboptimal motif matches.

We agree, and **have now added the conserved motif match counts to p. 6, paragraph “Conservation of motifs”**. We have also clarified that the threshold is not completely arbitrary and not based on MOODS default threshold. Rather, we have selected a fixed number of matches for each TF or pair (Methods, p.21, third paragraph.). This approach is more representative of physical reality, even if the precise meaningful threshold for each factor still is not identified. On aggregate, conservation increases when motif match threshold is made more strict (see below). **We have also now included the threshold-analysis in Extended Data Fig.8c.**